# WHAT DOES IT MEAN TO BE A TRANSFORMER?
# INSIGHTS FROM A THEORETICAL HESSIAN ANALYSIS

**Weronika Ormaniec**[*]
ETH Zürich
Zürich, Switzerland

**Felix Dangel**
Vector Institute
Toronto, Canada

**Sidak Pal Singh**
ETH Zürich
Zürich, Switzerland

## ABSTRACT

The Transformer architecture has inarguably revolutionized deep learning, over-taking classical architectures like multi-layer perceptrons (MLPs) and convolutional neural networks (CNNs). At its core, the attention block differs in form and functionality from most other architectural components in deep learning—to the extent that, in comparison to MLPs/CNNs, Transformers are more often accompanied by adaptive optimizers, layer normalization, learning rate warmup, etc. The root causes behind these outward manifestations and the precise mechanisms that govern them remain poorly understood. In this work, we bridge this gap by providing a fundamental understanding of *what distinguishes the Transformer from the other architectures—grounded in a theoretical comparison of the (loss) Hessian.* Concretely, for a single self-attention layer, **(a)** we first entirely derive the Transformer's Hessian and express it in matrix derivatives; **(b)** we then characterize it in terms of data, weight, and attention moment dependencies; and **(c)** while doing so further highlight the important structural differences to the Hessian of classical networks. Our results suggest that various common architectural and optimization choices in Transformers can be traced back to their highly non-linear dependencies on the data and weight matrices, which vary heterogeneously across parameters. Ultimately, our findings provide a deeper understanding of the Transformer's unique optimization landscape and the challenges it poses.

## 1 INTRODUCTION AND RELATED WORK

The Transformer architecture (Vaswani et al., 2017) has shown remarkable success across natural language processing (Devlin et al., 2018; Radford et al., 2018) and vision (Dosovitskiy et al., 2020) tasks. In particular, its self-attention mechanism has become a mainstay of modern architectures, enabling parallelization while effectively capturing long-range and modality-agnostic dependencies. Yet, despite their impressive performance, a significant gap remains in our understanding of the properties of Transformer-based models relative to traditional architectures such as multi-layer perceptrons (MLPs, Rosenblatt, 1958) or convolutional neural networks (CNNs, LeCun et al., 1998).

**Transformers' unique architectural built.** Compared to the classical architectures, Transformers are unique in several ways. Firstly, the data (tokens) enters the Transformer, through the self-attention, multiple times. Secondly, the softmax nonlinearity inside self-attention differs from the piece-wise linear activations, like ReLU (Glorot et al., 2011). Thirdly, the query-key attention incorporates two (instead of one) directly multiplied weight matrices within a single architectural block.

**The side-factors making Transformers' unique architecture work.** These architectural differences render practical approaches for training a Transformer distinct compared to classical nets (Popel & Bojar, 2018; Bengio, 2012). E.g., Transformers are usually trained with adaptive optimizers like Adam(W) (Kingma & Ba, 2015; Loshchilov & Hutter, 2019) and require architectural extensions such as skip connections (He et al., 2016) and layer norm (Xiong et al., 2020), learning rate warm-up (Goyal et al., 2017), and using different weight initializations (Huang et al., 2020).

**Aim.** Given these *outward and indirect* manifestations of the Transformer's presence, it is unclear how these are *explicitly* triggered due to particular loss landscape geometry endowed by Transformers. To address this, we theoretically investigate the Transformer's loss landscape by analyzing, in detail, the structure of the Hessian matrix as well as its data dependency and behavior across layers.

---

[*]Correspondence to wormaniec@ethz.ch, fdangel@vectorinstitute.ai, contact@sidakpal.com.

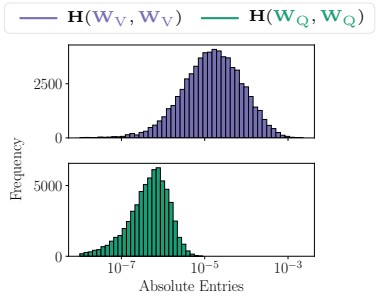
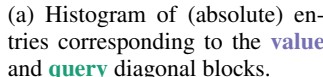
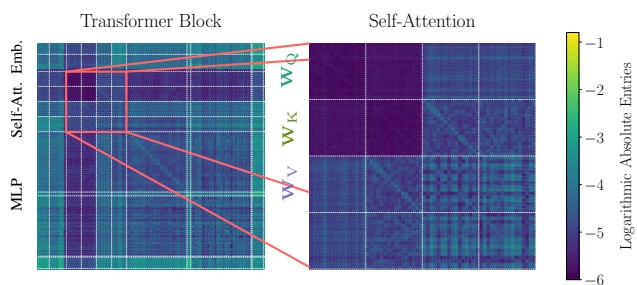

(a) Histogram of (absolute) entries corresponding to the **value** and **query** diagonal blocks.

(b) On the left, the Hessian matrix for a minimal Transformer, and, on the right, the zoomed-in block w.r.t. **query**, **key**, and **value** parameters.

Figure 1: **Disparity in the Hessian blocks of a Transformer seen quantitatively and qualitatively.** We used a single-block GPT-2 Transformer at initialization applied to the next token prediction task (for details see appendix F). We observe block heterogeneity in the magnitudes of Hessian entries—those of the query block are significantly smaller than those of the value block.

**Prior Hessian-related work.** The Hessian matrix is a fundamental object for optimization (Martens, 2010; Schaul et al., 2013; Cohen et al., 2021), generalization (Keskar et al., 2016; Jiang et al., 2019), and more (LeCun et al., 1989; Singh & Alistarh, 2020). *(a) Empirical work:* For traditional architectures, the Hessian has received a lot of attention through empirical studies on its bulk-outlier spectrum and eigenvalue density (Sagun et al., 2016; 2017; Ghorbani et al., 2019; Papyan, 2020; Yao et al., 2020) and in turn how it is affected by architectural components and how it changes during training. *(b) Theoretical studies in classical architectures:* More recently, several theoretical works have also analyzed, in detail, the Hessian structure and rank (Singh et al., 2021; 2023), beyond Random-Matrix-Theory based approximations (Pennington & Bahri, 2017). However, this latter line of work has been restricted to fully-connected and convolutional architectures.

**Transformer-related Hessian studies.** Park & Kim (2022) provide empirical evidence that the multi-head self-attention mechanism in Transformers leads to a more non-convex but smoother loss landscape than that of CNNs, as the Hessian has more negative eigenvalues, but of smaller magnitude. Zhang et al. (2024) studied the Hessian spectra of Transformers to explain the need for adaptive optimizers for successful training. They empirically showed that although the full Hessian spectra of Transformers are quite similar to those of CNNs, the Hessian diagonal blocks in Transformers are much more heterogeneous—a possible cause for the need for adaptive optimizers. *Barring these few empirical studies, to the best of our knowledge, a theoretical treatment of the Hessian in Transformers remains lacking.*

**Goals and contributions.** By theoretically analyzing the Hessian of Transformers, we aim to identify and explicitly state the differences to classical architectures. *We believe this provides the foundation for a deeper understanding of the unique loss landscape features and challenges the Transformer poses.* Our detailed contributions are:

**(i)** We derive the Hessian of a single self-attention layer and exhibit the structure contained within its well-known positive-definite and indefinite parts (theorems 3.1 and 3.2).

**(ii)** We categorize its dependencies into data, weights, and attention moments, find that the Hessian is highly non-linear and heterogeneous for different parameter groups within a self-attention block, and show how various Transformer-related tricks can be understood through these characteristics.

**(iii)** We explicitly establish how individual components of self-attention, like the softmax and query-key-parameterization, result in a more non-linear and heterogeneous structure within the Hessian (see fig. 1), then contrast the Transformer's Hessian with that of traditional architectures and uncover pronounced differences.

## 2  SETUP AND BACKGROUND

**Single self-attention layer.** We consider a single self-attention layer with a slightly generalized definition, which will later enable us to investigate the impact of specific components. The layer maps token embeddings $\mathbf{X} \in \mathbb{R}^{L \times d_V}$ of a sequence of length $L$ with embedding dimension $d_V$

$$\mathbf{F} \qquad \operatorname{vec}_r(\mathbf{F}) \qquad \frac{\partial \mathbf{F}}{\partial \mathbf{W}_i} := \frac{\partial \operatorname{vec}_r \mathbf{F}}{\partial \operatorname{vec}_r \mathbf{W}_i} \qquad \operatorname{vec}_r\left(\frac{\partial \mathbf{F}}{\partial \mathbf{W}_i}\right) \qquad \frac{\partial^2 \mathbf{F}}{\partial \mathbf{W}_i \partial \mathbf{W}_j} := \frac{\partial \operatorname{vec}_r \frac{\partial \mathbf{F}}{\partial \mathbf{W}_i}}{\partial \operatorname{vec}_r \mathbf{W}_j}$$

Figure 2: Construction of the second derivative matrix of a matrix-valued network $\mathbf{F}$. Taking the second derivative of $\mathbf{F}$ using row-wise vectorization and numerator layout is equivalent to computing the second derivatives of each entry separately and stacking them into a column block matrix.

into a new sequence $\mathbf{F}(\mathbf{X}) \in \mathbb{R}^{L \times d_V}$ by computing the values $\mathbf{X}\mathbf{W}_V$ using value weights $\mathbf{W}_V \in \mathbb{R}^{d_V \times d_V}$ and re-weighting them with the self-attention matrix (or map) $\mathbf{A}(\mathbf{X}) \in \mathbb{R}^{L \times L}$, i.e.

$$\mathbf{F}(\mathbf{X}) = \mathbf{A}(\mathbf{X})\mathbf{X}\mathbf{W}_V \quad \text{where} \quad \mathbf{A}(\mathbf{X}) = a\left(\mathbf{T}(\mathbf{X})\right) . \tag{1}$$

Here, we decompose the attention map into two components, the query-key similarity transformation $\mathbf{T}(\mathbf{X}) \in \mathbb{R}^{L \times L}$ which may introduce additional trainable parameters, and the activation function $a : \mathbb{R}^{L \times L} \to \mathbb{R}^{L \times L}$. The classical self-attention from Vaswani et al. (2017) uses $\mathbf{T}(\mathbf{X}) = \mathbf{X}\mathbf{W}_Q\mathbf{W}_K^\top\mathbf{X}^\top / \sqrt{d_K}$, where $a = \operatorname{softmax}$ (row-wise), with learnable query and key weight matrices $\mathbf{W}_Q, \mathbf{W}_K \in \mathbb{R}^{d_V \times d_K}$. Further, our single self-attention layer from eq. (1) feeds into a loss function $\ell : \mathbb{R}^{L \times d_V} \times \mathbb{R}^{L \times d_V} \to \mathbb{R}$ that measures the discrepancy between the predicted sequence $\mathbf{F}(\mathbf{X})$ and the sequence labels $\mathbf{Y}$. We will assume square loss (where $\| \cdot \|_F$ denotes the Frobenius norm of a matrix), $\ell(\mathbf{F}(\mathbf{X}), \mathbf{Y}) = \|\mathbf{F}(\mathbf{X}) - \mathbf{Y}\|_F^2 / (L d_V)$.

**Hessian, block structure, and Gauss-Newton decomposition.** Our goal is to compute the full Hessian of the loss w.r.t. the learnable, matrix-shaped, parameters $\{\mathbf{W}_i \in \mathbb{R}^{q_i \times p_i}\}_i$ of the attention layer (for instance $i \in \{K, Q, V\}$ for canonical self-attention). To break down this task, first notice that this matrix consists of blocks $\partial^2(\ell \circ \mathbf{F})/\partial \mathbf{W}_i \partial \mathbf{W}_j$. Further, the Hessian decomposes into two components as a consequence of the chain rule applied to the functional split $\ell \circ \mathbf{F}$,

$$\frac{\partial^2 (\ell \circ \mathbf{F})}{\partial \mathbf{W}_i \partial \mathbf{W}_j}(\cdot) = \underbrace{\frac{\partial \mathbf{F}}{\partial \mathbf{W}_i}(\cdot)^\top \frac{\partial^2 \ell}{\partial \mathbf{F}^2}(\mathbf{F}(\cdot)) \frac{\partial \mathbf{F}}{\partial \mathbf{W}_j}(\cdot)}_{\substack{\mathbf{F}\text{-outer-product Hessian} \\ \mathbf{H}_o(\mathbf{W}_i, \mathbf{W}_j)}} + \underbrace{\left(\frac{\partial \ell}{\partial \mathbf{F}}(\mathbf{F}(\cdot)) \otimes \mathbf{I}_{p_i q_i}\right) \frac{\partial^2 \mathbf{F}}{\partial \mathbf{W}_i \partial \mathbf{W}_j}(\cdot)}_{\substack{\mathbf{F}\text{-functional Hessian} \\ \mathbf{H}_f(\mathbf{W}_i, \mathbf{W}_j)}} . \tag{2}$$

Equation (2) is known as Gauss-Newton decomposition and such splits can be widely seen in the literature (Sagun et al., 2017; Kunstner et al., 2019; Martens, 2020; Dangel et al., 2020; Singh et al., 2021). We adopt the terminology provided in Singh et al. (2021; 2023) by referring to the term containing second-order derivatives of $\ell$ as outer-product Hessian (often called generalized Gauss-Newton (Schraudolph, 2002) matrix), and to the term capturing second-order derivatives of the network as functional Hessian. To emphasize the split at which the decomposition through the chain rule was applied, we added the prefix $\mathbf{F}$ above. This will later become useful to identify expressions in the Hessian as stemming from different splits (see section 3 and appendix D). But until then and unless stated otherwise, we will consider the split $\ell \circ \mathbf{F}$ and thus omit specifying this hereafter. For classical self-attention, this yields the Hessian decomposition $\mathbf{H} = \mathbf{H}_o + \mathbf{H}_f$ with

$$\mathbf{H}_\bullet = \begin{bmatrix} \mathbf{H}_\bullet(\mathbf{W}_Q, \mathbf{W}_Q) & \mathbf{H}_\bullet(\mathbf{W}_Q, \mathbf{W}_K) & \mathbf{H}_\bullet(\mathbf{W}_Q, \mathbf{W}_V) \\ \mathbf{H}_\bullet(\mathbf{W}_K, \mathbf{W}_Q) & \mathbf{H}_\bullet(\mathbf{W}_K, \mathbf{W}_K) & \mathbf{H}_\bullet(\mathbf{W}_K, \mathbf{W}_V) \\ \mathbf{H}_\bullet(\mathbf{W}_V, \mathbf{W}_Q) & \mathbf{H}_\bullet(\mathbf{W}_V, \mathbf{W}_K) & \mathbf{H}_\bullet(\mathbf{W}_V, \mathbf{W}_V) \end{bmatrix} \tag{3}$$

for $\bullet \in \{o, f\}$ and where $\mathbf{H}_\bullet(\mathbf{W}_i, \mathbf{W}_j) = (\mathbf{H}_\bullet(\mathbf{W}_j, \mathbf{W}_i))^\top$ due to symmetry.

**Matrix calculus.** So far, we have slightly abused notation and did not define formally what it means to take derivatives of a matrix-valued object w.r.t. a matrix-shaped argument. We will follow the recipe of matrix calculus (Magnus & Neudecker, 2019) which reduces those derivatives to the vector case by introducing a flattening convention (we will use row-stacking, indicated by the operation $\operatorname{vec}_r$). Consider two vectors $\mathbf{a} \in \mathbb{R}^A$ and $\mathbf{b}(\mathbf{a}) \in \mathbb{R}^B$. The Jacobian $\partial \mathbf{b}/\partial \mathbf{a} \in \mathbb{R}^{B \times A}$ collects the first-order derivatives such that $[\partial \mathbf{b}/\partial \mathbf{a}]_{i,j} = \partial b_i / \partial a_j$. The Hessian $\partial^2 \mathbf{b}/\partial \mathbf{a}^2 \in \mathbb{R}^{BA \times A}$ is a column block matrix that concatenates the Hessians for each entry of $\mathbf{b}$ w.r.t. $\mathbf{a}$; equivalent to flattening the Jacobian, then differentiating it a second time. To generalize these definitions

to matrix-valued objects, we simply flatten all arguments before differentiation (see fig. 2 for a visualization). Applied to the expressions in eq. (2), we have,

$$\frac{\partial \mathbf{F}}{\partial \mathbf{W}_i} := \frac{\partial \operatorname{vec}_r \mathbf{F}}{\partial \operatorname{vec}_r \mathbf{W}_i} \in \mathbb{R}^{Ld_V \times p_i q_i} \quad \text{and} \quad \frac{\partial^2 \mathbf{F}}{\partial \mathbf{W}_i \partial \mathbf{W}_j} := \frac{\partial \operatorname{vec}_r \frac{\partial \mathbf{F}}{\partial \mathbf{W}_i}}{\partial \operatorname{vec}_r \mathbf{W}_j} \in \mathbb{R}^{Ld_V p_i q_i \times p_j q_j} . \quad (4)$$

For the square loss discussed above, the Jacobian $\partial \ell / \partial \mathbf{F} \in \mathbb{R}^{1 \times Ld_V}$ and Hessian $\partial^2 \ell / \partial^2 \mathbf{F} \in \mathbb{R}^{Ld_V \times Ld_V}$ w.r.t. the network's prediction simplify to $2 \operatorname{vec}_r (\mathbf{F}(\cdot) - \mathbf{Y})^\top / (Ld_V)$ and $2\mathbf{I}_{Ld_v} / (Ld_V)$ (i.e., a scaled identity matrix) respectively.

**Hessian of MLPs and CNNs.** Since one of our main interests is stating explicit differences between the Transformer Hessian and that of traditional architectures, we briefly recap the Hessian structure from Singh et al. (2021; 2023), for the case of linear MLPs, $\mathbf{F}(\mathbf{x}) = \mathbf{W}_L \cdots \mathbf{W}_1 \mathbf{x}$. As an illustration, we will only consider the outer-product Hessian and look at its diagonal block corresponding to the $i$-th layer parameter matrix, $\mathbf{W}_i$, which takes the form

$$\mathbf{H}_\mathrm{o}(\mathbf{W}_i, \mathbf{W}_i) = \mathbf{W}_{i+1}^\top \cdots \mathbf{W}_L^\top \mathbf{W}_L \cdots \mathbf{W}_{i+1} \otimes \mathbf{W}_{i-1} \cdots \mathbf{W}_1 \mathbf{\Sigma_{xx}} \mathbf{W}_1^\top \cdots \mathbf{W}_{i-1}^\top ,$$

with $\mathbf{\Sigma_{xx}} = \mathbf{E}\left[\mathbf{xx}^\top\right]$, the uncentered data covariance over the dataset. Observe that the above expression depends quadratically on the data through $\mathbf{\Sigma_{xx}}$. Likewise, the functional Hessian for MLPs, has a quadratic dependence on the data, but through a different matrix which carries the covariance of the residuals $\delta_{\mathbf{x},\mathbf{y}} = \mathbf{F}(\mathbf{x}) - \mathbf{y}$ with the input $\mathbf{x}$. Importantly, the Hessian diagonal blocks are entirely comprised of the contribution from the outer-product Hessian, as the functional Hessian has a block-hollow structure (zero diagonal blocks), which also extends to any piecewise nonlinearity in the sense of almost everywhere.

A notable difference from our setup is that these works aggregate the Hessian over the dataset, while, for brevity, we focus on a single sample setting.[1] As a side-benefit, this allows us to cast their (non-sequential) setup one-to-one here *by considering their whole dataset in the form of a single sequence, where the MLP is applied separately to each element of the sequence.* This point of view helps facilitate a precise comparison between the two settings.

## 3    EXACT STRUCTURE OF THE SELF-ATTENTION HESSIAN

We start by studying the Hessian of standard self-attention with square loss. We present our main results regarding the exact Hessian in the form of two theorems, each of them targeting one term in the Gauss-Newton decomposition from eq. (2), further broken down into parameter blocks from eq. (3). Later, our goal will be to analyze and interpret them further through simplifications. For brevity, we omit the blocks w.r.t. the key weight matrix as they are essentially symmetric to the ones w.r.t. the query weight matrix (modulo differences in arrangement) and defer the proofs to the appendix.

> **Theorem 3.1. Outer-product Hessian $\mathbf{H}_\mathrm{o}$.**    For a single self-attention layer, eq. (1), with classical self-attention that feeds into the square loss, the blocks of $\mathbf{H}_\mathrm{o}$ are
>
> $$\mathbf{H}_\mathrm{o}\left(\mathbf{W}_\mathrm{V}, \mathbf{W}_\mathrm{V}\right) = \frac{2}{Ld_V} \mathbf{M}_1^\top \mathbf{M}_1 \otimes \mathbf{I}_{d_V},$$
>
> $$\mathbf{H}_\mathrm{o}\left(\mathbf{W}_\mathrm{Q}, \mathbf{W}_\mathrm{Q}\right) = \frac{2}{Ld_V d_K} \left(\mathbf{I}_{d_V} \otimes \mathbf{W}_\mathrm{K}^\top\right) \mathbf{Z}_1^\top \left(\mathbf{I}_L \otimes \mathbf{W}_\mathrm{V} \mathbf{W}_\mathrm{V}^\top\right) \mathbf{Z}_1 \left(\mathbf{I}_{d_V} \otimes \mathbf{W}_\mathrm{K}\right),$$
>
> $$\mathbf{H}_\mathrm{o}\left(\mathbf{W}_\mathrm{V}, \mathbf{W}_\mathrm{Q}\right) = \frac{2}{Ld_V \sqrt{d_K}} \left(\mathbf{M}_1^\top \otimes \mathbf{W}_\mathrm{V}^\top\right) \mathbf{Z}_1 \left(\mathbf{I}_{d_V} \otimes \mathbf{W}_\mathrm{K}\right),$$
>
> $$\mathbf{H}_\mathrm{o}\left(\mathbf{W}_\mathrm{Q}, \mathbf{W}_\mathrm{K}\right) = \frac{2}{Ld_V d_K} \left(\mathbf{I}_{d_V} \otimes \mathbf{W}_\mathrm{K}^\top\right) \mathbf{Z}_1^\top \left(\mathbf{I}_L \otimes \mathbf{W}_\mathrm{V} \mathbf{W}_\mathrm{V}^\top\right) \mathbf{Z}_1 \left(\mathbf{W}_\mathrm{Q} \otimes \mathbf{I}_{d_V}\right) \mathbf{K}_{d_K, d_V},$$
>
> with the first attention moment matrix $\mathbf{M}_1 := \mathbf{A}\mathbf{X} \in \mathbb{R}^{L \times d_V}$ (see section 3.2) and where $\mathbf{Z}_1 := (\mathbf{I}_L \otimes \mathbf{X}^\top)(\partial \mathbf{A} / \partial \mathbf{T})(\mathbf{X} \otimes \mathbf{X}) \in \mathbb{R}^{Ld_V \times d_V^2}$ contains first derivatives of the softmax.

To arrive at these expressions we start from eq. (2), insert $\partial^2 \ell / \partial^2 \mathbf{F}$, and use the self-attention Jacobians derived by Noci et al. (2022). See appendix B for details. We note that the value diagonal

---

[1]Here we discuss the Hessian in a single data point setting to improve exposition, but our insights translate directly into a batch setting—one just needs to average the Hessian formulas over the data points.

block is structurally identical to an MLP's outer-product Hessian, while the query diagonal and the mixed query-key blocks display similar Kronecker structures, but with varying weight matrices.

> **Theorem 3.2. Functional Hessian $\mathbf{H}_f$.** For the setup of theorem 3.1, the functional Hessian w.r.t. the value weight matrix $\mathbf{H}_f(\mathbf{W}_V, \mathbf{W}_V)$ is zero and the remaining blocks are given by
>
> $$\mathbf{H}_f(\mathbf{W}_Q, \mathbf{W}_Q) = \frac{2}{L d_V d_K} \mathbf{R}_{d_V d_K} \left(\mathbf{I}_L \otimes \mathbf{W}_V^\top \otimes \mathbf{I}_{d_V} \otimes \mathbf{W}_K^\top\right) \mathbf{Z}_2 \left(\mathbf{I}_{d_V} \otimes \mathbf{W}_K\right),$$
>
> $$\mathbf{H}_f(\mathbf{W}_V, \mathbf{W}_Q) = \frac{2}{L d_V \sqrt{d_K}} \mathbf{R}_{d_V^2} \left(\mathbf{I}_L \otimes \mathbf{S}\right) \mathbf{Z}_1 \left(\mathbf{I}_{d_V} \otimes \mathbf{W}_K\right),$$
>
> $$\mathbf{H}_f(\mathbf{W}_Q, \mathbf{W}_K) = \frac{2}{L d_V d_K} \mathbf{R}_{d_V d_K} \left(\mathbf{I}_L \otimes \mathbf{W}_V^\top \otimes \mathbf{I}_{d_V} \otimes \mathbf{W}_K^\top\right) \mathbf{Z}_2 \left(\mathbf{W}_Q \otimes \mathbf{I}_{d_V}\right) \mathbf{K}_{d_K, d_V}$$
> $$+ \frac{2}{L d_V \sqrt{d_K}} \mathbf{R}_{d_V} \left(\mathbf{I}_L \otimes \mathbf{W}_V^\top \otimes \mathbf{I}_{d_V}\right) \left(\mathbf{Z}_1 \otimes \mathbf{I}_{d_V}\right) \mathbf{S} \otimes \mathbf{I}_{d_K},$$
>
> with the duplicated residual $\mathbf{R}_m := \mathrm{vec}_r\left(\mathbf{F}(\mathbf{X}) - \mathbf{Y}\right)^\top \otimes \mathbf{I}_m \in \mathbb{R}^{m \times mL d_V}$, a shuffling matrix $\mathbf{S} := \left(\mathbf{I}_{d_V} \otimes \mathbf{K}_{d_V, d_V}\right)\left(\mathrm{vec}_r \mathbf{I}_{d_V} \otimes \mathbf{I}_{d_V}\right) \in \mathbb{R}^{d_V^3 \times d_V}$ where $\mathbf{K}_{d_V, d_V}$ is a commutation matrix (see e.g. lemma B.2), $\mathbf{Z}_1$ defined as in theorem 3.1, and $\mathbf{Z}_2 := \left(\mathbf{I}_L \otimes \mathbf{X}^\top \otimes \mathbf{X}^\top \otimes \mathbf{X}^\top\right)\left(\partial^2 \mathbf{A}/\partial \mathbf{T}^2\right)\left(\mathbf{X} \otimes \mathbf{X}\right) \in \mathbb{R}^{L d_V^3 \times d_V^2}$ containing second-order softmax derivatives.

For theorem 3.2, we differentiate the self-attention Jacobian, and multiply it with the square loss gradient $\partial \ell / \partial \mathbf{F}$. See appendix B for details. Similarly to the MLP functional Hessian, the diagonal value block of the self-attention functional Hessian disappears. Again, structurally similar expressions appear in the query-key mixed term, and the query diagonal block, with the query-key mixed term being the most involved and consisting of two summands.

**Categorizing the constituent terms.** To simplify the analysis of theorems 3.1 and 3.2, we divide the terms into four categories. First, the least important are layout matrices that copy or shuffle entries via Kronecker products with an identity or permutation matrix. These serve to correctly arrange the entries in the Hessian from our matrix layout in eq. (4) and therefore we will often ignore them. Second, we have the matrices $\mathbf{Z}_1, \mathbf{Z}_2$, which contain first- and second-order derivatives of the softmax and have explicit dependencies on the data $\mathbf{X}$ (section 3.1). Third, we identify first-order moments $\mathbf{M}_1$ of the attention matrix, as well as higher moments in the attention derivatives in $\mathbf{Z}_{1,2}$ (section 3.2). Fourth, the Hessian depends on the weight matrices $\mathbf{W}_V, \mathbf{W}_Q, \mathbf{W}_K$ (section 3.3).

As one of our main interests is the scaling behavior of blocks w.r.t. different dependencies, we will simplify expressions using the Landau notation from Martens (2020, Section 14), where for two matrices $\mathbf{A}, \mathbf{B}$, denoting $f \in \mathcal{O}(\mathbf{AB})$ means all entries of $f$ are $\mathcal{O}(A_{i,j} B_{k,l})$ for any $i, j, k, l$. Due to their boundedness to $[0, 1]$, we can ignore the softmax expressions in the attention matrix.

## 3.1 DATA DEPENDENCE VARIES ACROSS HESSIAN BLOCKS

One characteristic of self-attention is that the data $\mathbf{X}$ enters multiple times: as keys, queries, and values. This leads to highly non-linear data dependencies in the Hessian. For brevity omitting the dependence of all the functional Hessian blocks on the residual $\delta_{\mathbf{XY}} := \mathrm{vec}_r\left(\mathbf{F}(\mathbf{X}) - \mathbf{Y}\right)^\top$, we find that for the expressions in theorems 3.1 and 3.2

$$\mathbf{H}_o \in \begin{array}{c} \\ \text{Q} \\ \text{K} \\ \text{V} \end{array} \overset{\begin{array}{ccc} \text{Q} & \text{K} & \text{V} \end{array}}{\begin{bmatrix} \mathcal{O}(\mathbf{X}^6) & \mathcal{O}(\mathbf{X}^6) & \mathcal{O}(\mathbf{X}^4) \\ \cdot & \mathcal{O}(\mathbf{X}^6) & \mathcal{O}(\mathbf{X}^4) \\ \cdot & \cdot & \mathcal{O}(\mathbf{X}^2) \end{bmatrix}}, \mathbf{H}_f \in \begin{array}{c} \\ \text{Q} \\ \text{K} \\ \text{V} \end{array} \overset{\begin{array}{ccc} \text{Q} & \text{K} & \text{V} \end{array}}{\begin{bmatrix} \mathcal{O}(\mathbf{X}^5) & \mathcal{O}(\mathbf{X}^5 + \mathbf{X}^3) & \mathcal{O}(\mathbf{X}^3) \\ \cdot & \mathcal{O}(\mathbf{X}^5) & \mathcal{O}(\mathbf{X}^3) \\ \cdot & \cdot & \mathcal{O}(1) \end{bmatrix}}. \quad (5)$$

**Data heterogeneity.** Throughout different Hessian blocks, we observe a varying dependence on the embedding matrix $\mathbf{X}$. The most data-dependent blocks are the key and query ones. For the outer-product Hessian $\mathbf{H}_o$, each key and query weight contributes a cubic dependence on $\mathbf{X}$ while value weight matrix $\mathbf{W}_V$ brings a linear dependence on $\mathbf{X}$. This observation confirms the finding of Noci et al. (2022) who show $\|\partial \mathbf{F}/\partial \mathbf{W}_V\|_F \in \mathcal{O}(\|\mathbf{X}\|_F)$, while

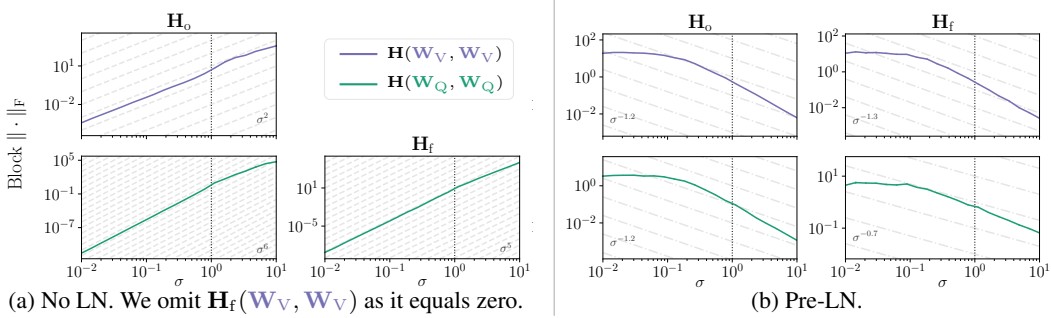

(a) No LN. We omit $\mathbf{H}_f(\mathbf{W}_V, \mathbf{W}_V)$ as it equals zero.   (b) Pre-LN.

Figure 3: (Plotted in log-log scale.) **Empirical verification with a CE loss confirms derived growth rates w.r.t. magnitude $\sigma$ of $\mathbf{X}$ from eq. (5).** We show the growth rates through the Frobenius norm $\| \cdot \|_\mathrm{F}$ of value and query diagonal blocks. The dashed lines correspond to the trend (a) predicted by theory as in eq. (5), (b) estimated from the Frobenius norm measurements on the log-log scale by the linear regression slope. For details on the experimental setting, see appendix F. $\sigma < 1$ (LHS of the vertical line) corresponds to practical values of $\sigma$.

$\|\partial \mathbf{F}/\partial \mathbf{W}_\mathrm{K}\|_\mathrm{F}, \|\partial \mathbf{F}/\partial \mathbf{W}_\mathrm{Q}\|_\mathrm{F} \in \mathcal{O}(\|\mathbf{X}\|_\mathrm{F}^3)$. The block-heterogeneous dependence is also visible in the functional Hessian $\mathbf{H}_\mathrm{f}$, which includes an additional dependence on $\mathbf{X}$ through the residual $\delta_{\mathbf{XY}}$. If we ignore $\delta_{\mathbf{XY}}$, then the outer-product Hessian dominates the Hessian for large $\mathbf{X}$.

**Empirical validation.** We test our theoretical derivations empirically using a single GPT-2 Transformer block (Radford et al., 2019) on a digit addition task adapted from Quirke & Barez (2024). The problem is set up as a next-token prediction task, *with cross-entropy (CE) loss* (see appendix F)—a setting closer to a realistic Transformer application. We initialize the embedding $\mathbf{X}$ from a distribution with varying standard deviation $\sigma$ and measure the Frobenius norm of the Hessian's diagonal blocks (fig. 3a). The growth rates from eq. (5) transfer to the dependence of the block Frobenius norm on $\sigma$. We omit the value functional Hessian block $\mathbf{H}_f(\mathbf{W}_V, \mathbf{W}_V)$ from the plot because it equals zero in both theory and our empirical evaluation. For the value block, the full Hessian follows the $\sigma^2$ trend from the outer-product Hessian from eq. (5), since the functional block is zero. The query block follows the functional Hessian's $\sigma^5$ trend, and not the $\sigma^6$ trend from the outer-product Hessian, which is likely suppressed by other non-data dependencies. *Overall, this confirms that although our theoretical results were derived for MSE loss and only a single self-attention layer, they extend faithfully to CE loss and a full Transformer block.*

## 3.2 ATTENTION MOMENTS AFFECT THE HESSIAN

**Attention scores induce distributions.** Each row of the self-attention matrix $\mathbf{A}(\mathbf{X})$ is a normalized distribution over the input tokens, and we already saw the first moment $\mathbf{M}_1$ (i.e. multiplying the sequence by the attention matrix) of those distributions emerge in theorems 3.1 and 3.2. To further simplify the $\mathbf{Z}_{1,2}$ matrices, we now introduce the natural generalization to higher-order centered attention moments. The $i$-th centered attention moment matrix $\mathbf{M}_i$ should be thought of as an $L \times d_V \times \ldots \times d_V$ tensor (where $d_V$ appears $i$ times) that stacks the $i$-th centered moments for each distribution, which is then flattened into a $Ld_V^{i-1} \times d_V$ matrix. In matrix notation, we have:

**Definition 3.1. Attention moment matrices.** Let $\mathbf{A} \in \mathbb{R}^{L \times L}$ be an attention matrix and $\mathbf{X} \in \mathbb{R}^{L \times d_V}$ be the token embeddings. We define the first attention moment matrix as

$$\mathbf{M}_1 := \mathbf{A}\mathbf{X} = \left[\mathbf{A}_{i,:}^\top \mathbf{X}\right]_{1 \le i \le L} \in \mathbb{R}^{L \times d_V}.$$

Similarly, the second and the third central moment matrices are

$$\mathbf{M}_2 := \left[\sum_{j=1}^{L} \mathbf{A}_{i,j} \left(\mathbf{X}_{j,:} - [\mathbf{M}_1]_{i,:}\right)\left(\mathbf{X}_{j,:} - [\mathbf{M}_1]_{i,:}\right)^\top\right]_{1 \le i \le L} \in \mathbb{R}^{Ld_V \times d_V}$$

$$\mathbf{M}_3 := \left[\sum_{j=1}^{L} \mathbf{A}_{i,j}(\mathbf{X}_{j,:} - [\mathbf{M}_1]_{i,:}) \otimes (\mathbf{X}_{j,:} - [\mathbf{M}_1]_{i,:})(\mathbf{X}_{j,:} - [\mathbf{M}_1]_{i,:})^\top\right]_{1 \le i \le L} \in \mathbb{R}^{Ld_V^2 \times d_V}.$$

**Transformer Hessian dependency on attention moments.**   Definition 3.1 allows us to further simplify the matrices $\mathbf{Z}_{1,2}$ from theorems 3.1 and 3.2. In addition to the dependency on the first attention moment matrix, we obtain dependencies on the second and third moment matrices.

> **Remark 3.1.** The data terms emerging in the self-attention Hessian can be expressed as functions of the self-attention central moment matrices (proof in appendix C),
>
> $$\mathbf{Z}_1 = \mathbf{X} * \mathbf{M}_2\,, \quad \text{and} \quad \mathbf{Z}_2 = \left(\mathbf{I}_L \otimes \mathbf{K}_{d_V, d_V} \otimes \mathbf{I}_{d_V}\right)\left(\mathbf{X} * \mathbf{X}^\top * \mathbf{M}_3\right)\,,$$
>
> where $*$ is the Khatri-Rao product (Khatri & Rao, 1968), see definition A.2.

Regarding attention moments, the value outer-product Hessian depends on the first moment, while the query-key outer-product Hessians are influenced by the second central moment. The query-key functional Hessian even exhibits a dependency on the third central moment.

**Influence of the attention moment matrices on the Hessian.**   If the attention scores were data-independent (similar to the uniform attention assumption in Noci et al. (2022)), which happens almost surely at initialization for large $d_K$, sequences with more similar words will result in a lower contribution of the query-key block. Considering an orthogonal setting with a fixed input sequence and studying the influence of varying attention scores, the query-key outer-product Hessian will dominate if the attention scores are highly dispersed across tokens. If instead, the attention matrix is sparse and some tokens attend to only one token (attention row is a one-hot vector), the contribution of the query-key part of the Hessian diminishes because the second and third central moment matrices of such one-hot self-attention distributions equal zero. To the best of our knowledge, our work is the first one to notice the relationship between the self-attention Hessian and self-attention moments. Knowing this dependence can contribute to a better understanding and interpretability of the Hessian—especially its evolution during training.

### 3.3   DEPENDENCE ON THE WEIGHT MATRICES

*The blocks in the self-attention Hessian vary significantly when it comes to the dependence on the weight matrices.* Among the terms on the diagonal of the outer-product Hessian, only the query and key components explicitly (and quadratically) depend on the weights beyond their influence through the attention scores. This dependence is on both the value weight matrix and one of the key and query matrices. The mixed terms involving queries and keys depend linearly on the keys and queries weights and quadratically on the value weights. Finally, the mixed terms involving values depend linearly on the selected weight matrices.

Also, the blocks of the functional Hessian depend on the weight matrix in a varied manner. The first observation is that the diagonal value functional Hessian block is always zero. Furthermore, the diagonal query block depends quadratically on the key weight matrix and linearly on the value weight matrix. The mixed value-query block depends only linearly on the key weight matrix.

> 💡 We find highly non-linear dependencies and block heterogeneity in terms of data, degrees of attention moments, and weights. We expect that identified sources of heterogeneity influence different algebraic properties, such as traces, norms, and eigenvalues of blocks—possibly explaining the varying block spectra, previously observed by Zhang et al. (2024).

## 4   IMPACT OF TRANSFORMER DESIGN COMPONENTS ON THE HESSIAN

The Transformer architecture has several key components that distinguish it from classical models like MLPs and CNNs. In the previous section, we analyzed how data dependence is one big distinguishing factor between the Hessian for Transformers and classical architectures. Now, we proceed to the other two significant departures within a Transformer, namely, the effects of having (i) a softmax nonlinearity within a layer block and (ii) a quadratic weight interaction through the query-key parameterization, by disabling them one at a time.

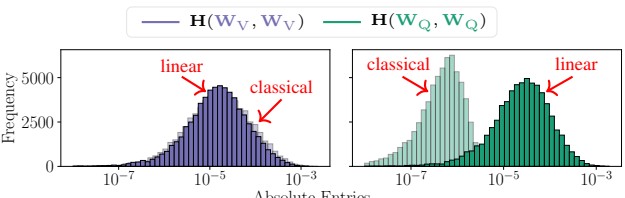

Figure 4: **Softmax results in heterogeneity in magnitudes of Hessian block entries.** Histogram of (absolute) entries corresponding to the **value** and **query** diagonal Hessian blocks of a single block Transformer. High and low saturation correspond to linear and classical self-attention respectively.

## 4.1 SOFTMAX ACTIVATION

Apart from the Transformer architecture, softmax is not a common choice for an internal activation function. It is usually used as the final layer activation in classification tasks to model a distribution. It is not an easy activation to work with, as it is prone to numerical instabilities and vanishing gradient problems Goodfellow et al. (2016); Noci et al. (2022); Wang et al. (2021).

**Hessian of linear self-attention.** To study the influence of softmax, we compare self-attention to *linear* self-attention, i.e. $a = \mathrm{id}$ and $\mathbf{T}(\mathbf{X}) = \mathbf{X}\mathbf{W}_{\mathrm{Q}}\mathbf{W}_{\mathrm{K}}^{\top}\mathbf{X}^{\top}/\sqrt{d_K}$ in eq. (1). In theorems 3.1 and 3.2, we now use that the first and second derivatives of $\mathbf{A}$ w.r.t. $\mathbf{T}$ are $\mathbf{I}_{d_V^2}$ and $\mathbf{0}$, respectively:

**Remark 4.1.** Assume a linear self-attention $\mathbf{F}$ with $a = \mathrm{id}$ and $\mathbf{T}(\mathbf{X}) = \mathbf{X}\mathbf{W}_{\mathrm{Q}}\mathbf{W}_{\mathrm{K}}^{\top}\mathbf{X}^{\top}/\sqrt{d_K}$ followed by square loss. Then the outer-product Hessian blocks are

$$\mathbf{H}_{\mathrm{o}}\left(\mathbf{W}_{\mathrm{V}}, \mathbf{W}_{\mathrm{V}}\right) = \frac{2L^2}{d_V d_K}\boldsymbol{\Sigma}_{\mathbf{XX}}\mathbf{W}_{\mathrm{K}}\mathbf{W}_{\mathrm{Q}}^{\top}\boldsymbol{\Sigma}_{\mathbf{XX}}\mathbf{W}_{\mathrm{Q}}\mathbf{W}_{\mathrm{K}}^{\top}\boldsymbol{\Sigma}_{\mathbf{XX}} \otimes \mathbf{I}_{d_V},$$

$$\mathbf{H}_{\mathrm{o}}\left(\mathbf{W}_{\mathrm{Q}}, \mathbf{W}_{\mathrm{Q}}\right) = \frac{2L^2}{d_V d_K}\boldsymbol{\Sigma}_{\mathbf{XX}} \otimes \mathbf{W}_{\mathrm{K}}^{\top}\boldsymbol{\Sigma}_{\mathbf{XX}}\mathbf{W}_{\mathrm{V}}\mathbf{W}_{\mathrm{V}}^{\top}\boldsymbol{\Sigma}_{\mathbf{XX}}\mathbf{W}_{\mathrm{K}},$$

$$\mathbf{H}_{\mathrm{o}}\left(\mathbf{W}_{\mathrm{V}}, \mathbf{W}_{\mathrm{Q}}\right) = \frac{2L^2}{d_V d_K}\left(\boldsymbol{\Sigma}_{\mathbf{XX}}\mathbf{W}_{\mathrm{K}}\mathbf{W}_{\mathrm{Q}}^{\top}\boldsymbol{\Sigma}_{\mathbf{XX}} \otimes \mathbf{W}_{\mathrm{V}}^{\top}\boldsymbol{\Sigma}_{\mathbf{XX}}\mathbf{W}_{\mathrm{K}}\right),$$

$$\mathbf{H}_{\mathrm{o}}\left(\mathbf{W}_{\mathrm{Q}}, \mathbf{W}_{\mathrm{K}}\right) = \frac{2L^2}{d_V d_K}\left(\boldsymbol{\Sigma}_{\mathbf{XX}}\mathbf{W}_{\mathrm{Q}} \otimes \mathbf{W}_{\mathrm{K}}^{\top}\boldsymbol{\Sigma}_{\mathbf{XX}}\mathbf{W}_{\mathrm{V}}\mathbf{W}_{\mathrm{V}}^{\top}\boldsymbol{\Sigma}_{\mathbf{XX}}\right)\mathbf{K}_{d_K, d_V},$$

where $\boldsymbol{\Sigma}_{\mathbf{XX}} := \mathbf{X}^{\top}\mathbf{X}/L$ is the empirical (uncentered) intra-sequence covariance.

Moreover, only the off-diagonal functional Hessian blocks are non-zero and equal to

$$\mathbf{H}_{\mathrm{f}}\left(\mathbf{W}_{\mathrm{V}}, \mathbf{W}_{\mathrm{Q}}\right) = \frac{2}{d_V \sqrt{d_K}}\left(\delta_{\mathbf{XY}}^{\top} \otimes \mathbf{I}_{d_V^2}\right)\left(\mathbf{I}_L \otimes \mathbf{S}\right)\left(\mathbf{X} \otimes \boldsymbol{\Sigma}_{\mathbf{XX}}\mathbf{W}_{\mathrm{K}}\right)$$

$$\mathbf{H}_{\mathrm{f}}\left(\mathbf{W}_{\mathrm{Q}}, \mathbf{W}_{\mathrm{K}}\right) = \frac{2}{d_V \sqrt{d_K}}\left(\delta_{\mathbf{XY}}^{\top}\left(\mathbf{X} \otimes \mathbf{W}_{\mathrm{V}}^{\top}\boldsymbol{\Sigma}_{\mathbf{XX}}\right) \otimes \mathbf{I}_{d_V}\right)\mathbf{S} \otimes \mathbf{I}_{d_K}$$

where $\delta_{\mathbf{XY}} := \mathrm{vec}_r\left(\mathbf{F}(\mathbf{X}) - \mathbf{Y}\right)$ and $\mathbf{S}$ is defined as in theorem 3.2.

Since linear self-attention is simple matrix multiplication, we observe a Kronecker product structure akin to that emerging from the outer product of matrix multiplication derivatives (eq. (15)). Moreover, we note that the functional Hessian has a block hollow structure, similar to the MLP functional Hessian as discussed in section 2.

**Softmax makes the Hessian block-heterogenous.** Removing the softmax from the self-attention definition makes the dependence of the Hessian blocks on the data more uniform across blocks. All blocks of the outer-product Hessian depend cubically on $\boldsymbol{\Sigma}_{\mathbf{XX}}$, while the non-zero functional Hessian blocks depend linearly on $\mathbf{X} \otimes \boldsymbol{\Sigma}_{\mathbf{XX}}$. Similarly, the number of weight matrices entering the expressions for the blocks for the Hessian without softmax is the same for the outer-product Hessian and across non-zero blocks of the functional Hessian.

Figure 4 empirically demonstrates that removing softmax from self-attention makes the magnitudes of the Hessian entries more homogeneous across blocks. For the classical attention (see also fig. 1a), we see that the distribution of Hessian block entries varies between query and value blocks—the entries of the query block are two orders of magnitude smaller than the entries of the value block.

We could explain this by $\mathbf{X}$ being initialized to values $< 1$ and the query Hessian block having a higher order dependence on $\mathbf{X}$ than the value Hessian block (see section 3.1). After removing the softmax (fig. 4), we see that both query and value histograms are largely similar as we would expect from remark 4.1. Figure 4 also informs us that softmax has a particularly strong influence on the magnitude of the query Hessian block and not on the value Hessian block.

**Softmax makes the Hessian diagonal blocks more indefinite.** When we remove the softmax from the self-attention definition, the functional Hessian has a block-hollow structure with zero blocks on the diagonal. This means that the Hessian blocks are fully defined by the outer-product Hessian, which ensures that they are positive-semidefinite. Park & Kim (2022) empirically noticed the more indefinite nature of Transformer Hessian compared to CNNs—our observation suggest that the reason for that could lie in the ubiquitous use of softmax in Transformers.

**Linear self-attention vs MLP.** The loss Hessian of an MLP and CNN has been previously theoretically studied in the setting without nonlinearities, so analyzing the linear self-attention Hessian lets us compare the two in a similar setting. The Transformer Hessian is much more data-dependent than that of MLPs and CNNs, and we summarize this in section 4.1 using the matrix big-$\mathcal{O}$ notation.

|  | MLP/CNN | Transformer |
|---|---|---|
| $\mathbf{H}_\mathrm{o}$ | $\mathcal{O}(\mathbf{\Sigma_{XX}})$ | $\mathcal{O}(\mathbf{\Sigma_{XX}^3})$ |
| $\mathbf{H}_\mathrm{f}$ | $\mathcal{O}(\mathbf{\Omega_{xy}})$ | $\mathcal{O}(\mathbf{\Omega_{xy}\Sigma_{XX}})$ |

Table 1: Dependence of the Hessian of a linear self-attention layer on the intra-sequence covariance $\mathbf{\Sigma_{XX}}$ and the input-residual covariance $\mathbf{\Omega_{xy}} := \frac{1}{L}\mathbf{X}^\top(\mathbf{F}(\mathbf{X}) - \mathbf{Y})$ in big-$\mathcal{O}$ notation (remark 4.1).

One crucial difference is the asymptotic growth rate w.r.t. depth. If we stack $D$ linear self-attention layers, the block-diagonal matrices of the Hessian will contain $3^D$ input intra-sequence covariances $\mathbf{\Sigma_{XX}}$. This result follows directly from the fact that a $D$-layer linear self-attention network, as well as its Jacobian w.r.t. any weight matrix, is a matrix chain involving $3^D$ input sequence matrices $\mathbf{X}$ due to the recurrence that the output of a layer is used three times by the consecutive layer. *This is in stark contrast to a deep linear MLP*, whose Jacobian is only linear in $\mathbf{x}$, and therefore a Hessian diagonal block contains only 2 data instances, independent of depth.

## 4.2 QUERY-KEY PARAMETERIZATION

In practice, self-attention parameterizes $\mathbf{T}$ using two matrices $\mathbf{W}_\mathrm{Q}$ and $\mathbf{W}_\mathrm{K}$. This ensures having clearly defined, interpretable token embeddings on the self-attention level and for small $d_K$ results in fewer parameters. From a function class perspective, we could equivalently replace the product $\mathbf{W}_\mathrm{Q}\mathbf{W}_\mathrm{K}^\top$ with a single matrix $\mathbf{W}_\mathrm{QK}$. However, this changes the landscape of loss.

**Query-key parameterization induces additional Hessian decomposition.** From theorem 3.2 we know that we can further decompose the query-key blocks of the functional Hessian of classical self-attention. We present this in detail in appendix D. This decomposition and specifically $\mathbf{T}$-functional Hessian from remark D.1 are by-products of parameterizing the self-attention matrix with two matrices $\mathbf{W}_\mathrm{Q}$ and $\mathbf{W}_\mathrm{K}$. To see that, let us consider a self-attention parameterized with a single matrix as another control model. In the definition of self-attention from eq. (1) we assume $a = \mathrm{softmax}$ applied row-wise and $\mathbf{T}(\mathbf{X}) = \mathbf{X}\mathbf{W}_\mathrm{QK}\mathbf{X}^\top$. In lemma 4.1 we present the Hessian of this model w.r.t. $\mathbf{W}_\mathrm{QK}$.

> **Lemma 4.1. Hessian of self-attention with single matrix attention parameterization.** Assume the self-attention definition from eq. (1), where $a = \mathrm{softmax}$ is applied row-wise, and $\mathbf{T}(\mathbf{X}) = \mathbf{X}\mathbf{W}_\mathrm{QK}\mathbf{X}^\top$ is followed by an MSE loss function. The loss Hessian w.r.t. $\mathbf{W}_\mathrm{QK}$ is
>
> $$\mathbf{H}(\mathbf{W}_\mathrm{QK}, \mathbf{W}_\mathrm{QK}) = \mathbf{Z}_1^\top \left(\mathbf{I}_L \otimes \mathbf{W}_\mathrm{V}\mathbf{W}_\mathrm{V}^\top\right) \mathbf{Z}_1 + \left(\delta_{\mathbf{XY}}^\top(\mathbf{I}_L \otimes \mathbf{W}_\mathrm{V}^\top) \otimes \mathbf{I}_{d_V^2}\right) \mathbf{Z}_2$$
>
> where $\mathbf{Z}_1, \mathbf{Z}_2, \delta_{\mathbf{XY}}$ are defined as in theorems 3.1 and 3.2 and remark D.1.

The expression for the Hessian $\mathbf{H}(\mathbf{W}_\mathrm{QK}, \mathbf{W}_\mathrm{QK})$ is part of the $\mathbf{T}$-outer-product Hessian of the classical self-attention in remark D.1, while there is no counterpart expression to the $\mathbf{T}$-functional Hessian. Double matrix parameterization of self-attention implies also that the query-key Hessian part explicitly depends on the query and key weight matrices $\mathbf{W}_\mathrm{Q}$ and $\mathbf{W}_\mathrm{K}$.

### 4.3 OTHER COMMON DESIGN CHOICES

**The influence of temperature on the Hessian terms varies across blocks.** Assume that in the definition of self-attention from eq. (1) we employ the classical self-attention but with an additional temperature scaling, namely $\mathbf{T}(\mathbf{X}) = \mathbf{X}\mathbf{W}_Q\mathbf{W}_K^\top\mathbf{X}^\top/(t\sqrt{d_K})$. This impacts the scaling factors in the $\mathbf{T}$-Gauss-Newton decomposition from remark D.1. The $\mathbf{T}$-outer-product Hessian $\mathbf{H}_o^{\mathbf{T}}$ will be scaled by $1/t^2$, and the $\mathbf{T}$-functional Hessian $\mathbf{H}_f^{\mathbf{T}}$ by $1/t$. To see that, note that $t$ is just an extra multiplier in front of $\sqrt{d_K}$ and the formulas for $\mathbf{H}_f^{\mathbf{T}}$ and $\mathbf{H}_o^{\mathbf{T}}$ depend on $1/\sqrt{d_K}$ linearly and quadratically, respectively. This implies that as $t$ grows, $\mathbf{H}_f^{\mathbf{T}}$ dominates the query-key part of the Hessian, and when $t \to 0$, $\mathbf{H}_o^{\mathbf{T}}$ becomes the more prominent part.

**Layer norm can reduce inter-block Hessian data heterogeneity.** Layer norm is used either in the original Post-LN version, where it is applied between residual blocks, or in the Pre-LN version (see Baevski & Auli (2018); Wang et al. (2019); Xiong et al. (2020)), where it is placed inside the residual connection and additionally after the final layer. The heavy dependence of the Hessian on the input matrix $\mathbf{X}$, growing super-exponentially with network depth (see section 4.1), means that, unless the data is standardized, we can observe exploding or vanishing phenomena, which will be more pronounced than in MLPs. This highlights the importance of layer norm in the Transformer architecture. Moreover, the proper placement of layer norm, as in the Pre-LN setting, addresses the data heterogeneity across Hessian blocks. In fig. 3b we plot the Frobenius norm of the Hessian blocks for the Transformer block including the Pre-LN. We verify that Pre-LN indeed addresses the block-heterogeneity w.r.t. data growth rates—the difference between the trend exponent in the two compared blocks is much smaller than in fig. 3.

## 5 CONCLUSION

**Summary.** In this work, we theoretically derived the entire structure of the self-attention Hessian and discussed, in detail, how it behaves. In particular, *we explicitly characterized that the self-attention Hessian blocks have heterogeneous dependence on the data, weight, and degree of the attention moment matrices.* Further, we identified that Transformer-specific design decisions, such as query-key parameterization and softmax, result in a more non-linear and heterogeneous Hessian structure. Thus, theoretical works should be mindful of not discarding these design choices for the sake of mathematical convenience, as they can significantly alter the Hessian structure across blocks.

**Discussion.** Our results contribute to the theoretical and practical discussion:

**(i)** *Understanding Transformer optimization.* The research community has been exploring why optimizing Transformers is more challenging than other architectures (Zhang et al., 2025; Xiong et al., 2020; Liu et al., 2019; Pan & Li, 2022). Ahn et al. (2024) proposed a linear self-attention model with a single weight matrix parameterization to examine the Transformer's loss landscape. Using this simplified model, they reproduced key optimization phenomena, such as the performance gap between Adam and SGD. Our work provides a Hessian-based perspective on this model, providing new hypotheses on which components may drive Transformer optimization challenges.

**(ii)** *Transformer-specific optimizers.* We believe that analyzing the block-diagonal structure can be beneficial for developing optimization algorithms specifically tailored to Transformer models. For example, there is evidence (Zhang et al., 2024; 2025) suggesting that the memory consumption of adaptive optimizers can be significantly reduced by assigning a single adaptive learning rate to parameters corresponding to a homogeneous block on the Hessian diagonal. Knowing the exact block structure in self-attention layers, we could now use it to adapt the learning rate.

**Limitations.** While our theoretical setting is limited to a simple, single-layer[2] model, we saw that it displays a rich algebraic structure that is contained within the Hessian. Nevertheless, extending it to multi-layer networks resembling fully-fledged Transformers, at least along some specific axes, would be interesting. Moreover, since usually the embedding matrix $\mathbf{X}$ is also learned, it would be worth studying its Hessian blocks.

*Notwithstanding, our work lays a crucial foundation for the theoretical analysis of the Transformer Hessian—to our knowledge, we are the first to study its exact expressions and their dependencies.*

---

[2]Theorems 3.1 and 3.2 directly generalize to the last layer of any deep self-attention network if we replace the data matrix $\mathbf{X}$ with the output from the penultimate layer. Moreover, thanks to the chain rule, the second derivatives of self-attention we derive are useful for the analysis of Transformers of any depth.

## REPRODUCIBILITY STATEMENT

We attach proofs of all theorems, lemmas, and more involved remarks presented in this manuscript in the appendix. Specifically, Appendix A discusses prerequisites, and Appendices B and C outline the proofs. Appendix F contains details on the experimental setup. The code used to generate numerical results is available at: https://github.com/dalab/transformer-hessian.

## ACKNOWLEDGMENTS

We would like to thank Thomas Hofmann, Antonio Orvieto, and the members of the DALab for their insightful comments. We also thank Michael Vollenweider for proofreading the manuscript. Felix Dangel would like to recognize that resources used in preparing this research were provided, in part, by the Province of Ontario, the Government of Canada through CIFAR, and companies sponsoring the Vector Institute. Sidak Pal Singh would like to acknowledge the financial support from Max Planck ETH Center for Learning Systems.

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

CONTENTS

## A  KNOWN DEFINITIONS AND PROPERTIES

This manuscript extensively uses the Kronecker product, as introduced in definition A.1. Some of our results rely on the generalization of the Kronecker product named Khatri–Rao product from definition A.2. This section lists matrix calculus and linear algebra properties used in this manuscript.

**Definition A.1. Kronecker product.** Let $\mathbf{A} \in \mathbb{R}^{m \times n}$ and $\mathbf{B} \in \mathbb{R}^{p \times q}$ matrix. The Kronecker product $\mathbf{A} \otimes \mathbf{B} \in \mathbb{R}^{mp \times nq}$ is a block matrix given by:

$$\mathbf{A} \otimes \mathbf{B} = \begin{pmatrix} a_{11}\mathbf{B} & a_{12}\mathbf{B} & \cdots & a_{1n}\mathbf{B} \\ a_{21}\mathbf{B} & a_{22}\mathbf{B} & \cdots & a_{2n}\mathbf{B} \\ \vdots & \vdots & \ddots & \vdots \\ a_{m1}\mathbf{B} & a_{m2}\mathbf{B} & \cdots & a_{mn}\mathbf{B} \end{pmatrix}.$$

**Definition A.2. Khatri–Rao product** Liu (1999)**.** Let $\mathbf{A}$ and $\mathbf{B}$ be block matrices with $n \times m$ blocks. The Khatri–Rao product $\mathbf{A} * \mathbf{B}$ is a block matrix with $n \times m$ blocks defined as:

$$\mathbf{A} * \mathbf{B} = (\mathbf{A}_{i,j} \otimes \mathbf{B}_{i,j})_{i,j} = \begin{pmatrix} \mathbf{A}_{1,1} \otimes \mathbf{B}_{1,1} & \cdots & \mathbf{A}_{1,m} \otimes \mathbf{B}_{1,m} \\ \vdots & \ddots & \vdots \\ \mathbf{A}_{n,1} \otimes \mathbf{B}_{n,1} & \cdots & \mathbf{A}_{n,m} \otimes \mathbf{B}_{n,m} \end{pmatrix},$$

where $\mathbf{A}_{l,k}$ and $\mathbf{B}_{l,k}$ are the blocks from $l^{\text{th}}$ block row and $k^{\text{th}}$ block column of $\mathbf{A}$ and $\mathbf{B}$, respectively.

**Basic Kronecker product properties.**  Here we list some useful properties of the Kronecker product. A discussion on the Kronecker product and proofs of the listed properties can be found in Magnus & Neudecker (2019). For all the listed properties, we assume that the matrices are of the appropriate dimensions to perform the operations.

$$\mathbf{A} \otimes (\mathbf{B} + \mathbf{C}) = \mathbf{A} \otimes \mathbf{B} + \mathbf{A} \otimes \mathbf{C} \ \text{ and } \ (\mathbf{B} + \mathbf{C}) \otimes \mathbf{A} = \mathbf{B} \otimes \mathbf{A} + \mathbf{C} \otimes \mathbf{A} \tag{6}$$

$$(k\mathbf{A}) \otimes \mathbf{B} = \mathbf{A} \otimes (k\mathbf{B}) = k(\mathbf{A} \otimes \mathbf{B}), \ k \in \mathbb{R} \tag{7}$$

$$(\mathbf{A} \otimes \mathbf{B}) \otimes \mathbf{C} = \mathbf{A} \otimes (\mathbf{B} \otimes \mathbf{C}) \tag{8}$$

$$(\mathbf{A} \otimes \mathbf{B})(\mathbf{C} \otimes \mathbf{D}) = (\mathbf{A}\mathbf{C}) \otimes (\mathbf{B}\mathbf{D}) \tag{9}$$

$$(\mathbf{A} \otimes \mathbf{B})^\top = \mathbf{A}^\top \otimes \mathbf{B}^\top \tag{10}$$

For $\mathbf{a}$, $\mathbf{b}$ being column vectors

$$\mathbf{a}^\top \otimes \mathbf{b} = \mathbf{b}\mathbf{a}^\top = \mathbf{b} \otimes \mathbf{a}^\top \tag{11}$$

**Vector-matrix Kronecker product.**  Assume that $\mathbf{A}$ and $\mathbf{B}$ can be multiplied. Based on the mixed product property and the fact that taking a Kronecker product of a matrix and a scalar is equivalent to only scaling matrix entries we can write

$$\mathbf{A}(\mathbf{v}^\top \otimes \mathbf{B}) = (1 \otimes \mathbf{A})(\mathbf{v}^\top \otimes \mathbf{B}) = (\mathbf{v}^\top \otimes \mathbf{A}\mathbf{B}). \tag{12}$$

**Vectorization & Kronecker product.**  Assume that $\mathbf{A} \in \mathbb{R}^{m \times n}$, then

$$\mathrm{vec}_r \, \mathbf{A} = (\mathbf{A} \otimes \mathbf{I}_n) \, \mathrm{vec}_r \, \mathbf{I}_n. \tag{13}$$

*Proof.* For a column-wise vectorization, theorem 2.2 from Magnus & Neudecker (2019) tells us that for a matrix $\mathbf{B} \in \mathbb{R}^{p \times q}$ it holds that $\mathrm{vec}_c \, \mathbf{B} = (\mathbf{I}_q \otimes \mathbf{B}) \, \mathrm{vec}_c \, \mathbf{I}_q$. Hence, we can write

$$\mathrm{vec}_r \, \mathbf{A} = \mathbf{K}_{m,n} \, \mathrm{vec}_c \, \mathbf{A} = \mathbf{K}_{m,n} \, (\mathbf{I}_n \otimes \mathbf{A}) \, \mathrm{vec}_c \, \mathbf{I}_n$$
$$= (\mathbf{A} \otimes \mathbf{I}_n) \, \mathbf{K}_{n,n} \, \mathrm{vec}_c \, \mathbf{I}_n = (\mathbf{A} \otimes \mathbf{I}_n) \, \mathrm{vec}_r \, \mathbf{I}_n.$$

$$\square$$

**Product of a block-diagonal matrix and a Kronecker product.**  Let for any $i \in \{1, \ldots, m\}$ $\mathbf{A} \in \mathbb{R}^{q \times r}$, $\mathbf{B} \in \mathbb{R}^{m \times n}$ and $\mathbf{C} \in \mathbb{R}^{r \times t}$. Then, thanks to the vector-matrix Kronecker product property (eq. (12)) the following expression holds

$$\begin{aligned}
\mathrm{blockdiag}\,(\mathbf{A}_i)\,(\mathbf{B} \otimes \mathbf{C}) &= \begin{bmatrix} \mathbf{A}_1 & \ldots & \mathbf{0} \\ \vdots & \ddots & \vdots \\ \mathbf{0} & \ldots & \mathbf{A}_m \end{bmatrix} \left( \begin{bmatrix} \mathbf{B}_{1,:}^\top \\ \vdots \\ \mathbf{B}_{m,:}^\top \end{bmatrix} \otimes \mathbf{C} \right) \\
&= \begin{bmatrix} \mathbf{A}_1 & \ldots & \mathbf{0} \\ \vdots & \ddots & \vdots \\ \mathbf{0} & \ldots & \mathbf{A}_m \end{bmatrix} \begin{bmatrix} \mathbf{B}_{1,:}^\top \otimes \mathbf{C} \\ \vdots \\ \mathbf{B}_{m,:}^\top \otimes \mathbf{C} \end{bmatrix} = \begin{bmatrix} \mathbf{A}_1 \left( \mathbf{B}_{1,:}^\top \otimes \mathbf{C} \right) \\ \vdots \\ \mathbf{A}_m \left( \mathbf{B}_{m,:}^\top \otimes \mathbf{C} \right) \end{bmatrix} \\
&= \begin{bmatrix} \mathbf{B}_{1,:}^\top \otimes \mathbf{A}_1 \mathbf{C} \\ \vdots \\ \mathbf{B}_{m,:}^\top \otimes \mathbf{A}_m \mathbf{C} \end{bmatrix}.
\end{aligned} \tag{14}$$

In the above equation $\mathbf{B}_{i,:}$ is the $i$-th row of $\mathbf{B}$ in column vector format.

**Derivative of a matrix product.**  If $\mathbf{F}(\mathbf{X}) = \mathbf{A}\mathbf{X}\mathbf{B}$, then

$$\nabla_{\mathbf{X}} \mathbf{F} = \mathbf{A} \otimes \mathbf{B}^\top. \tag{15}$$

The proof of the above formula can be found in Singh et al. (2021).

**Derivative of a Kronecker product.** Let's take $\mathbf{F}(\mathbf{X}) = \mathbf{X} \otimes \mathbf{Y}$, where $\mathbf{X} \in \mathbb{R}^{n \times q}$ and $\mathbf{Y} \in \mathbb{R}^{p \times r}$. Then

$$\nabla_{\mathbf{X}} \mathbf{F} = (\mathbf{I}_n \otimes \mathbf{K}_{p,q} \otimes \mathbf{I}_r) (\mathbf{I}_{nq} \otimes \mathrm{vec}_r \mathbf{Y}), \tag{16}$$

and analogously

$$\nabla_{\mathbf{Y}} \mathbf{F} = (\mathbf{I}_n \otimes \mathbf{K}_{p,q} \otimes \mathbf{I}_r) (\mathrm{vec}_r \mathbf{X} \otimes \mathbf{I}_{pr}). \tag{17}$$

*Proof.* The proof makes use of the same formula but for the column-wise definition of derivative that can be found in Magnus & Neudecker (2019) and states that the column-wise derivative of $\mathbf{F}$ w.r.t. $\mathbf{X}$ is given by $(\mathbf{I}_q \otimes \mathbf{K}_{r,n} \otimes \mathbf{I}_p) (\mathbf{I}_{nq} \otimes \mathrm{vec}_c \mathbf{Y})$. We will use the above formula with $\mathbf{X}^\top, \mathbf{Y}^\top$ instead of $\mathbf{X}, \mathbf{Y}$ respectively together with the first identification theorem (theorem 5.6 from Magnus & Neudecker (2019)).

$$\begin{aligned} d(\mathrm{vec}_r (\mathbf{X} \otimes \mathbf{Y})) &= d(\mathrm{vec}_c (\mathbf{X} \otimes \mathbf{Y})^\top) = d(\mathrm{vec}_c (\mathbf{X}^\top \otimes \mathbf{Y}^\top)) \\ &= (\mathbf{I}_n \otimes \mathbf{K}_{p,q} \otimes \mathbf{I}_r) (\mathbf{I}_{nq} \otimes \mathrm{vec}_c \mathbf{Y}^\top) d(\mathrm{vec}_c \mathbf{X}^\top) \\ &= (\mathbf{I}_n \otimes \mathbf{K}_{p,q} \otimes \mathbf{I}_r) (\mathbf{I}_{nq} \otimes \mathrm{vec}_r \mathbf{Y}) d(\mathrm{vec}_r \mathbf{X}), \end{aligned}$$

so again by the first identification theorem, we obtain eq. (16).

To prove eq. (17) we follow similar steps as above and use the formula for column-wise derivative of $\mathbf{F}$ w.r.t. $\mathbf{Y}$ from Magnus & Neudecker (2019), namely $(\mathbf{I}_q \otimes \mathbf{K}_{r,n} \otimes \mathbf{I}_p) (\mathrm{vec}_c \mathbf{X} \otimes \mathbf{I}_{pr})$. As for the previous formula, we again use the first identification theorem.

$$\begin{aligned} d(\mathrm{vec}_r (\mathbf{X} \otimes \mathbf{Y})) &= d(\mathrm{vec}_c (\mathbf{X} \otimes \mathbf{Y})^\top) = d(\mathrm{vec}_c (\mathbf{X}^\top \otimes \mathbf{Y}^\top)) \\ &= (\mathbf{I}_n \otimes \mathbf{K}_{p,q} \otimes \mathbf{I}_r) (\mathrm{vec}_c \mathbf{X}^\top \otimes \mathbf{I}_{pr}) d(\mathrm{vec}_c \mathbf{Y}^\top) \\ &= (\mathbf{I}_n \otimes \mathbf{K}_{p,q} \otimes \mathbf{I}_r) (\mathrm{vec}_r \mathbf{X} \otimes \mathbf{I}_{pr}) d(\mathrm{vec}_r \mathbf{Y}), \end{aligned}$$

$\square$

**Derivative of transposition.** Let $\mathbf{F}(\mathbf{X}) = \mathbf{X}^\top$ where $\mathbf{X} \in \mathbb{R}^{n \times q}$. Then

$$d \, \mathrm{vec}_r \mathbf{F}(\mathbf{X}) = d \, \mathrm{vec}_r \mathbf{X}^\top = \mathbf{K}_{q,n} \, d \, \mathrm{vec}_r \mathbf{X},$$

so from the first identification theorem (theorem 5.6 from Magnus & Neudecker (2019))

$$\nabla_{\mathbf{X}} \mathbf{F} = \mathbf{K}_{q,n}. \tag{18}$$

## B  Jacobian and Hessian Expressions

In this section, we derive the expression for the self-attention Hessian. Firstly, we focus on the classical self-attention which we discuss in section 3. Next, we move on to self-attention parameterized by a single query-key matrix, as discussed in section 4.2. For clarity, in this section, we will refer to the functional and outer-product Hessians as $\mathbf{F}$-functional and $\mathbf{F}$-outer-product Hessians.

### B.1  Classical Self-Attention

Knowing the Jacobian of self-attention is a crucial step in deriving the Hessian (the loss gradient's Jacobian). We start by recalling the formulas for self-attention Jacobians w.r.t. weight matrices derived in Noci et al. (2022).

**Lemma B.1. Jacobians of self-attention Noci et al. (2022).** The Jacobians of the classical self-attention layer (Vaswani et al., 2017) have the following form:

$$\frac{\partial \mathbf{F}}{\partial \mathbf{W_V}} = \mathrm{softmax}\left(\frac{\mathbf{XW_Q W_K^\top X^\top}}{\sqrt{d_K}}\right)\mathbf{X} \otimes \mathbf{I}_{d_V},$$

$$\frac{\partial \mathbf{F}}{\partial \mathbf{W_Q}} = \left(\mathbf{I}_L \otimes \mathbf{W_V^\top X^\top}\right)\frac{\partial \mathbf{A}}{\partial \mathbf{T}}\left(\frac{\mathbf{X} \otimes \mathbf{XW_K}}{\sqrt{d_K}}\right),$$

where the Jacobian of the row-wise softmax w.r.t. its inputs is as follows:

$$\frac{\partial \mathbf{A}}{\partial \mathbf{T}} = \mathrm{blockdiag}\left(\frac{\partial \mathbf{A}_{i,:}}{\partial \mathbf{T}_{i,:}}\right), \tag{19}$$

and where

$$\frac{\partial \mathbf{A}_{i,:}}{\partial \mathbf{T}_{i,:}} = \mathrm{diag}(\mathbf{A}_{i,:}) - \mathbf{A}_{i,:}\mathbf{A}_{i,:}^\top,$$

with $\mathbf{A}_{i,:}$ being the $i$-th row of $\mathbf{A}$ in column vector format.

With the Jacobians of self-attention w.r.t. the weight matrices established, we can introduce the Hessian. We start with the $\mathbf{F}$-outer-product Hessian and then proceed to the $\mathbf{F}$-functional Hessian.

**Theorem 3.1. Outer-product Hessian $\mathbf{H}_o$.** For a single self-attention layer, eq. (1), with classical self-attention that feeds into the square loss, the blocks of $\mathbf{H}_o$ are

$$\mathbf{H}_o\left(\mathbf{W_V}, \mathbf{W_V}\right) = \frac{2}{Ld_V}\mathbf{M}_1^\top \mathbf{M}_1 \otimes \mathbf{I}_{d_V},$$

$$\mathbf{H}_o\left(\mathbf{W_Q}, \mathbf{W_Q}\right) = \frac{2}{Ld_V d_K}\left(\mathbf{I}_{d_V} \otimes \mathbf{W_K^\top}\right)\mathbf{Z}_1^\top\left(\mathbf{I}_L \otimes \mathbf{W_V W_V^\top}\right)\mathbf{Z}_1\left(\mathbf{I}_{d_V} \otimes \mathbf{W_K}\right),$$

$$\mathbf{H}_o\left(\mathbf{W_V}, \mathbf{W_Q}\right) = \frac{2}{Ld_V\sqrt{d_K}}\left(\mathbf{M}_1^\top \otimes \mathbf{W_V^\top}\right)\mathbf{Z}_1\left(\mathbf{I}_{d_V} \otimes \mathbf{W_K}\right),$$

$$\mathbf{H}_o\left(\mathbf{W_Q}, \mathbf{W_K}\right) = \frac{2}{Ld_V d_K}\left(\mathbf{I}_{d_V} \otimes \mathbf{W_K^\top}\right)\mathbf{Z}_1^\top\left(\mathbf{I}_L \otimes \mathbf{W_V W_V^\top}\right)\mathbf{Z}_1\left(\mathbf{W_Q} \otimes \mathbf{I}_{d_V}\right)\mathbf{K}_{d_K,d_V},$$

with the first attention moment matrix $\mathbf{M}_1 := \mathbf{AX} \in \mathbb{R}^{L \times d_V}$ (see section 3.2) and where $\mathbf{Z}_1 := (\mathbf{I}_L \otimes \mathbf{X}^\top)(\partial \mathbf{A}/\partial \mathbf{T})(\mathbf{X} \otimes \mathbf{X}) \in \mathbb{R}^{Ld_V \times d_V^2}$ contains first derivatives of the softmax.

*Proof.* Hessian of the mean-square error loss w.r.t. network prediction is an identity matrix scaled by $\frac{2}{Ld_V}$. Hence, computing the $\mathbf{F}$-outer-product Hessian block is just taking the outer products of the appropriate self-attention Jacobians from lemma B.1 and scaling them by $\frac{2}{Ld_V}$. The expressions can be further simplified using the mixed-product property of the Kronecker product (eq. (9)).

Let us prove the exact formula for the mixed query-value block. The remaining formulas follow the same steps.

$$\mathbf{H}_o\left(\mathbf{W_V}, \mathbf{W_Q}\right) = \frac{2}{Ld_V}\frac{\partial \mathbf{F}}{\partial \mathbf{W_V}}^\top \frac{\partial \mathbf{F}}{\partial \mathbf{W_Q}}$$

$$= \frac{2}{Ldv}\left(\mathbf{X^\top A^\top} \otimes \mathbf{I}_{d_V}\right)\left(\mathbf{I}_L \otimes \mathbf{W_V^\top X^\top}\right)\frac{\partial \mathbf{A}}{\partial \mathbf{T}}\left(\frac{\mathbf{X} \otimes \mathbf{XW_K}}{\sqrt{d_K}}\right)$$

$$= \frac{2}{Ld_V\sqrt{d_K}}\left(\mathbf{X^\top A^\top} \otimes \mathbf{W_V^\top X^\top}\right)\frac{\partial \mathbf{A}}{\partial \mathbf{T}}\left(\frac{\mathbf{X} \otimes \mathbf{XW_K}}{\sqrt{d_K}}\right)$$

$$= \frac{2}{Ld_V\sqrt{d_K}}\left(\mathbf{T}_1^\top \otimes \mathbf{W_V^\top}\right)\left(\mathbf{I}_L \otimes \mathbf{X^\top}\right)\frac{\partial \mathbf{A}}{\partial \mathbf{T}}\left(\mathbf{X} \otimes \mathbf{X}\right)\left(\mathbf{I}_{d_V} \otimes \mathbf{W_K}\right)$$

$$= \frac{2}{Ld_V\sqrt{d_K}}\left(\mathbf{T}_1^\top \otimes \mathbf{W_V^\top}\right)\mathbf{Z}_1\left(\mathbf{I}_{d_V} \otimes \mathbf{W_K}\right) \in \mathbb{R}^{Ld_V \times d_V^2}.$$

$\square$

Before we derive the $\mathbf{F}$-functional Hessian expressions, we prove two helper lemmas which allow us to unify the structuring expressions that emerge in the $\mathbf{F}$-functional Hessian. The proof of lemma B.2 is a translation of tensor networks (Bridgeman & Chubb, 2017) to index notation. For a more transparent presentation, in this lemma, we denote the matrix sizes using capital letters.

First, we briefly discuss commutation matrices. For more information on computation matrices, see Magnus & Neudecker (2019). Let us consider a matrix $\mathbf{A} \in \mathbb{R}^{M \times N}$ and its row-flattened version $\text{vec}_r \mathbf{A} \in \mathbb{R}^{MN}$. The commutation matrix $\mathbf{K}_{N,M}$ provides a way to transpose the indices of $\mathbf{A}$ using the flattened convention.

Precisely, if we apply $\mathbf{K}_{N,M}$ to the flattened version of $\mathbf{A}$, we get the flattened version of $\mathbf{A}^\top$:

$$\mathbf{K}_{N,M} \text{vec}_r \mathbf{A} = \text{vec}_r (\mathbf{A}^\top).$$

The commutation matrix can also be used to permute two indices in a super-index, i.e. suppose $\mathbf{a} \in \mathbb{R}^{MN}$ is a vector indexed by a super-index $(m, n)$, where $1 \leq n \leq N$ and $1 \leq m \leq M$. Then $[\mathbf{K}_{N,M}\mathbf{a}]_{(n,m)} = [\mathbf{a}]_{(m,n)}$, i.e. the commutation matrix swaps the index order inside the super-index. To see this, note that for any $\mathbf{a} \in \mathbb{R}^{MN}$, we can find $\mathbf{A} \in \mathbb{R}^{M \times N}$ such that $\mathbf{a} = \text{vec}_r \mathbf{A}$, and

$$
\begin{aligned}
[\mathbf{K}_{N,M}\mathbf{a}]_{(n,m)} &= [\mathbf{K}_{N,M} \text{vec}_r \mathbf{A}]_{(n,m)} \\
&= [\text{vec}_r (\mathbf{A}^\top)]_{(n,m)} \\
&= [\text{vec}_r \mathbf{A}]_{(m,n)} \\
&= [\mathbf{a}]_{(m,n)}.
\end{aligned}
$$

Finally, let us note that in index notation, the entries of $\mathbf{K}_{N,M}$ are given by

$$[\mathbf{K}_{N,M}]_{(n,m),(m',n')} = \delta_{n,n'}\delta_{m,m'}.$$

**Lemma B.2.** Let $\mathbf{K}_{N,M} \in \mathbb{R}^{MN \times MN}$ denote the commutation matrix, then

$$(\mathbf{I}_M \otimes \mathbf{K}_{N,M})(\text{vec}_r \mathbf{I}_M \otimes \mathbf{I}_N) = (\mathbf{K}_{M,N} \otimes \mathbf{I}_M)(\mathbf{I}_N \otimes \text{vec}_r \mathbf{I}_M). \tag{20}$$

Specifically, for $M = N$ we obtain

$$(\mathbf{I}_M \otimes \mathbf{K}_{M,M})(\text{vec}_r \mathbf{I}_M \otimes \mathbf{I}_M) = (\mathbf{K}_{M,M} \otimes \mathbf{I}_M)(\mathbf{I}_M \otimes \text{vec}_r \mathbf{I}_M).$$

*Proof.* Consider the left-hand side of eq. (20) in index notation and simplify, which yields

$$
\begin{aligned}
&[(\mathbf{I}_M \otimes \mathbf{K}_{N,M})(\text{vec}_r \mathbf{I}_M \otimes \mathbf{I}_N)]_{(m_1,n_1,m_2),n_2} \\
&= \sum_{m_1',m_2',n_1'} [\mathbf{I}_M \otimes \mathbf{K}_{N,M}]_{(m_1,n_1,m_2),(m_1',m_2',n_1')} [\text{vec}_r \mathbf{I}_M \otimes \mathbf{I}_N]_{(m_1',m_2',n_1'),n_2} \\
&= \sum_{m_1',m_2',n_1'} \delta_{m_1,m_1'}\delta_{m_2,m_2'}\delta_{n_1,n_1'}\delta_{m_1',m_2'}\delta_{n_1',n_2} \\
&= \sum_{m_1',m_2',n_1'} \delta_{m_1',m_1,m_2',m_2}\delta_{n_1',n_1,n_2}.
\end{aligned}
$$

Simplifying the right-hand side of eq. (20) in index notation yields the same expression and thereby establishes the equality:

$$
\begin{aligned}
&[(\mathbf{K}_{M,N} \otimes \mathbf{I}_M)(\mathbf{I}_N \otimes \text{vec}_r \mathbf{I}_M)]_{(m_1,n_1,m_2),n_2} \\
&= \sum_{m_1',m_2',n_1'} [\mathbf{K}_{M,N} \otimes \mathbf{I}_M]_{(m_1,n_1,m_2),(n_1',m_1',m_2')} [\mathbf{I}_N \otimes \text{vec}_r \mathbf{I}_M]_{(n_1',m_1',m_2'),n_2} \\
&= \sum_{m_1',m_2',n_1'} \delta_{m_1,m_1'}\delta_{n_1,n_1'}\delta_{m_2,m_2'}\delta_{n_1',n_2}\delta_{m_1',m_2'} \\
&= \sum_{m_1',m_2',n_1'} \delta_{m_1',m_1,m_2',m_2}\delta_{n_1',n_1,n_2}.
\end{aligned}
$$

$\square$

**Lemma B.3.** For any matrix $\mathbf{A} \in \mathbb{R}^{m \times n}$ it holds that

$$\left(\mathbf{I}_m \otimes \mathbf{K}_{m,n}\right)\left(\text{vec}_r \, \mathbf{A} \otimes \mathbf{A}\right) = \left(\mathbf{A} \otimes \mathbf{A} \otimes \mathbf{I}_n\right)\left(\mathbf{I}_n \otimes \mathbf{K}_{n,n}\right)\left(\text{vec}_r \, \mathbf{I}_n \otimes \mathbf{I}_n\right).$$

*Proof.* The equality follows from the chain of transformations

$$
\begin{aligned}
\left(\mathbf{I}_m \otimes \mathbf{K}_{m,n}\right)\left(\text{vec}_r \, \mathbf{A} \otimes \mathbf{A}\right) &= \left(\mathbf{I}_m \otimes \mathbf{K}_{m,n}\right)\left(\left(\mathbf{A} \otimes \mathbf{I}_n\right) \text{vec}_r \, \mathbf{I}_n \otimes \mathbf{A}\right) \\
&= \left(\mathbf{I}_m \otimes \mathbf{K}_{m,n}\right)\left(\mathbf{A} \otimes \mathbf{I}_n \otimes \mathbf{A}\right)\left(\text{vec}_r \, \mathbf{I}_n \otimes \mathbf{I}_n\right) \\
&= \left(\mathbf{A} \otimes \mathbf{K}_{m,n}\left(\mathbf{I}_n \otimes \mathbf{A}\right)\right)\left(\text{vec}_r \, \mathbf{I}_n \otimes \mathbf{I}_n\right) \\
&= \left(\mathbf{A} \otimes \left(\mathbf{A} \otimes \mathbf{I}_n\right)\mathbf{K}_{n,n}\right)\left(\text{vec}_r \, \mathbf{I}_n \otimes \mathbf{I}_n\right) \\
&= \left(\mathbf{A} \otimes \mathbf{A} \otimes \mathbf{I}_n\right)\left(\mathbf{I}_n \otimes \mathbf{K}_{n,n}\right)\left(\text{vec}_r \, \mathbf{I}_n \otimes \mathbf{I}_n\right),
\end{aligned}
$$

were the first equality comes from eq. (13) and the remaining ones are consequences of the mixed product property (eq. (9)) and fundamental properties of the commutation matrix. $\square$

**Theorem 3.2. Functional Hessian $\mathbf{H}_f$.** For the setup of theorem 3.1, the functional Hessian w.r.t. the value weight matrix $\mathbf{H}_f\left(\mathbf{W}_V, \mathbf{W}_V\right)$ is zero and the remaining blocks are given by

$$\mathbf{H}_f\left(\mathbf{W}_Q, \mathbf{W}_Q\right) = \frac{2}{L d_V d_K} \mathbf{R}_{d_V d_K}\left(\mathbf{I}_L \otimes \mathbf{W}_V^\top \otimes \mathbf{I}_{d_V} \otimes \mathbf{W}_K^\top\right)\mathbf{Z}_2\left(\mathbf{I}_{d_V} \otimes \mathbf{W}_K\right),$$

$$\mathbf{H}_f\left(\mathbf{W}_V, \mathbf{W}_Q\right) = \frac{2}{L d_V \sqrt{d_K}} \mathbf{R}_{d_V^2}\left(\mathbf{I}_L \otimes \mathbf{S}\right)\mathbf{Z}_1\left(\mathbf{I}_{d_V} \otimes \mathbf{W}_K\right),$$

$$
\begin{aligned}
\mathbf{H}_f\left(\mathbf{W}_Q, \mathbf{W}_K\right) =\; & \frac{2}{L d_V d_K} \mathbf{R}_{d_V d_K}\left(\mathbf{I}_L \otimes \mathbf{W}_V^\top \otimes \mathbf{I}_{d_V} \otimes \mathbf{W}_K^\top\right)\mathbf{Z}_2\left(\mathbf{W}_Q \otimes \mathbf{I}_{d_V}\right)\mathbf{K}_{d_K, d_V} \\
& + \frac{2}{L d_V \sqrt{d_K}} \mathbf{R}_{d_V}\left(\mathbf{I}_L \otimes \mathbf{W}_V^\top \otimes \mathbf{I}_{d_V}\right)\left(\mathbf{Z}_1 \otimes \mathbf{I}_{d_V}\right)\mathbf{S} \otimes \mathbf{I}_{d_K},
\end{aligned}
$$

with the duplicated residual $\mathbf{R}_m := \text{vec}_r\left(\mathbf{F}(\mathbf{X}) - \mathbf{Y}\right)^\top \otimes \mathbf{I}_m \in \mathbb{R}^{m \times m L d_V}$, a shuffling matrix $\mathbf{S} := \left(\mathbf{I}_{d_V} \otimes \mathbf{K}_{d_V, d_V}\right)\left(\text{vec}_r \, \mathbf{I}_{d_V} \otimes \mathbf{I}_{d_V}\right) \in \mathbb{R}^{d_V^3 \times d_V}$ where $\mathbf{K}_{d_V, d_V}$ is a commutation matrix (see e.g. lemma B.2), $\mathbf{Z}_1$ defined as in theorem 3.1, and $\mathbf{Z}_2 := \left(\mathbf{I}_L \otimes \mathbf{X}^\top \otimes \mathbf{X}^\top \otimes \mathbf{X}^\top\right)\left(\partial^2 \mathbf{A} / \partial \mathbf{T}^2\right)\left(\mathbf{X} \otimes \mathbf{X}\right) \in \mathbb{R}^{L d_V^3 \times d_V^2}$ containing second-order softmax derivatives.

*Proof.* We derive the $\mathbf{F}$-functional Hessian block by block. For each block, we derive the second derivative matrix of $\mathbf{F}$, as in fig. 2. Each expression can be obtained by multiplying the second derivative matrix by the gradient of the loss according to the definition of $\mathbf{F}$-functional Hessian.

**F-functional Hessian w.r.t. $\mathbf{W}_V$ & $\mathbf{W}_Q$.** Let us find the formula for the second derivative of the Hessian w.r.t. value and key weight matrices $\mathbf{W}_V$ & $\mathbf{W}_Q$. To do that we take the Jacobian of $\partial \mathbf{F} / \partial \mathbf{W}_V$ w.r.t. the query weight matrix $\mathbf{W}_Q$. To simplify, consider the following assignments

$$\mathbf{F}_1 := \frac{\partial \mathbf{F}}{\partial \mathbf{W}_V} = \mathbf{F}_2 \otimes \mathbf{I}_{d_V},$$

$$\mathbf{F}_2 := \text{softmax}\left(\frac{\mathbf{X}\mathbf{W}_Q \mathbf{W}_K^\top \mathbf{X}^\top}{\sqrt{d_K}}\right)\mathbf{X}.$$

By the chain rule, we know that

$$\frac{\partial^2 \mathbf{F}}{\partial \mathbf{W}_V \partial \mathbf{W}_Q} = \frac{\partial \mathbf{F}_1}{\partial \mathbf{W}_Q} = \frac{\partial \mathbf{F}_1}{\partial \mathbf{F}_2}\frac{\partial \mathbf{F}_2}{\partial \mathbf{W}_Q}.$$

From eq. (16) we obtain

$$
\begin{aligned}
\frac{\partial \mathbf{F}_1}{\partial \mathbf{F}_2} &= \left(\mathbf{I}_L \otimes \mathbf{K}_{d_V, d_V} \otimes \mathbf{I}_{d_V}\right)\left(\mathbf{I}_{L d_V} \otimes \text{vec}_r(\mathbf{I}_{d_V})\right) \\
&= \mathbf{I}_L \otimes \underbrace{\left(\mathbf{K}_{d_V, d_V} \otimes \mathbf{I}_{d_V}\right)\left(\mathbf{I}_{d_V} \otimes \text{vec}_r(\mathbf{I}_{d_V})\right)}_{\mathbf{S}},
\end{aligned}
$$

where $\mathbf{S}$ is as in the theorem statement.

Moreover, by observing that $\mathbf{F}_2$ differs from $\mathbf{A}$ only by the presence of matrix $\mathbf{W}_V$, from lemma B.1 we get the derivative

$$\frac{\partial \mathbf{F}_2}{\partial \mathbf{W}_Q} = \left(\mathbf{I}_L \otimes \mathbf{X}^\top\right) \frac{\partial \mathbf{A}}{\partial \mathbf{T}} \left(\frac{\mathbf{X} \otimes \mathbf{X}\mathbf{W}_K}{\sqrt{d_K}}\right).$$

Plugging the formulas together yields the formula for the second derivative. The formula for the second derivative w.r.t. value and key weight matrices can be derived in the same way, with the only exception being that we take a derivative of $\mathbf{F}_2$ w.r.t. $\mathbf{W}_K$ instead of w.r.t. $\mathbf{W}_Q$.

**F-functional Hessian w.r.t. $\mathbf{W}_Q$.** To obtain the query-query block, we will start by differentiating the Jacobian to obtain the formula for the second derivative of self-attention w.r.t. query weight matrix $\mathbf{W}_Q$. Let

$$\mathbf{F}_1 := \frac{\partial \mathbf{F}}{\partial \mathbf{W}_Q} = \left(\mathbf{I}_L \otimes \mathbf{W}_V{}^\top \mathbf{X}^\top\right) \mathbf{G} \left(\frac{\mathbf{X} \otimes \mathbf{X}\mathbf{W}_K}{\sqrt{d_K}}\right),$$

$$\mathbf{G} := \frac{\partial \mathbf{A}}{\partial \mathbf{T}} = \text{blockdiag}\left(\frac{\partial \mathbf{A}_{i,:}}{\partial \mathbf{T}_{i,:}}\right),$$

$$\mathbf{T} := \frac{1}{\sqrt{d_K}} \mathbf{X}\mathbf{W}_Q (\mathbf{X}\mathbf{W}_K)^\top,$$

as in lemma B.1. By the chain rule, we obtain

$$\frac{\partial^2 \mathbf{F}}{\partial \mathbf{W}_Q \partial \mathbf{W}_Q} = \frac{\partial \mathbf{F}_1}{\partial \mathbf{W}_Q} = \frac{\partial \mathbf{F}_1}{\partial \mathbf{G}} \frac{\partial \mathbf{G}}{\partial \mathbf{T}} \frac{\partial \mathbf{T}}{\partial \mathbf{W}_Q}.$$

Thanks to eq. (15) and basic properties of the Kronecker product (eqs. (8) and (10)) we can write that

$$\frac{\partial \mathbf{F}_1}{\partial \mathbf{G}} = \left(\mathbf{I}_L \otimes \mathbf{W}_V{}^\top \mathbf{X}^\top\right) \otimes \left(\frac{\mathbf{X} \otimes \mathbf{X}\mathbf{W}_K}{\sqrt{d_K}}\right)^\top$$

$$= \left(\frac{\mathbf{I}_L \otimes \mathbf{W}_V{}^\top \mathbf{X}^\top \otimes \mathbf{X}^\top \otimes \mathbf{W}_K{}^\top \mathbf{X}^\top}{\sqrt{d_K}}\right),$$

$$\frac{\partial \mathbf{T}}{\partial \mathbf{W}_Q} = \left(\frac{\mathbf{X} \otimes \mathbf{X}\mathbf{W}_K}{\sqrt{d_K}}\right).$$

Additionally, we note that $\partial \mathbf{G}/\partial \mathbf{T}$ is the second derivative of row-wise softmax

$$\frac{\partial \mathbf{G}}{\partial \mathbf{T}} = \frac{\partial^2 \mathbf{A}}{\partial \mathbf{T} \partial \mathbf{T}},$$

which we describe in detail in appendix C.1. Plugging in the above formulas to the chain rule yields the formula for the second derivative we were looking for.

**F-functional Hessian w.r.t. $\mathbf{W}_Q$ & $\mathbf{W}_K$.** We proceed by differentiating the Jacobian of $\mathbf{F}$ w.r.t $\mathbf{W}_Q$. Let us again start with defining some helper terms

$$\mathbf{F}_1 := \left(\mathbf{I}_L \otimes \mathbf{W}_V{}^\top \mathbf{X}^\top\right),$$

$$\mathbf{G} := \frac{\partial \mathbf{A}}{\partial \mathbf{T}} = \text{blockdiag}\left(\frac{\partial \mathbf{A}_{i,:}}{\partial \mathbf{T}_{i,:}}\right),$$

$$\mathbf{F}_2 := \left(\frac{\mathbf{X} \otimes \mathbf{X}\mathbf{W}_K}{\sqrt{d_K}}\right).$$

Note that only $\mathbf{G}$ and $\mathbf{F}_2$ depend on $\mathbf{W}_K$ so by applying the chain rule to the product of functions and using a derivative of a matrix product formula (eq. (15)) we get

$$\frac{\partial^2 \mathbf{F}}{\partial \mathbf{W}_Q \partial \mathbf{W}_K} = \frac{\partial \mathbf{F}_1 \mathbf{G} \mathbf{F}_2}{\partial \mathbf{W}_K} = \left(\mathbf{F}_1 \otimes \mathbf{F}_2^\top\right) \frac{\partial \mathbf{G}}{\partial \mathbf{W}_K} + \left(\mathbf{F}_1 \mathbf{G} \otimes \mathbf{I}_{d_V d_K}\right) \frac{\partial \mathbf{F}_2}{\partial \mathbf{W}_K}. \tag{21}$$

Now, thanks to the chain rule, a derivative of a matrix product (eq. (15)) and the derivative of a transposition (eq. (18))

$$\frac{\partial \mathbf{G}}{\partial \mathbf{W}_{\mathrm{K}}} = \frac{\partial^2 \mathbf{A}}{\partial \mathbf{T} \partial \mathbf{T}} \left( \frac{\mathbf{X}\mathbf{W}_{\mathrm{Q}} \otimes \mathbf{X}}{\sqrt{d_K}} \right) \mathbf{K}_{d_K, d_V}.$$

Additionally, due to the chain rule and the formula for the Kronecker product derivative (eq. (17))

$$
\begin{aligned}
\frac{\partial \mathbf{F}_2}{\partial \mathbf{W}_{\mathrm{K}}} &= (\mathbf{I}_L \otimes \mathbf{K}_{L,d_V} \otimes \mathbf{I}_{d_K}) \left( \mathrm{vec}_r \, \mathbf{X} \otimes \mathbf{I}_{Ld_K} \right) \left( \frac{\mathbf{X} \otimes \mathbf{I}_{d_K}}{\sqrt{d_K}} \right) \\
&= \frac{1}{\sqrt{d_K}} (\mathbf{I}_L \otimes \mathbf{K}_{L,d_V} \otimes \mathbf{I}_{d_K}) \left( \mathrm{vec}_r \, \mathbf{X} \otimes \mathbf{X} \otimes \mathbf{I}_{d_K} \right) \\
&= \frac{1}{\sqrt{d_K}} (\mathbf{I}_L \otimes \mathbf{K}_{L,d_V}) \left( \mathrm{vec}_r \, \mathbf{X} \otimes \mathbf{X} \right) \otimes \mathbf{I}_{d_K} \\
&= \frac{1}{\sqrt{d_K}} (\mathbf{X} \otimes \mathbf{X} \otimes \mathbf{I}_{d_V}) (\mathbf{I}_{d_V} \otimes \mathbf{K}_{d_V, d_V}) (\mathrm{vec}_r \, \mathbf{I}_{d_V} \otimes \mathbf{I}_{d_V}) \otimes \mathbf{I}_{d_K} \\
&= \frac{1}{\sqrt{d_K}} (\mathbf{X} \otimes \mathbf{X} \otimes \mathbf{I}_{d_V}) \underbrace{(\mathbf{K}_{d_V, d_V} \otimes \mathbf{I}_{d_V}) (\mathbf{I}_{d_V} \otimes \mathrm{vec}_r(\mathbf{I}_{d_V}))}_{\mathbf{S}} \otimes \mathbf{I}_{d_K}
\end{aligned}
$$

where the first two transformations follow from the mixed product property (eqs. (9) and (12)) and the remaining two from lemmas B.2 and B.3. Plugging in all the terms we obtain the Hessian block. □

### B.1.1   THE JACOBIAN

We highlight that our observations regarding the Hessian made throughout the paper align well with those concerning the loss gradient w.r.t. self-attention parameters and the self-attention Jacobian. Liu et al. (2020) observed that the gradient w.r.t. the self-attention parameters is unbalanced, with the gradients w.r.t. the key and query parameters being of a smaller norm than those w.r.t. the value weight matrix. From the self-attention Jacobian formulas derived by Noci et al. (2022) (see lemma B.1), we observe a heterogeneous data dependence in the self-attention Jacobians. These formulas suggest that the Frobenius norm of the Jacobian w.r.t. the query parameter should scale cubically with the magnitude of $X$, whereas the Jacobian w.r.t. the value parameter only linearly. Moreover, the value Jacobian depends on the first moment of attention, while the query and key Jacobian depend on the second moment matrix. Finally, the Jacobians exhibit heterogeneous dependence on the scale of the weight matrices, as previously noted by Noci et al. (2022) in theorem 3.1.

### B.2   SELF-ATTENTION WITH SINGLE-MATRIX PARAMETERIZATION

**Lemma 4.1. Hessian of self-attention with single matrix attention parameterization.**  Assume the self-attention definition from eq. (1), where $a = \mathrm{softmax}$ is applied row-wise, and $\mathbf{T}(\mathbf{X}) = \mathbf{X}\mathbf{W}_{\mathrm{QK}}\mathbf{X}^{\top}$ is followed by an MSE loss function. The loss Hessian w.r.t. $\mathbf{W}_{\mathrm{QK}}$ is

$$\mathbf{H}(\mathbf{W}_{\mathrm{QK}}, \mathbf{W}_{\mathrm{QK}}) = \mathbf{Z}_1^{\top} \left( \mathbf{I}_L \otimes \mathbf{W}_{\mathrm{V}} \mathbf{W}_{\mathrm{V}}^{\top} \right) \mathbf{Z}_1 + \left( \delta_{\mathbf{XY}}^{\top} (\mathbf{I}_L \otimes \mathbf{W}_{\mathrm{V}}^{\top}) \otimes \mathbf{I}_{d_V^2} \right) \mathbf{Z}_2$$

where $\mathbf{Z}_1, \mathbf{Z}_2, \delta_{\mathbf{XY}}$ are defined as in theorems 3.1 and 3.2 and remark D.1.

*Proof.* The Jacobian of self-attention for $\mathbf{W}_{\mathrm{QK}}$ matrix is given by

$$\frac{\partial \mathbf{F}}{\partial \mathbf{W}_{\mathrm{QK}}} = \left( \mathbf{I}_L \otimes \mathbf{W}_{\mathrm{V}}^{\top} \mathbf{X}^{\top} \right) \frac{\partial \mathbf{A}}{\partial \mathbf{T}} \left( \mathbf{X} \otimes \mathbf{X} \right), \tag{22}$$

where $\partial \mathbf{A} / \partial \mathbf{T}$ is defined as in lemma B.1. The proof follows from applying the chain rule and the derivative of matrix multiplication as in eq. (15).

Now, to obtain the $\mathbf{F}$-functional Hessian formula, let

$$\mathbf{F}_1 := \frac{\partial \mathbf{F}}{\partial \mathbf{W}_{\mathrm{QK}}} = \left(\mathbf{I}_L \otimes \mathbf{W}_{\mathrm{V}}^{\top}\mathbf{X}^{\top}\right)\mathbf{G}\left(\mathbf{X} \otimes \mathbf{X}\right),$$

$$\mathbf{G} := \frac{\partial \mathbf{A}}{\partial \mathbf{T}} = \mathrm{blockdiag}\left(\frac{\partial \mathbf{A}_{i,:}}{\partial \mathbf{T}_{i,:}}\right),$$

$$\mathbf{T} := \mathbf{X}\mathbf{W}_{\mathrm{QK}}\mathbf{X}^{\top}.$$

From the chain rule, we obtain

$$\frac{\partial^2 \mathbf{F}}{\partial \mathbf{W}_{\mathrm{QK}}\partial \mathbf{W}_{\mathrm{QK}}} = \frac{\partial \mathbf{F}_1}{\partial \mathbf{G}}\frac{\partial \mathbf{G}}{\partial \mathbf{T}}\frac{\partial \mathbf{T}}{\partial \mathbf{W}_{\mathrm{QK}}}$$

$$= \left(\mathbf{I}_L \otimes \mathbf{W}_{\mathrm{V}}^{\top}\mathbf{X}^{\top} \otimes \mathbf{X}^{\top} \otimes \mathbf{X}^{\top}\right)\frac{\partial^2 \mathbf{A}}{\partial \mathbf{T}\partial \mathbf{T}}\left(\mathbf{X} \otimes \mathbf{X}\right),$$

where to get the second line we use the formula for the derivative of a matrix product (eq. (15)) twice. The formula from the lemma statement follows directly from plugging in the above components to eq. (2). □

## C  SELF-ATTENTION MOMENT MATRICES

In this section, we prove remark 3.1. To notice the dependence of the Hessian on the second moment matrix, we need to derive and simplify the second derivative of the row-wise softmax—we focus on that in appendix C.1. In appendix C.2 we move on to interpreting matrices $\mathbf{Z}_1$ and $\mathbf{Z}_2$ through the lens of the self-attention moment matrices.

### C.1  THE SECOND DERIVATIVE OF THE ROW-WISE SOFTMAX

To gain some intuition on the structure of the second derivative of the row-wise softmax, we begin with the simplest possible case, assuming a sequence length of $L = 2$. Later on, we will generalize the expressions to sequences of any length.

**Remark C.1.** Assume that the sequences are of length $L = 2$, meaning that the attention matrix is of the form

$$\mathbf{A} = \begin{bmatrix} a_{1,1} & a_{1,2} \\ a_{2,1} & a_{2,2} \end{bmatrix}.$$

By lemma B.1 the Jacobian of the row-wise softmax is given by matrix $\dfrac{\partial \mathbf{A}}{\partial \mathbf{T}}$ of the form

$$\begin{bmatrix} a_{1,1} - a_{1,1}^2 & -a_{1,1}a_{1,2} & 0 & 0 \\ -a_{1,1}a_{1,2} & a_{1,2} - a_{1,2}^2 & 0 & 0 \\ 0 & 0 & a_{2,1} - a_{2,1}^2 & -a_{2,1}a_{2,2} \\ 0 & 0 & -a_{2,1}a_{2,1} & a_{2,2} - a_{2,2}^2 \end{bmatrix}.$$

By vectorizing the above matrix row-wise, computing derivatives w.r.t. corresponding softmax inputs, and defining $b_{i,j} := 1 - 2a_{i,j}$ we obtain that the second derivative matrix is of the form

$$\frac{\partial^2 \mathbf{F}}{\partial \mathbf{T} \partial \mathbf{T}} = \left[ \begin{array}{cc:cc} (a_{1,1} - a_{1,1}^2)b_{1,1} & -a_{1,1}a_{1,2}b_{1,1} & 0 & 0 \\ -a_{1,1}a_{1,2}b_{1,1} & -a_{1,1}a_{1,2}b_{1,2} & 0 & 0 \\ 0 & 0 & 0 & 0 \\ 0 & 0 & 0 & 0 \\ -a_{1,1}a_{1,2}b_{1,1} & -a_{1,1}a_{1,2}b_{1,2} & 0 & 0 \\ -a_{1,1}a_{1,2}b_{1,2} & (a_{1,2} - a_{1,2}^2)b_{1,2} & 0 & 0 \\ 0 & 0 & 0 & 0 \\ 0 & 0 & 0 & 0 \\ \hdashline 0 & 0 & 0 & 0 \\ 0 & 0 & 0 & 0 \\ 0 & 0 & (a_{2,1} - a_{2,1}^2)b_{2,1} & -a_{2,1}a_{2,2}b_{2,1} \\ 0 & 0 & -a_{2,1}a_{2,2}b_{2,1} & -a_{2,1}a_{2,2}b_{2,2} \\ 0 & 0 & 0 & 0 \\ 0 & 0 & 0 & 0 \\ 0 & 0 & -a_{2,1}a_{2,2}b_{2,1} & -a_{2,1}a_{2,2}b_{2,2} \\ 0 & 0 & -a_{2,1}a_{2,2}b_{2,2} & (a_{2,2} - a_{2,2}^2)b_{2,2} \end{array} \right].$$

To obtain each entry of the matrix, we simply apply the chain rule. This means that each entry is a sum of products, where the first factor is the derivative of the entries of $\partial \mathbf{A}/\partial \mathbf{T}$ w.r.t. $a_{i,j}$, and the second factor is the derivative of $a_{i,j}$ w.r.t. the entries of $\mathbf{T}$. These derivatives can be found directly in the $\partial \mathbf{A}/\partial \mathbf{T}$ matrix. The blocks of zeros occur because the softmax function is applied independently to each row, so the second derivative for mixed-row entries is always zero.

Lemma C.1 generalizes the above observations to any sequence length $L$.

**Lemma C.1. Second derivative of row-wise softmax.** Assume self-attention matrix $\mathbf{A} = [a]_{i,j \leq L}$. Then the second derivative of the row-wise softmax has a block-diagonal structure, namely

$$\frac{\partial^2 \mathbf{F}}{\partial \mathbf{T} \partial \mathbf{T}} = \text{blockdiag}\, \mathbf{D}_i \in \mathbb{R}^{L^4 \times L^2},$$

where $\mathbf{D}_i \in \mathbb{R}^{L^3 \times L}$ is of the form

$$\mathbf{D}_i = \begin{bmatrix} \mathbf{D}_{i,1} \\ \vdots \\ \mathbf{D}_{i,L,} \end{bmatrix}, \text{ with } \mathbf{D}_{i,j} = \mathbf{e}_i \otimes \frac{\partial^2 \mathbf{A}_{i,j}}{\partial \mathbf{T}_{i,:} \partial \mathbf{T}_{i,:}},$$

and $\mathbf{e}_i$ being unit vectors in the standard basis. Moreover, the single-element second derivatives can be expressed as

$$\frac{\partial^2 \mathbf{A}_{i,j}}{\partial \mathbf{T}_{i,:} \partial \mathbf{T}_{i,:}} = \mathbf{A}_{i,j} \left( 2\mathbf{A}_{i,:} \mathbf{A}_{i,:}^\top + \mathbf{E}_{j,j}^{L,L} - \text{diag}(\mathbf{A}_{i,:}) - \mathbf{e}_j \mathbf{A}_{i,:}^\top - \mathbf{A}_{i,:} \mathbf{e}_j^\top \right) \in \mathbb{R}^{L \times L},$$

where $\mathbf{E}_{j,j}^{m,n} = \mathbf{e}_j \mathbf{e}_j^\top \in \mathbb{R}^{m \times n}$ is a matrix filled with zeros except for entry in $j^{th}$ row and $j^{th}$ column which is 1.

Additionally, if we structure the single entry second derivatives into a block column matrix $\frac{\partial^2 \mathbf{A}_{i,:}}{\partial \mathbf{T}_{i,:} \partial \mathbf{T}_{i,:}} = \left[ \frac{\partial^2 \mathbf{A}_{i,1}}{\partial \mathbf{T}_{i,:} \partial \mathbf{T}_{i,:}}, \ldots, \frac{\partial^2 \mathbf{A}_{i,L}}{\partial \mathbf{T}_{i,:} \partial \mathbf{T}_{i,:}} \right]^\top$ we can rewrite $\mathbf{D}_i$ concisely as

$$\mathbf{D}_i = \left( (\mathbb{1}_{L,1} \otimes \mathbf{e}_i) * \frac{\partial^2 \mathbf{A}_{i,:}}{\partial \mathbf{T}_{i,:} \partial \mathbf{T}_{i,:}} \right), \tag{23}$$

where $*$ represents Khatri-Rao product Liu (1999) (see definition A.2) with blocks defined by the standard basis vectors in the LHS matrix and by the second derivatives of a single row entry in the RHS matrix.

*Proof.* Let us start with discussing the general structure of the second derivative matrix. Later we will focus on specific matrix entries.

**Block-column structure.** Recall that we are computing the second derivative of a matrix-valued function $\mathbf{A}$ that takes a matrix $\mathbf{T}$ as an argument. In the layout and vectorization scheme assumed in this manuscript, the Hessians of every entry of $\mathbf{A}$ in the row-wise order are placed consecutively into a block-column matrix (see fig. 2).

**Block-diagonal structure.** Let us consider a single block from the block-column structure discussed in the previous paragraph. Since the softmax acts on every row of $\mathbf{T}$ separately, and the Hessian of $\mathbf{A}_{i,j}$ is computed w.r.t. all $L^2$ entries of $\mathbf{T}$, the Hessian will have potentially non-zero values only in one sub-block on the diagonal whose rows end columns have indices in the range from $(i-1)L + 1$ to $iL$ inclusive. This translates into the derivative of the row-wise softmax $\partial^2 \mathbf{F}/\partial \mathbf{T} \partial \mathbf{T}$ also having the block-diagonal structure. Specifically, let us enumerate the $L^2 \times L^2$ single-entry Hessians placed into this block-column matrix with an index $J$. For any integer $i$ such that $1 \leq i \leq L$ when $(i-1)L + 1 \leq J \leq iL$ we have possible non-zero values only in columns $iL$ to $(i+1)L - 1$ inclusive. Now let us group these Hessians into $L$ larger blocks corresponding to a single row of $\mathbf{A}$—the whole such block corresponding to $\mathbf{A}_{i,:}$ has possible non-zero entries only in columns from $(i-1)L + 1$ to $iL$ inclusive. Hence the structure of the second derivative of self-attention is block-diagonal if we consider only the possibly non-zero sub-blocks. We refer to the sub-block corresponding to row $\mathbf{A}_{i,:}$ w.r.t $\mathbf{T}_{i,:}$ by $\mathbf{D}_i$.

**Block-column structure of $\mathbf{D}_i$.** Finally, $\mathbf{D}_i$ consists of blocks, cut out of the element-wise Hessians we have just discussed, which explains the Khatri-Rao product in eq. (23).

**Non-zero elements.** The non-zero entries of $\mathbf{D}_i$ correspond to

$$\frac{\partial^2 \mathbf{A}_{i,j}}{\partial \mathbf{T}_{i,:} \partial \mathbf{T}_{i,:}}.$$

We now find its exact formula.

Recall from lemma B.1 that

$$\frac{\partial \mathbf{A}_{i,:}}{\partial \mathbf{T}_{i,:}} = \operatorname{diag}(\mathbf{A}_{i,:}) - \mathbf{A}_{i,:}\mathbf{A}_{i,:}^\top,$$

which means that

$$\frac{\partial \mathbf{A}_{i,j}}{\partial \mathbf{T}_{i,:}} = (\mathbf{e}_j - \mathbf{A}_{i,:})^\top \mathbf{A}_{i,j} \in \mathbb{R}^{1 \times L}.$$

To get the second derivative it is enough to differentiate the transpose of the above function using the Leibniz product rule, namely

$$\frac{\partial^2 \mathbf{A}_{i,j}}{\partial \mathbf{T}_{i,:}\partial \mathbf{T}_{i,:}} = \frac{\partial(\mathbf{e}_j - \mathbf{A}_{i,:})}{\partial \mathbf{T}_{i,:}}\mathbf{A}_{i,j} + (\mathbf{e}_j - \mathbf{A}_{i,:})\frac{\partial \mathbf{A}_{i,j}}{\partial \mathbf{T}_{i,:}}$$

$$= -\frac{\partial \mathbf{A}_{i,:}}{\partial \mathbf{T}_{i,:}}\mathbf{A}_{i,j} + (\mathbf{e}_j - \mathbf{A}_{i,:})\frac{\partial \mathbf{A}_{i,j}}{\partial \mathbf{T}_{i,:}}.$$

Since we already know both derivatives present in this expression from lemma B.1 it is enough to substitute them to obtain

$$\frac{\partial^2 \mathbf{A}_{i,j}}{\partial \mathbf{T}_{i,:}\partial \mathbf{T}_{i,:}} = -\left(\operatorname{diag}(\mathbf{A}_{i,:}) - \mathbf{A}_{i,:}\mathbf{A}_{i,:}^\top\right)\mathbf{A}_{i,j} + (\mathbf{e}_j - \mathbf{A}_{i,:})(\mathbf{e}_j - \mathbf{A}_{i,:})^\top \mathbf{A}_{i,j}$$

$$= \mathbf{A}_{i,j}\left(2\mathbf{A}_{i,:}\mathbf{A}_{i,:}^\top + \mathbf{e}_j\mathbf{e}_j^\top - \operatorname{diag}(\mathbf{A}_{i,:}) - \mathbf{e}_j\mathbf{A}_{i,:}^\top - \mathbf{A}_{i,:}\mathbf{e}_j^\top\right).$$

$\square$

## C.2    SELF-ATTENTION MOMENT MATRICES

**Remark 3.1.** The data terms emerging in the self-attention Hessian can be expressed as functions of the self-attention central moment matrices (proof in appendix C),

$$\mathbf{Z}_1 = \mathbf{X} * \mathbf{M}_2, \quad \text{and} \quad \mathbf{Z}_2 = \left(\mathbf{I}_L \otimes \mathbf{K}_{d_V, d_V} \otimes \mathbf{I}_{d_V}\right)\left(\mathbf{X} * \mathbf{X}^\top * \mathbf{M}_3\right),$$

where $*$ is the Khatri-Rao product (Khatri & Rao, 1968), see definition A.2.

*Proof.* Before starting the derivations, let us specify that for the Khatri-Rao product in the theorem statement, $\mathbf{X}$ is split row-wise and $\mathbf{M}_k$ is split block-row-wise into central moment matrices corresponding to attention rows as in definition 3.1. The proof follows straight from transforming matrices $\mathbf{Z}_1$ and $\mathbf{Z}_2$.

**Dependence on the second central moment matrix $\mathbf{M}_2$.** It holds that

$$\mathbf{Z}_1 = \left(\mathbf{I}_L \otimes \mathbf{X}^\top\right)\frac{\partial \mathbf{A}}{\partial \mathbf{T}}\left(\mathbf{X} \otimes \mathbf{X}\right)$$

$$= \operatorname{blockdiag}\left(\mathbf{X}^\top \frac{\partial \mathbf{A}_{i,:}}{\partial \mathbf{T}_{i,:}}\right)\left(\mathbf{X} \otimes \mathbf{X}\right)$$

$$= \begin{bmatrix} \mathbf{X}_{1,:}^\top \otimes \mathbf{X}^\top \frac{\partial \mathbf{A}_{1,:}}{\partial \mathbf{T}_{1,:}^\top}\mathbf{X} \\ \vdots \\ \mathbf{X}_{L,:}^\top \otimes \mathbf{X}^\top \frac{\partial \mathbf{A}_{L,:}}{\partial \mathbf{T}_{L,:}^\top}\mathbf{X} \end{bmatrix} = \mathbf{X} * \mathbf{M}_2,$$

where the first line follows from the two first matrices in the product being block diagonal and the second from eq. (14).

**Dependence on the third central moment matrix $\mathbf{M}_3$.** Let's recall from theorem 3.2 that

$$
\begin{aligned}
\mathbf{Z}_2 &= \left(\mathbf{I}_L \otimes \mathbf{X}^\top \otimes \mathbf{X}^\top \otimes \mathbf{X}^\top\right) \frac{\partial^2 \mathbf{A}}{\partial \mathbf{T} \partial \mathbf{T}} \left(\mathbf{X} \otimes \mathbf{X}\right) \\
&= \left(\mathbf{I}_L \otimes \mathbf{X}^\top \otimes \mathbf{X}^\top \otimes \mathbf{X}^\top\right) \text{blockdiag}\left(\mathbf{D}_i\right) \left(\mathbf{X} \otimes \mathbf{X}\right).
\end{aligned}
$$

The first two matrices in the above matrix product are block-diagonal with $L$ equal-sized blocks, so their product is also a block-diagonal matrix with $L$ equal-sized blocks of a form $\left(\mathbf{X}^\top \otimes \mathbf{X}^\top \otimes \mathbf{X}^\top\right) \mathbf{D}_i$. Hence, by eq. (14) we know that $\mathbf{Z}_2$ has a block structure $\mathbf{Z}_2 = [\mathbf{Z}_{2;i}]_{1 \leq i \leq L}$, where

$$
\mathbf{Z}_{2;i} = \left(\mathbf{X}^\top \otimes \mathbf{X}^\top \otimes \mathbf{X}^\top\right) \mathbf{D}_i \left(\mathbf{X}_{i,:}^\top \otimes \mathbf{X}\right),
$$

with $\mathbf{X}_{i,:} \in \mathbb{R}^{d_V}$ being the $i^{th}$ row of $\mathbf{X}$ as a column vector.

Let us recall lemma C.1 that gives us the $\mathbf{D}_i$ formula to transform the expression for $\mathbf{Z}_{2;i}$.

$$
\begin{aligned}
\mathbf{Z}_{2;i} &= \left(\mathbf{X}^\top \otimes \mathbf{X}^\top \otimes \mathbf{X}^\top\right) \mathbf{D}_i \left(\mathbf{X}_{i,:}^\top \otimes \mathbf{X}\right) \\
&= \left(\mathbf{X}^\top \otimes \mathbf{X}^\top \otimes \mathbf{X}^\top\right) \left(\left(\mathbb{1}_{L,1} \otimes \mathbf{e}_i\right) * \frac{\partial^2 \mathbf{A}_i}{\partial \mathbf{T}_{i,:} \partial \mathbf{T}_{i,:}}\right) \left(\mathbf{X}_{i,:}^\top \otimes \mathbf{X}\right) \\
&= \begin{bmatrix} \mathbf{x}_1 \otimes \mathbf{X}^\top \otimes \mathbf{X}^\top & \cdots & \mathbf{x}_L \otimes \mathbf{X}^\top \otimes \mathbf{X}^\top \end{bmatrix} \begin{bmatrix} \mathbf{e}_i \otimes \dfrac{\partial^2 \mathbf{A}_{i,1}}{\partial \mathbf{T}_{i,:} \partial \mathbf{T}_{i,:}} \\ \vdots \\ \mathbf{e}_i \otimes \dfrac{\partial^2 \mathbf{A}_{i,L}}{\partial \mathbf{T}_{i,:} \partial \mathbf{T}_{i,:}} \end{bmatrix} \left(\mathbf{X}_{i,:}^\top \otimes \mathbf{X}\right) \\
&= \left(\sum_{j=1}^{L} \left(\mathbf{X}_{j,:} \otimes \mathbf{X}^\top \otimes \mathbf{X}^\top\right) \left(\mathbf{e}_i \otimes \frac{\partial^2 \mathbf{A}_{i,j}}{\partial \mathbf{T}_{i,:} \partial \mathbf{T}_{i,:}}\right)\right) \left(\mathbf{X}_{i,:}^\top \otimes \mathbf{X}\right) \\
&= \left(\sum_{j=1}^{L} \left(\mathbf{X}_{j,:} \otimes \mathbf{X}^\top\right) \mathbf{e}_i \otimes \mathbf{X}^\top \frac{\partial^2 \mathbf{A}_{i,j}}{\partial \mathbf{T}_{i,:} \partial \mathbf{T}_{i,:}}\right) \left(\mathbf{X}_{i,:}^\top \otimes \mathbf{X}\right) \\
&= \sum_{j=1}^{L} \left(\mathbf{X}_{j,:} \otimes \mathbf{X}^\top\right) \mathbf{e}_i \mathbf{X}_{i,:}^\top \otimes \mathbf{X}^\top \frac{\partial^2 \mathbf{A}_{i,j}}{\partial \mathbf{T}_{i,:} \partial \mathbf{T}_{i,:}} \mathbf{X} \\
&= \sum_{j=1}^{L} \mathbf{X}_{j,:} \otimes \mathbf{X}_{i,:} \mathbf{X}_{i,:}^\top \otimes \mathbf{X}^\top \frac{\partial^2 \mathbf{A}_{i,j}}{\partial \mathbf{T}_{i,:} \partial \mathbf{T}_{i,:}} \mathbf{X},
\end{aligned}
$$

where the last three equalities follow from the mixed product property of the Kronecker product. More precisely, the last equality follows from $\left(\mathbf{X}_{j,:} \otimes \mathbf{X}^\top\right) \mathbf{e}_i \mathbf{X}_{i,:}^\top = \left(\mathbf{X}_{j,:}^\top \otimes \mathbf{X}^\top\right) \left(1 \otimes \mathbf{e}_i \mathbf{X}_{i,:}^\top\right) = \mathbf{X}_{j,:} \otimes \mathbf{X}^\top \mathbf{e}_i \mathbf{X}_{i,:}^\top = \mathbf{X}_{j,:} \otimes \mathbf{X}_{i,:} \mathbf{X}_{i,:}^\top$.

Using known properties of the commutation matrix and the mixed product property (eq. (9)), we can further simplify the expression

$$\mathbf{Z}_{2;i} = \sum_{j=1}^{L} \mathbf{X}_{j,:} \otimes \mathbf{X}_{i,:} \mathbf{X}_{i,:}^{\top} \otimes \mathbf{X}^{\top} \frac{\partial^2 \mathbf{A}_{i,j}}{\partial \mathbf{T}_{i,:} \partial \mathbf{T}_{i,:}} \mathbf{X}$$

$$= \sum_{j=1}^{L} \mathbf{K}_{d_V, d_V} \left( \mathbf{X}_{i,:} \mathbf{X}_{i,:}^{\top} \otimes \mathbf{X}_{j,:} \right) \otimes \mathbf{X}^{\top} \frac{\partial^2 \mathbf{A}_{i,j}}{\partial \mathbf{T}_{i,:} \partial \mathbf{T}_{i,:}} \mathbf{X}$$

$$= \sum_{j=1}^{L} \left( \mathbf{K}_{d_V, d_V} \otimes \mathbf{I}_{d_V} \right) \left( \mathbf{X}_{i,:} \mathbf{X}_{i,:}^{\top} \otimes \mathbf{X}_{j,:} \otimes \mathbf{X}^{\top} \frac{\partial^2 \mathbf{A}_{i,j}}{\partial \mathbf{T}_{i,:} \partial \mathbf{T}_{i,:}} \mathbf{X} \right)$$

$$= \left( \mathbf{K}_{d_V, d_V} \otimes \mathbf{I}_{d_V} \right) \left( \mathbf{X}_{i,:} \mathbf{X}_{i,:}^{\top} \otimes \underbrace{\sum_{j=1}^{L} \mathbf{X}_{j,:} \otimes \mathbf{X}^{\top} \frac{\partial^2 \mathbf{A}_{i,j}}{\partial \mathbf{T}_{i,:} \partial \mathbf{T}_{i,:}} \mathbf{X}}_{\mathbf{N}} \right).$$

Finally, after plugging in $\partial^2 \mathbf{A}_{i,j} / \partial \mathbf{T}_{i,:} \partial \mathbf{T}_{i,:}$ from lemma C.1 and extracting $\mathbf{A}_{i,j}$ in front of the Kronecker product we simplify the summation expression $\mathbf{N}$

$$\mathbf{N} = \sum_{j=1}^{L} \mathbf{X}_{j,:} \otimes \mathbf{X}^{\top} \mathbf{A}_{i,j} \left( 2\mathbf{A}_{i,:} \mathbf{A}_{i,:}^{\top} + \mathbf{E}_{j,j}^{L,L} - \mathrm{diag}(\mathbf{A}_{i,:}) - \mathbf{e}_j \mathbf{A}_{i,:}^{\top} - \mathbf{A}_{i,:} \mathbf{e}_j^{\top} \right) \mathbf{X}$$

$$= \sum_{j=1}^{L} \mathbf{A}_{i,j} \mathbf{X}_{j,:} \otimes \left( 2[\mathbf{M}_1]_{i,:}[\mathbf{M}_1]_{i,:}^{\top} + \mathbf{X}_{j,:} \mathbf{X}_{j,:}^{\top} - \mathbf{X}^{\top} \mathrm{diag}(\mathbf{A}_{i,:}) \mathbf{X} - \mathbf{X}_{j,:}[\mathbf{M}_1]_{i,:}^{\top} - [\mathbf{M}_1]_{i,:} \mathbf{X}_{j,:}^{\top} \right).$$

This can be further simplified. We firstly note that

$$[\mathbf{M}_1]_{i,:}[\mathbf{M}_1]_{i,:}^{\top} + \mathbf{X}_{j,:} \mathbf{X}_{j,:}^{\top} - \mathbf{X}_{j,:}[\mathbf{M}_1]_{i,:}^{\top} - [\mathbf{M}_1]_{i,:} \mathbf{X}_{j,:}^{\top} = ([\mathbf{M}_1]_{i,:} - \mathbf{X}_{j,:}) ([\mathbf{M}_1]_{i,:} - \mathbf{X}_{j,:})^{\top}.$$

Moreover, from the fact that Kronecker product distributes over addition (eq. (6))

$$\sum_{j=1}^{L} \mathbf{A}_{i,j} \mathbf{X}_{j,:} \otimes \left( [\mathbf{M}_1]_{i,:}[\mathbf{M}_1]_{i,:}^{\top} - \mathbf{X}^{\top} \mathrm{diag}(\mathbf{A}_{i,:}) \mathbf{X} \right)$$

$$= \left( \sum_{j=1}^{L} \mathbf{A}_{i,j} \mathbf{X}_{j,:} \right) \otimes \left( [\mathbf{M}_1]_{i,:}[\mathbf{M}_1]_{i,:}^{\top} - \mathbf{X}^{\top} \mathrm{diag}(\mathbf{A}_{i,:}) \mathbf{X} \right)$$

$$= [\mathbf{M}_1]_{i,:} \otimes \left( [\mathbf{M}_1]_{i,:}[\mathbf{M}_1]_{i,:}^{\top} - \mathbf{X}^{\top} \mathrm{diag}(\mathbf{A}_{i,:}) \mathbf{X} \right).$$

These two equations give us

$$\mathbf{N} = \sum_{j=1}^{L} \mathbf{A}_{i,j} \mathbf{X}_{j,:} \otimes \left( [\mathbf{M}_1]_{i,:}[\mathbf{M}_1]_{i,:}^{\top} - \mathbf{X}^{\top} \mathrm{diag}(\mathbf{A}_{i,:}) \mathbf{X} \right)$$

$$+ \sum_{j=1}^{L} \mathbf{A}_{i,j} \mathbf{X}_{j,:} \otimes \left( [\mathbf{M}_1]_{i,:}[\mathbf{M}_1]_{i,:}^{\top} + \mathbf{X}_{j,:} \mathbf{X}_{j,:}^{\top} - [\mathbf{M}_1]_{i,:} \mathbf{X}_{j,:}^{\top} - \mathbf{X}_{j,:}[\mathbf{M}_1]_{i,:}^{\top} \right)$$

$$= [\mathbf{M}_1]_{i,:} \otimes \left( [\mathbf{M}_1]_{i,:}[\mathbf{M}_1]_{i,:}^{\top} - \mathbf{X}^{\top} \mathrm{diag}(\mathbf{A}_{i,:}) \mathbf{X} \right)$$

$$+ \sum_{j=1}^{L} \mathbf{A}_{i,j} \mathbf{X}_{j,:} \otimes ([\mathbf{M}_1]_{i,:} - \mathbf{X}_{j,:}) ([\mathbf{M}_1]_{i,:} - \mathbf{X}_{j,:})^{\top}.$$

We are now at the finish line of obtaining the desired formula. Let us note that similar expression appear in both summands of the equation above, specifically

$$\left( \mathbf{X}^{\top} \mathrm{diag}(\mathbf{A}_{i,:}) \mathbf{X} - [\mathbf{M}_1]_{i,:}[\mathbf{M}_1]_{i,:}^{\top} \right) = \sum_{j=1}^{L} \mathbf{A}_{i,j} ([\mathbf{M}_1]_{i,:} - \mathbf{X}_{j,:}) ([\mathbf{M}_1]_{i,:} - \mathbf{X}_{j,:})^{\top}.$$

Hence,

$$
\begin{aligned}
\mathbf{N} &= [\mathbf{M}_1]_{i,:} \otimes \left([\mathbf{M}_1]_{i,:}[\mathbf{M}_1]_{i,:}^\top - \mathbf{X}^\top \operatorname{diag}(\mathbf{A}_{i,:})\mathbf{X}\right) \\
&\quad + \sum_{j=1}^{L} \mathbf{A}_{i,j}\mathbf{X}_{j,:} \otimes \left([\mathbf{M}_1]_{i,:} - \mathbf{X}_{j,:}\right)\left([\mathbf{M}_1]_{i,:} - \mathbf{X}_{j,:}\right)^\top \\
&= -\sum_{j=1}^{L} \mathbf{A}_{i,j}[\mathbf{M}_1]_{i,:} \otimes \left([\mathbf{M}_1]_{i,:} - \mathbf{X}_{j,:}\right)\left([\mathbf{M}_1]_{i,:} - \mathbf{X}_{j,:}\right)^\top \\
&\quad + \sum_{j=1}^{L} \mathbf{A}_{i,j}\mathbf{X}_{j,:} \otimes \left([\mathbf{M}_1]_{i,:} - \mathbf{X}_{j,:}\right)\left([\mathbf{M}_1]_{i,:} - \mathbf{X}_{j,:}\right)^\top \\
&= \sum_{j=1}^{L} \mathbf{A}_{i,j}\left(\mathbf{X}_{j,:} - [\mathbf{M}_1]_{i,:}\right) \otimes \left([\mathbf{M}_1]_{i,:} - \mathbf{X}_{j,:}\right)\left([\mathbf{M}_1]_{i,:} - \mathbf{X}_{j,:}\right)^\top \\
&= [\mathbf{M}_3]_{i,:}.
\end{aligned}
$$

After plugging in the above into the $\mathbf{Z}_{2;i}$ formula, we obtain

$$
\mathbf{Z}_{2;i} = \left(\mathbf{K}_{d_V, d_V} \otimes \mathbf{I}_{d_V}\right)\left(\mathbf{X}_{i,:}\mathbf{X}_{i,:}^\top \otimes [\mathbf{M}_3]_{i,:}\right).
$$

Finally, note that

$$
\mathbf{X} * \mathbf{X}^\top * \mathbf{M}_3 = [\mathbf{X}_{i,:}\mathbf{X}_{i,:}^\top \otimes [\mathbf{M}_3]_{i,:}]_{1 \le i \le L} \in \mathbb{R}^{L d_V^2 \times d_V},
$$

and hence

$$
\mathbf{Z}_2 = \left(\mathbf{I}_L \otimes \mathbf{K}_{d_V, d_V} \otimes \mathbf{I}_{d_V}\right)\left(\mathbf{X} * \mathbf{X}^\top * \mathbf{M}_3\right).
$$

$\square$

## D   MORE ON THE QUERY-KEY HESSIAN BLOCK

**Nested structure of the query-key Hessian.**    The query-key Hessian blocks exhibit a nested structure which prompts us to define a Gauss-Newton-like decomposition of the query-key Hessian blocks. The mixed query-key block of the Hessian consists of three summands—one coming from the $\mathbf{F}$-outer-product Hessian and two from the $\mathbf{F}$-functional Hessian. The $\mathbf{F}$-outer-product term as well as one of the $\mathbf{F}$-functional terms are structurally similar to the query blocks we can find on the diagonal. Together they can be expressed as an outer product. This observation results in the decomposition from remark D.1.

**Remark D.1.** The query-key part of the Hessian from theorems 3.1 and 3.2 can be equivalently decomposed into a sum of $\mathbf{T}$-outer-product and $\mathbf{T}$-functional Hessians, respectively given by

$$
\mathbf{H}_o^{\mathbf{T}}\left(\mathbf{W}_{\mathrm{QK}}, \mathbf{W}_{\mathrm{QK}}\right) =
$$
$$
\frac{1}{d_K}\mathbf{V}^\top \underbrace{\left(\mathbf{Z}_1^\top\left(\mathbf{I}_L \otimes \mathbf{W}_V \mathbf{W}_V^\top\right)\mathbf{Z}_1 + \left(\delta_{\mathbf{XY}}^\top\left(\mathbf{I}_L \otimes \mathbf{W}_V^\top\right) \otimes \mathbf{I}_{d_V^2}\right)\mathbf{Z}_2\right)}_{\mathbf{U}}\mathbf{V},
$$

$$
\mathbf{H}_f^{\mathbf{T}}\left(\mathbf{W}_{\mathrm{QK}}, \mathbf{W}_{\mathrm{QK}}\right) = \frac{1}{\sqrt{d_K}}\begin{bmatrix} \mathbf{0} & \mathbf{B}^\top \otimes \mathbf{I}_{d_K} \\ \mathbf{B} \otimes \mathbf{I}_{d_K} & \mathbf{0} \end{bmatrix} = \frac{1}{\sqrt{d_K}}\begin{bmatrix} \mathbf{0} & \mathbf{B}^\top \\ \mathbf{B} & \mathbf{0} \end{bmatrix} \otimes \mathbf{I}_{d_K}.
$$

In the above formulas $\mathbf{W}_{\mathrm{QK}} := \begin{bmatrix} \mathbf{W}_Q \\ \mathbf{W}_K \end{bmatrix}$, $\delta_{\mathbf{XY}} := \operatorname{vec}_r\left(\mathbf{F}(\mathbf{X}) - \mathbf{Y}\right)$ and,

$$
\mathbf{V} := \left[\left(\mathbf{W}_Q \otimes \mathbf{I}_{d_V}\right)\mathbf{K}_{d_K, d_V} \quad \mathbf{I}_{d_V} \otimes \mathbf{W}_K\right], \quad \mathbf{B} := \mathbf{R}_{d_V}\left(\mathbf{I}_L \otimes \mathbf{W}_V^\top \otimes \mathbf{I}_{d_V}\right)\left(\mathbf{Z}_1 \otimes \mathbf{I}_{d_V}\right)\mathbf{S}.
$$

This decomposition can be thought of as a Gauss-Newton decomposition when we split the function composition $\ell \circ (\mathbf{A} \mapsto \mathbf{AXW}_V) \circ a \circ \mathbf{T}$ at the level of $\mathbf{T}(\mathbf{X}) = \mathbf{XW}_Q\mathbf{W}_K^\top\mathbf{X}^\top / \sqrt{d_K}$.

**Query-key Hessian eigenspectrum.** The structure of the query-key Hessian block implies a specific configuration of the eigenvalues of its summands. The $\mathbf{T}$-functional Hessian has a characteristic block-hollow structure with blocks of zeros on the diagonal, which makes it responsible for the bulk eigenvalues of the query-key Hessian. This specific structure allows us to reason about the eigenspectrum of the query-key Hessian. Eigenvalues of $\mathbf{T}$-functional Hessian come in pairs $\pm\lambda_i$ for $1 \leq i \leq d_V$. To see that it is enough to note that if $\lambda_i$ is an eigenvalue with an eigenvector $[\mathbf{v}_1^\top, \mathbf{v}_2^\top]^\top$, then also $[-\mathbf{v}_1^\top, \mathbf{v}_2^\top]^\top$ is an eigenvector with corresponding eigenvalue $-\lambda_i$.

Moreover, by theorem 2.1 from Magnus & Neudecker (2019) eigenvalues of the Kronecker product are products of the factor matrices' eigenvalues. Since all $d_K$ eigenvalues of $\mathbf{I}_{d_K}$ are ones, eigenvalues of $\mathbf{T}$-functional Hessian are the same as eigenvalues of

$$\frac{1}{\sqrt{d_K}} \begin{bmatrix} \mathbf{0} & \mathbf{B}^\top \\ \mathbf{B} & \mathbf{0} \end{bmatrix},$$

each with multiplicity $d_K$. This is exactly like the eigenvalue structure of the functional Hessian of a two-layer MLP, where the eigenvalues are known to come in positive-negative pairs Singh et al. (2021), each with multiplicity $d_K$.

Furthermore, the $\mathbf{T}$-outer-product Hessian has at most $2d_K d_V - d_K^2$ non-zero eigenvalues when $d_V > d_K$ and at most $d_V^2$ non-zero eigenvalues when $d_V \leq d_K$. This is because of the rank bound from lemma D.2 and the fact that rank is equal to the number of non-zero eigenvalues for symmetric matrices.

**Lemma D.1.** Singh et al. (2021)
Let $\mathbf{A} \in \mathbb{R}^{m \times n}$ and $\mathbf{B} \in \mathbb{R}^{p \times q}$. Then

$$\mathrm{rk}\left(\begin{bmatrix} \mathbf{I}_q \otimes \mathbf{A} \\ \mathbf{B} \otimes \mathbf{I}_n \end{bmatrix}\right) = q\,\mathrm{rk}\,(\mathbf{A}) + n\,\mathrm{rk}\,(\mathbf{B}) - \mathrm{rk}\,(\mathbf{A})\,\mathrm{rk}\,(\mathbf{B}).$$

**Lemma D.2.** Under the same assumptions as remark D.1, and additionally assuming that $\mathbf{W}_K$ and $\mathbf{W}_Q$ are full rank, the rank of the $\mathbf{T}$-outer-product Hessian can by bounded by

$$\mathrm{rk}\left(\mathbf{H}_o^{\mathbf{T}}(\mathbf{W}_{QK}, \mathbf{W}_{QK})\right) \leq \begin{cases} 2d_K d_V - d_K^2 & \text{if } d_K < d_V \\ d_V^2 & \text{if } d_K \geq d_V. \end{cases}$$

*Proof.*

$$\mathrm{rk}\left(\mathbf{H}_o^{\mathbf{T}}(\mathbf{W}_{QK}, \mathbf{W}_{QK})\right) = \mathrm{rk}\left(\frac{1}{d_K}\mathbf{V}^\top \mathbf{U} \mathbf{V}\right) \leq \min\left(\mathrm{rk}\,(\mathbf{U}), \mathrm{rk}\,(\mathbf{V}^\top)\right) \leq \mathrm{rk}\,(\mathbf{V}^\top).$$

To get the rank of $\mathbf{V}^\top$ we use lemma D.1.

$$\begin{aligned}
\mathrm{rk}\,(\mathbf{V}^\top) &\leq \mathrm{rk}\left(\begin{bmatrix} \mathbf{W}_Q^\top \otimes \mathbf{I}_{d_V} \\ \mathbf{I}_{d_V} \otimes \mathbf{W}_K^\top \end{bmatrix}\right) \\
&= \mathrm{rk}\left(\begin{bmatrix} \mathbf{I}_{d_V} \otimes \mathbf{W}_K^\top \\ \mathbf{W}_Q^\top \otimes \mathbf{I}_{d_V} \end{bmatrix}\right) \\
&= d_V\,\mathrm{rk}\,(\mathbf{W}_K) + d_V\,\mathrm{rk}\,(\mathbf{W}_Q) - \mathrm{rk}\,(\mathbf{W}_K)\,\mathrm{rk}\,(\mathbf{W}_Q) \\
&= \begin{cases} 2d_V d_K - d_K^2 & \text{when } d_K < d_V, \\ d_V^2 & \text{when } d_K \geq d_V. \end{cases}
\end{aligned}$$

$\square$

# E   MULTI-HEAD SELF-ATTENTION HESSIAN

Multi-head self-attention, a mechanism that allows the model to jointly attend to information from different representation subspaces (Vaswani et al., 2017), enforces an interesting structure in self-attention Jacobian and the Hessian of the self-attention loss. A multi-head self-attention layer can

be defined as

$$\mathbf{F}(\mathbf{X}) = \sum_h^H \mathbf{F}^h(\mathbf{X}) = \sum_h^H \mathbf{A}^h(\mathbf{X})\mathbf{X}\mathbf{W}_V^h \quad \text{where} \quad \mathbf{A}^h(\mathbf{X}) = a\left(\mathbf{T}^h(\mathbf{X})\right),$$

where $\mathbf{T}^h$ do not share weights. For example, in the classical definition of multi-head self-attention, we have $\mathbf{T}^h(\mathbf{X}) = \mathbf{X}\mathbf{W}_Q^h \mathbf{W}_K^{h\top} \mathbf{X}^\top / \sqrt{d_K}$ with different $\mathbf{W}_Q^h$ and $\mathbf{W}_K^h$ for every head.

**Multi-head classical self-attention Jacobian blocks depend only on single-head parameters.** Note that $\mathbf{F}$ is a sum of completely independent functions of $\mathbf{X}$, as every weight matrix enters $\mathbf{F}^h$ for exactly one $h$. This implies that a block of the Jacobian corresponding to weights parameterizing $\mathbf{F}^h$ depends only on $\mathbf{W}_K^h, \mathbf{W}_Q^h$ and $\mathbf{W}_V^h$. Recalling the formulas for the gradient from lemma B.1, we arrive at the following remark E.1.

> **Remark E.1.** The Jacobians of the multi-head classical self-attention layer (Vaswani et al., 2017) have the following form:
>
> $$\frac{\partial \mathbf{F}}{\partial \mathbf{W}_V^h} = \frac{\partial \mathbf{F}^h}{\partial \mathbf{W}_V^h} = \text{softmax}\left(\frac{\mathbf{X}\mathbf{W}_Q^h \mathbf{W}_K^{h\top} \mathbf{X}^\top}{\sqrt{d_K}}\right)\mathbf{X} \otimes \mathbf{I}_{d_V},$$
>
> $$\frac{\partial \mathbf{F}}{\partial \mathbf{W}_Q^h} = \frac{\partial \mathbf{F}^h}{\partial \mathbf{W}_Q^h} = \left(\mathbf{I}_L \otimes \mathbf{W}_V^{h\top} \mathbf{X}^\top\right)\frac{\partial \mathbf{A}^h}{\partial \mathbf{T}^h}\left(\frac{\mathbf{X} \otimes \mathbf{X}\mathbf{W}_K^h}{\sqrt{d_K}}\right),$$
>
> where the Jacobian of the row-wise softmax w.r.t. its inputs is defined in lemma B.1.

**With an increasing number of heads, the outer-product Hessian dominates the Hessian.** Note that the mixed inter-head terms of the Hessian are fully defined by the outer-product Hessian part. This is because the inter-head functional Hessian blocks are always zero, as the second derivative of $\mathbf{F}$ w.r.t. arbitrary inter-head weight matrices $\mathbf{W}_c^{h_i} \in \mathbb{R}^{p_c \times q_c}$ and $\mathbf{W}_t^{h_j} \in \mathbb{R}^{p_t \times q_t}$ for $h_i \neq h_j$ is zero. This results from the Jacobian expressions in remark E.1, which always depend on weight matrices parameterizing just a single self-attention head. Hence,

$$\mathbf{H}_{\text{f}}\left(\mathbf{W}_c^{h_i}, \mathbf{W}_t^{h_j}\right) = \left(\frac{\partial \ell}{\partial \mathbf{F}}(\mathbf{F}(\cdot)) \otimes \mathbf{I}_{p_c q_c}\right)\underbrace{\frac{\partial^2 \mathbf{F}}{\partial \mathbf{W}_c^{h_i} \partial \mathbf{W}_t^{h_j}}(\cdot)}_{=0, \text{ when } h_i \neq h_j} = 0.$$

This implies that, for a fixed $d_V$ and $d_K = d_V/H$, the functional Hessian becomes increasingly sparse as the number of heads grows. This leads to most entries of the Hessian being fully defined by the outer-product Hessian. Notably, this observation holds irrespective of the loss function used.

### E.1 INFLUENCE OF THE LOW-RANK NATURE OF $\mathbf{W}_V$ ON THE SELF-ATTENTION HESSIAN

A practical implementation of the multi-head self-attention assumes that $\mathbf{W}_V^h$ is low-rank, namely, it is parameterized as $\mathbf{W}_V^h = \mathbf{W}_O^h \mathbf{W}_U^h$, for some $\mathbf{W}_O^h \in \mathbb{R}^{d_V \times d_K}$ and $\mathbf{W}_U^h \in \mathbb{R}^{d_K \times d_V}$. In this setting, we compute the Hessian w.r.t. $\mathbf{W}_O^h$ and $\mathbf{W}_U^h$ instead of $\mathbf{W}_V^h$ and the only self-attention Hessian blocks that get affected are the ones w.r.t. $\mathbf{W}_O^h$ or $\mathbf{W}_U^h$.

**The $(\mathbf{W}_O^h, \mathbf{W}_U^h)$ Hessian block resembles the Hessian of a linear two-layer MLP.** Note that

$$\mathbf{F}^h(\mathbf{X}) = \mathbf{A}^h(\mathbf{X})\mathbf{X}\mathbf{W}_O^h \mathbf{W}_U^h = \mathbf{M}_1 \mathbf{W}_O^h \mathbf{W}_U^h \tag{24}$$

resembles a two-layer linear MLP if we treated $\mathbf{M}_1$ as a single input matrix. Hence, the discussion from section 2 on the Hessian of MLPs applies to the value matrix block of the self-attention Hessian. For instance, its diagonal consists entirely of the outer-product Hessian entries, since its functional Hessian counterpart has a block-hollow structure.

The main difference compared to a full rank parameterization by a single matrix $\mathbf{W}_V^h$ is that now the Hessian also includes mixed blocks w.r.t. $\mathbf{W}_O^h$ and $\mathbf{W}_U^h$. For the MSE loss, the functional part

of these blocks has a quadratic dependence on the first moment matrix $\mathbf{M}_1$, as both the derivative of the loss w.r.t. the model output $\partial \ell / \partial \mathbf{F}$, and the mixed second derivative matrix of $\mathbf{F}^h$ w.r.t. $\mathbf{W}_{\mathrm{O}}^h$ and $\mathbf{W}_{\mathrm{U}}^h$ (see eq. (2)) depend on it linearly. Additionally, the outer-product Hessian block w.r.t. the value matrices $\mathbf{W}_{\mathrm{O}}^h$ and $\mathbf{W}_{\mathrm{U}}^h$ has an extra quadratic dependence on the selected value matrices themselves and on the first moment matrix $\mathbf{M}_1$.

## F    EXPERIMENTAL SETUP

For the numerical experiments, we adapt the setting from Quirke & Barez (2024). They frame number addition as the next token prediction task and generate a custom dataset, which we also use in our experiments. For 5-digit number addition that we use, the sequence length is equal to $L = 17$. The model we consider is (unless stated otherwise) a single-block GPT-2 Transformer (Radford et al., 2019) from TransformerLens (Nanda & Bloom, 2022), in most experiments without layer normalization unless noted otherwise. The tokens and their positions are embedded and passed into a single-head self-attention layer, followed by a two-layer MLP. In all figures except for figs. 7 and 8 we use cross-entropy (CE) as a loss function.

To obtain figs. 1 and 4 we initialize the weights of the model the same way as GPT-2, so biases are initialized to $0$ and weights are initialized by sampling from $\mathcal{N}(0, 0.64/d_V)$. In fig. 1 we use the classical definition of self-attention, while in fig. 4 we additionally compare to self-attention without softmax. The size of the model is $d_V = d_K = 16$, and the latent dimension of the MLP is $64$. The Hessian is computed using $64$ data samples.

To obtain figs. 3 and 5 we vary the standard deviation of the initialization of the embedding layer while leaving the initialization scheme of the rest of the model unchanged. We compare (fig. 3a) the model without layer normalization with (fig. 3b) a model with Pre-LN (meaning that layer normalization is applied before the self-attention layer, before the MLP layer, and after the last Transformer block as in GPT-2 Radford et al. (2019)). The size of the Transformer block is $d_V = d_K = 128$ and the Hessian is computed using $64$ sequences. We estimate the block Frobenius norm using the Hutchinson trace estimator of the block outer product, implemented in the CurvLinOps library (Dangel et al., 2025). Every configuration is repeated $20$ times and we report the mean and standard error of the mean. For the Pre-LN we select the slope of the dashed trend lines by fitting a linear regression model into pairs of points $(\log \sigma, \log \bar{f}(\sigma))$ where $\bar{f}(\sigma)$ is the mean Frobenius norm of the block, estimated at $\sigma$.

The setting of fig. 6 is the same as in fig. 3a with the only difference that we consider multi-layer GPT-2 Transformers without layer normalization.

To obtain fig. 7 we use multi-layer linear self-attention networks, meaning that we simply chain self-attention layers w/o softmax. The loss we use in this experiment is mean squared error (MSE). Similar to the experiment in fig. 3, we vary the standard deviation $\sigma$ used to initialize the embedding matrix $\mathbf{X}$.

To obtain fig. 8 we use linear MLPs with the hidden dimension $128$. The loss used in this experiment is also MSE.

## G    ADDITIONAL EXPERIMENTAL RESULTS

**Hessian blocks at different layers of a multi-layer Transformer follow the growth rates predicted for a single-layer self-attention model.**    With the experiment in fig. 6 we try to answer a question: How well do our results for a single-layer network generalize to an isolated layer inside a deep network? To do that we empirically demonstrate how the self-attention (with softmax) Hessian blocks of multi-layer Transformers scale with the input matrix $\mathbf{X}$ scale $\sigma$.

We observe that for a realistic range of $\sigma \in (0, 1)$, which we refer to as the practical range, the query outer-product and functional blocks as well as the value outer-product block scale exactly as the prediction for a single layer. Moreover, the lines corresponding to earlier layers are lower on the log-log scale for $\sigma \in (0, 1)$, which means that their Hessian blocks have smaller multipliers as part of their expressions.

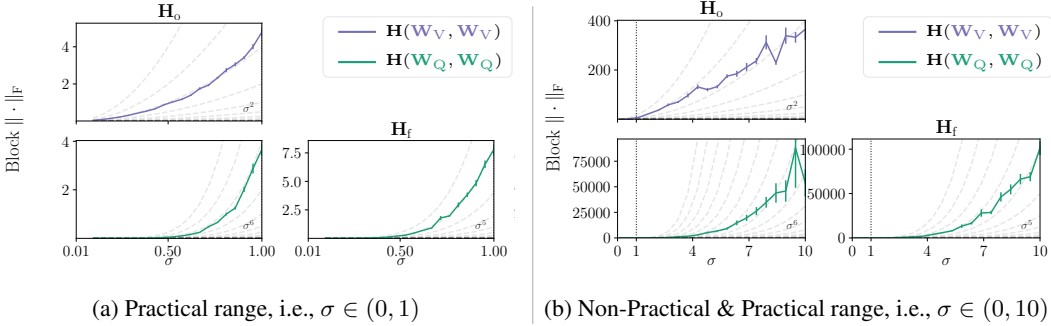

(a) Practical range, i.e., $\sigma \in (0, 1)$        (b) Non-Practical & Practical range, i.e., $\sigma \in (0, 10)$

Figure 5: (Plotted in linear scale.) **Empirical verification with a CE loss confirms derived growth rates w.r.t. magnitude $\sigma$ of X from eq. (5).** We demonstrate the growth rates through the Frobenius norm $\| \cdot \|_F$ of value and query diagonal blocks for (a) practical range $\sigma \in (0, 1)$ and (b) bigger $\sigma$ values $\sigma \in (0, 10)$. The dashed lines correspond to the trend predicted by theory as in eq. (5). For details on the experimental setting, see appendix F. **This figure presents the same data as in fig. 3a but using a linear scale on both axes instead of a log-log scale.**

For $\sigma > 1$ and deeper layers we start observing some higher-order dependencies on $\sigma$. This suggests that similarly to the deep linear attention networks (see section 4.1), the Hessian of a multi-layer Transformer with softmax attention exhibits a dependence on X that grows with depth. These higher order dependencies are not visible for $\sigma < 1$, because they converge to zero quicker than the lower order dependencies. We also note, that the Frobenius norm of the query Hessian block and the value outer-product Hessian block corresponding to the top layer quite closely follow our prediction for a single self-attention layer, also for deeper networks and $\sigma > 1$.

For layers other than the top one, the value functional Hessian block does not follow our theoretical prediction, because, for deeper layers, this Hessian block no longer equals zero. Note that the top layer (largest layer identifier in the plot) is not present in the figure because it equals zero.

**Empirical comparison between self-attention networks and MLPs.** In figs. 7 and 8 we compare the diagonal blocks of the linear self-attention and MLP Hessians. In fig. 7a we observe the growth rates predicted by remark 4.1 for a single layer of linear self-attention. As expected, the diagonal blocks are fully driven by the outer-product Hessian, as the functional Hessian blocks equal zero. For growing network depth $D$, we observe that the growth rates of the outer-product Hessian $\mathbf{H}_o$ blocks become super-exponential. The complete Hessian $\mathbf{H}$ blocks also follow this trend—the last layer for any $\sigma$ and the remaining ones for bigger $\sigma$. This is inline with the discussion in section 4.1.

In contrast, the diagonal blocks of a linear MLP Hessian fig. 8 grow quadratically with $\sigma$, for any of the tested network depths.

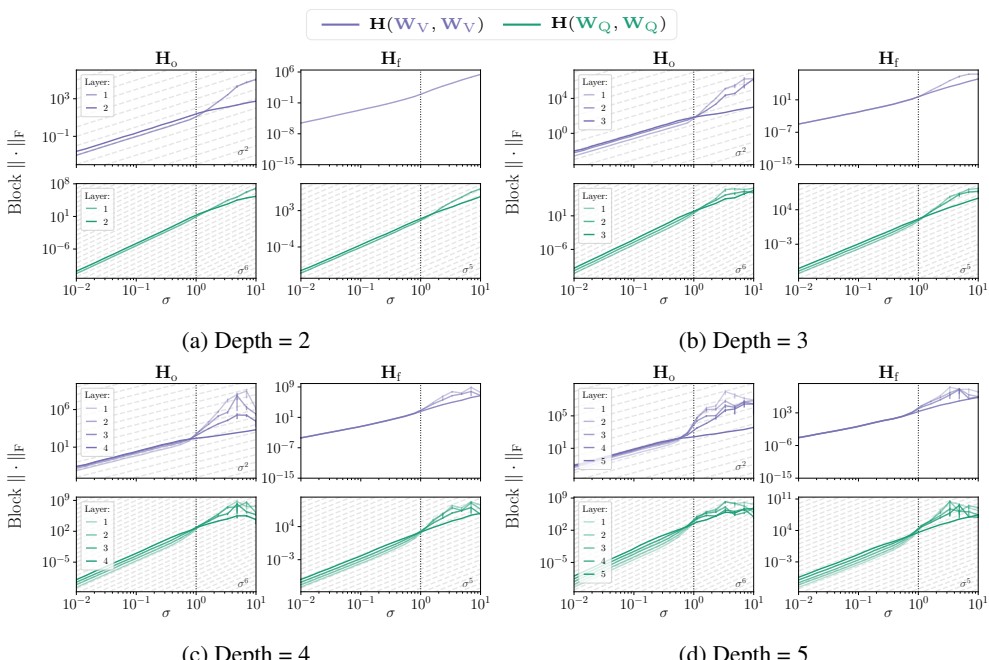

Figure 6: (Plotted in log-log scale.) **Value and query Hessian diagonal blocks at different layers follow the predicted theoretical growth rates for practical ranges of the input.** Frobenius norm $\|\cdot\|_F$ of the self-attention Hessian blocks for multi-layer GPT-2 Transformers without layer normalization on the next token prediction task, split by Transformer block (1 corresponds to the input Transformer block). We indicate the growth rates predicted by theorems 3.1 and 3.2 with the gray dashed lines and the annotation in the bottom right corners. As for the single layer, the complete Hessian $\mathbf{H}$ value and query blocks follow the trend of the outer-product and functional Hessian blocks respectively.

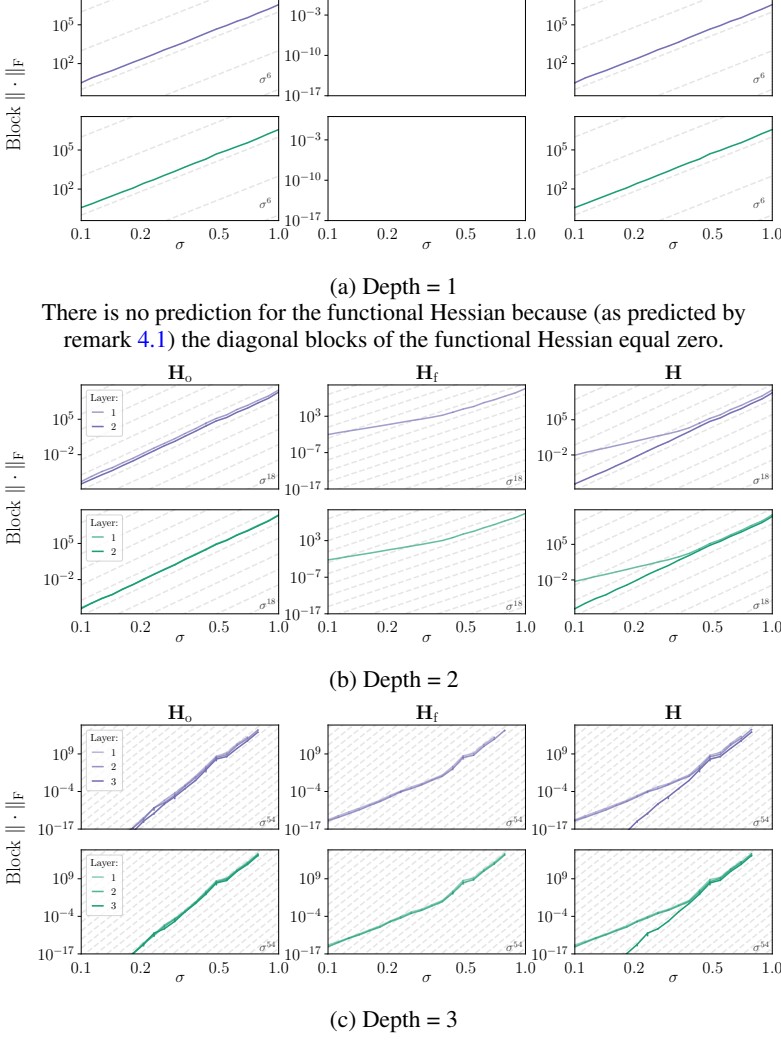

(a) Depth = 1

There is no prediction for the functional Hessian because (as predicted by remark 4.1) the diagonal blocks of the functional Hessian equal zero.

(b) Depth = 2

(c) Depth = 3

Figure 7: (Plotted in log-log scale.) **In linear self-attention networks value and query Hessian diagonal blocks at different layers partially follow the super-exponential growth rate with depth.** Frobenius norm $\| \cdot \|_F$ of the self-attention Hessian blocks for multi-layer self-attention network on the next token prediction task, split by layer (1 corresponds to the input layer). We limit the range of sigma and the network depth, due to numerical problems caused by the super-exponential growth rate for larger $\sigma$ and deeper networks. The dashed lines indicate the trend $\sigma^{2\cdot 3^D}$, where $D$ is the network depth.

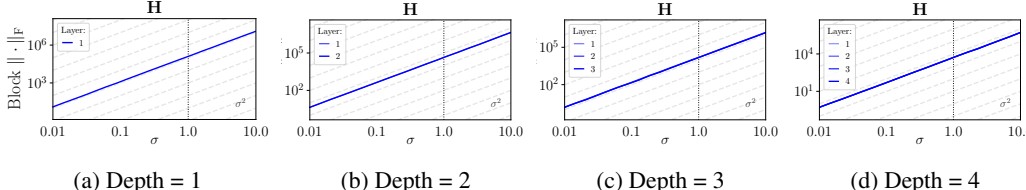

(a) Depth = 1           (b) Depth = 2           (c) Depth = 3           (d) Depth = 4

Figure 8: (Plotted in log-log scale.) **Diagonal blocks of a linear MLP grow the same with $\sigma$ irrespective of network depth.** Frobenius norm $\| \cdot \|_F$ of diagonal Hessian blocks for a linear MLP on the next token prediction task, split by layer (1 corresponds to the input layer). For a linear MLP the diagonal blocks of a functional Hessian are always zero (Singh et al., 2021), so we simply plot the complete Hessian diagonal blocks, without splitting them into outer-product and functional Hessians. We plot the block Frobenius norm separately for every layer, but they perfectly overlap.

