# OpenReview forum: "What Does It Mean to Be a Transformer? Insights from a Theoretical Hessian Analysis"
_ICLR.cc/2025/Conference — ICLR 2025 Spotlight_

### Official Review · Reviewer_gfTd · 2024-10-28

**Soundness:** 3
**Presentation:** 3
**Contribution:** 3
**Rating:** 8
**Confidence:** 3

**Summary:**

The paper is interested in deriving the full expression of the Hessian for a single self-attention layer, wrt the learned matrix parameters of query, key and values. The hessian is decomposed into two terms, the outer product and functional hessians, and their expressions are respectively given in Theorems 3.1 and 3.2. Then, the paper analyzes the dependence on the data and how different components of the architecture affect the hessian, such as the softmax activation or the position of the layer normalization.

**Strengths:**

- **Originality**: To my knowledge, this is the first paper deriving the full expression of the hessian for the self-attention operation.
- **Significance**: As mentioned in the conclusion of the paper, this work can serve as foundation for better understanding the role of the self attention operation in Transformers. As discussed and shown throughout the paper, the self attention layer has a singular behavior compared to better-understood convolutional or feed-forward layers in neural networks.
- **Quality**: Although I did not check all the proofs in details, a lot of work has been put to derive Theorems 3.1 and 3.2. The experiments presented in Figure 3 also validates to some extent the theoretical results obtained, in terms of dependence to the training data of two of the diagonal terms.
- **Clarity**: I appreciated the color-coding of the terms within equations throughout the paper. It makes the reading and understanding of the results easier.

**Weaknesses:**

- **Clarity**: The links between the empirical results shown in the various figures and the insights derived from the expressions of the Hessian are not always clear. For instance, the experiments and what is plotted in Figure 1 are never described.
- **Quality**: It is difficult to evaluate the validity of all theoretical insights derived from the hessian since the settings of the experiments are not always described. More specifically, settings behind experiments to obtain Figure 1, Figure 3 and Figure 4.

**Questions:**

- I would be interested in having more details about the settings of the experiments leading to the figures shown in the paper, more precisely for Figure 1 and Figure 4, and the dashed lines in Figure 3. What is exactly plotted ? What kind of data were used to obtain these ? How does it confirm the insights derived from the theoretical derivations ?
- In Figure 3b, what does "the dashed lines correspond to the trend estimated from the data points by the linear regression coefficient" mean ? Can the authors describe the setting behind this experiment and how the dashed lines are obtained ?
- In Figure 3, all the trends in dashed lines are linear, even though the order of the dependence is changing. This makes me think that the range of values considered for $\sigma$ is too small to clearly evaluate whether the empirical dependence are following the theoretical ones. Can the authors discuss that, and if possible, show results with a bigger range of values for $\sigma$ ?

---

> ### Author Response · Authors · 2024-11-25
> **Response (1/2) to Reviewer gfTd**
>
> We thank the reviewer for their feedback. We addressed their concerns about the clarity of the experimental setting by
> adding a more detailed description of the experimental setup in Appendix F and
> by updating the figures' descriptions.
>
> While answering the reviewer questions, we have split question 1 into multiple answers. We also clarified the link between the theory and experiments (see the last question).
>
> > I would be interested in having more details about the settings of the experiments [...] What kind of data were used to obtain these?
>
> We adapted the experimental setup from Quirke & Barez (2024) and used their dataset, which frames digit addition as a next token prediction task. For more information please see Appendix F.
>
> > I would be interested in having more details about the settings of the experiments [...], more precisely for Figure 1 and Figure 4 [...]. What is exactly plotted?
>
> * To obtain Figure 1a and Figure 4:
>     * We consider a single Transformer block without layer normalization (for details see Appendix F) and the cross-entropy loss. The model in Figure 1a uses a classical self-attention layer and for Figure 4 we compare models with classical self-attention and self-attention without softmax.
>     * We compute the diagonal blocks of the Hessian corresponding to the query and value weight matrices and plot the histogram of the absolute values of their entries.
>
> * Figure 1b depicts the full Hessian of the model with classical self-attention.
>
> * Figure 3 depicts the Frobenius norm of the Hessian blocks for different standard deviations of the distribution used to initialize the embedding matrices. We again use a single block GPT-2 Transformer (see Appendix F).
>
> > In Figure 3, all the trends in dashed lines are linear, even though the order of the dependence is changing. This makes me think that the range of values considered for σ is too small to clearly evaluate whether the empirical dependence are following the theoretical ones. Can the authors discuss that, and if possible, show results with a bigger range of values for σ?
>
> * We believe that there is a misunderstanding of the axis scales used in Figure 3. The scale is logarithmic on both axes. Note that on a log-log plot, a power-law relationship appears as a straight line with the slope indicating the power used in the power function. As most of our theoretically derived dependencies on $\mathbf{X}$ are its power functions, we expect them to be lines on the log-log scale. Our theory dictates the slopes the lines should have, and empirically they do.
>
> * Nevertheless, as per the reviewer’s request, we extended the range of $\sigma$ from the original $(0.1, 2.0)$ to $(0.01, 10.0)$ and confirmed that the empirical dependences align with the theoretical predictions across this broader range.
> * The alignment is especially good for the practical, smaller $\sigma \in (0.01, 1)$.
> * In Figure 5 of the revision, we included a plot with a linear scale on both axes to complement the log-log plot in the main text.
>
> > In Figure 3b, what does "the dashed lines correspond to the trend estimated from the data points by the linear regression coefficient" mean? Can the authors describe the setting behind this experiment and how the dashed lines are obtained?
>
> * For a Transformer block with layer normalization, we don’t have a theoretical prediction of the Hessian block norm and the scale of $\mathbf{X}$. Nevertheless, we wanted to experimentally demonstrate, that when layer norm is applied, the trends we observe for the query and value blocks become more similar.
>
> * Since the theoretical trends we have for the case without layer norm are power functions or polynomials, we wanted to know, a power function of what degree best fits the Frobenius norm measurement we experimentally obtain for the Transformer with layer norm. As we noted in our previous answer, the power function on the log-log plot should be a line with a slope corresponding to its degree. Hence, obtaining the degree is equivalent to finding the slope of the best-fitting line in the log-log scale.
>
> * To do that:
>     1. We fitted a linear regression model to points $(\log{\sigma}), \log{\bar{f}(\sigma)})$ where $\bar{f}$ is the empirical Frobenius norm of the Hessian block plotted in the figure.
>     2. We then considered the fitted linear regression coefficient as the slope $s$.
>     3. Finally, we plotted the dashed lines  $c \cdot \sigma^s$, where $c$ were some selected integers.

---

> ### Author Response · Authors · 2024-11-25
> **Response (2/2) to Reviewer gfTd**
>
> >The links between the empirical results shown in the various figures and the insights derived from the expressions of the Hessian are not always clear. [...] How does [the experiment] confirm the insights derived from the theoretical derivations?
>
> Here we clarify the links between the experiments that we ran and our derivations and insights.
>
> 1. The main takeaways of Fig. 3 are:
>     * In Fig. 3a (no LN) the scalings of green and purple lines are **different** i.e. there is  hetereogeneity between value and query Hessians in the absence of LN
>     * In Fig. 3b (LN) the scalings of green and purple lines are **similar** i.e. there is less hetereogeneity between value and query Hessians in the presence of LN
>
>     We can see that the exponents are more similar in the presence of LN:
>     * No LN (3a): Differences in exponents by 4 and 5
>     * With LN (3b): Differences in exponents by 0 and 0.6
>
>     Figure 3a additionally confirms our theoretical dependencies on the input matrix $\mathbf{X}$. We amended the discussion of Figure 3b in Section 4.1 and hope that together with the clarified experiment setup, the link between the empirical result and the insights is now clearer.
>
>
> 2. Figures 1a and 4 demonstrate that the softmax is responsible for a difference in the magnitudes of the elements of the self-attention Hessian blocks.
>     * For the softmax attention (figure 1a and the low-saturation plot in Figure 4), we see that the distribution of Hessian block entries varies between query and value blocks - the entries of the query block (green histogram on the right) are two orders of magnitude smaller than the entries of the value block (purple histogram on the left).
>     * After removing the softmax from the attention mechanism (fully saturated histograms in Figure 4) we see that both (green and purple) histograms are largely similar.
>
>     We extended the discussion of this experiment and how it relates to our theoretical findings in Section 4.1.

---

> > ### Comment · Reviewer_gfTd · 2024-11-28
> >
> > Thank you for the detailed answer and the revision of the paper.
> > - I appreciate the inclusion of all the additional details and explanations about the experiments into the paper. It makes the link between the theoretical and experimental results clearer.
> > - I didn't notice that the scale in Figure 3 was a log-log scale. Indeed, in that case, the linear behavior of the plots makes more sense. Nevertheless, I appreciate the increase in the range of $\sigma$ and the additional plots in linear scale.
> > - I also appreciate the discussion about the Hessian of multi-head self-attention in the appendix. It was mentioned by other reviewers, but I also think it is a valuable addition.
> >
> > I don't have any other concerns and think the paper is in a better shape now. I have increased my rating.

---

### Official Review · Reviewer_TFvz · 2024-11-01

**Soundness:** 3
**Presentation:** 4
**Contribution:** 3
**Rating:** 8
**Confidence:** 3

**Summary:**

This work derives the Hessian of Transformers to analyze the data dependence of different components in attention and to compare Transformers with other classical models.

**Strengths:**

$\bullet$ The derivation of the Hessian of Transformers provides a new framework of analyzing the dynamics of different components of self-attention.

$\bullet$ The discovery of the data dependencies among key, query, and value matrices is fundamental for future works, both in theoretical understanding and practical applications.

**Weaknesses:**

1. The omission of the $\textbf{F}$–functional Hessian blocks ($\delta_{XY}$) weakens the overall results, as the influence of $\delta_{XY}$ on the Hessian remains unclear, and there is no detailed discussion about its role.

2. The analysis is focused on a single self-attention layer and does not extend directly to more complex and practical Transformer architectures. The results are insightful but could benefit from further extensions to deeper, more realistic Transformer models.

3. There is no empirical comparison between Transformers and MLPs/CNNs. Including such empirical comparisons would make the findings more compelling and straightforward to interpret.

**Questions:**

1. How do you justify the omission of $\delta_{XY}$ in Equation (5)? If the elements of $\delta_{XY}$ are significantly larger than those of $X$, wouldn't the dependency on $X$ in Equation (5) become trivial?

2. Could you clarify the experimental settings used for Figure 4? You mentioned that Softmax significantly reduces the magnitude of the query Hessian block entries, but this effect isn't very apparent in Figure 4.

---

> ### Author Response · Authors · 2024-11-25
> **Response to Reviewer TFvz**
>
> Thanks a lot for your strong support! We answer your questions below.
>
> > The omission of the F–functional Hessian blocks ($\delta_{XY}$) weakens the overall results, as the influence of $\delta_{XY}$ on the Hessian remains unclear, and there is no detailed discussion about its role. [...] How do you justify the omission of  $\delta_{XY}$  in Equation (5)? If the elements of $\delta_{XY}$ are significantly larger than those of $\mathbf{X}$, wouldn't the dependency on $\mathbf{X}$ in Equation (5) become trivial?
>
> * $\delta_{X,Y}$ is part of the $\mathbf{R}$ matrices in Theorem 3.2. It encapsulates the derivative of the loss function w.r.t. the model output. We omit it from Equation 5 just for brevity.
> * For common loss functions it shouldn’t make the dependencies on $\mathbf{X}$ trivial:
>     * In the case of MSEloss, $\delta_{X,Y}$ introduces an additional dependence on $\mathbf{X}$, which we mention in Section 3.1. To see that, note that $\delta_{X,Y} = vec_r(\mathbf{F}(\mathbf{X}) - \mathbf{Y})$, where $\mathbf{F} = A\mathbf{X}W_V$, so its scale in Landau notation is driven by $\mathbf{X}$ since the attention scores are bounded by 1. If we now assume that $\mathbf{Y}$ is of a similar scale as $\mathbf{X}$, $\delta_{X,Y}$ scales linearly with $\mathbf{X}$, and brings in an additional dependence on $\mathbf{X}$ to non-zero functional Hessian blocks.
> * We don’t provide the theoretical derivation of this in the paper, but for the cross entropy loss (which we consider in our experiments),  $\delta_{X,Y}$ should be $\mathcal{O}(1)$, since $\delta_{X,Y} = vec_r(softmax(\mathbf{F}(\mathbf{X})) - \mathbf{Y})$. This is simply a gradient of the cross entropy loss applied to multiple sequence elements at once. In this expression, softmax is applied to the rows of $\mathbf{F}(\mathbf{X})$ and $\mathbf{Y}$ is a matrix of one-hot encoded targets. This explains why, in our experiments, we do not observe an additional dependence of the functional Hessian blocks on $\mathbf{X}$.
>
> > The analysis is focused on a single self-attention layer and does not extend directly to more complex and practical Transformer architectures. The results are insightful but could benefit from further extensions to deeper, more realistic Transformer models.
>
> Please, take a look at the general response for a discussion on how our results relate to the multi-layer setup.
>
> Moreover, please note, that:
> * Our experiments confirming the theoretical analysis concern the self-attention of a whole Transformer block (albeit mostly without layer normalization).
> * We discuss how other design choices like layer norm (see Figure 3b and discussion in Section 4.3) in Transformers influence the Hessian.
> * Following one of the other reviewer’s questions we also added a discussion of the multi-head attention Hessian to Appendix E.
>
> If there are other components of the Transformer architecture, the influence of which on the Hessian you would like us to comment on, please let us know.
>
> > Could you clarify the experimental settings used for Figure 4? You mentioned that Softmax significantly reduces the magnitude of the query Hessian block entries, but this effect isn't very apparent in Figure 4.
>
> * Figure 4 demonstrates the histogram of absolute Hessian block entries (value and query blocks on the left and right respectively) for a Transformer block with softmax self-attention (low saturation) and without linear self-attention (high saturation).
>
> * We added a more detailed description of the experimental setup in Appendix F and clarified its interpretation of Figure 4 in Section 4.1.
>
> * Please note the log scale on the OX axis in Figure 4 – there is a two-order magnitude difference between the modes of the empirical distributions of the query Hessian block entries for linear and classical self-attention. We believe that to be a significant difference.
>
> > There is no empirical comparison between Transformers and MLPs/CNNs. Including such empirical comparisons would make the findings more compelling and straightforward to interpret.
>
> We now include an empirical comparison between linear MLPs and linear self-attention networks in Appendix G. The comparison:
> * confirms and illustrates our claims from Section 4.1 that the Transformer Hessian is much more data-dependent than that of an MLP
> * highlights that this stark contrast only grows with depth.

---

### Official Review · Reviewer_xwuX · 2024-11-03

**Soundness:** 3
**Presentation:** 3
**Contribution:** 2
**Rating:** 5
**Confidence:** 3

**Summary:**

This paper derived the expression of the Hessian of one self-attention layer and discussed how the special structure of Hessian makes transformer special.

**Strengths:**

- This paper provides a detailed expression of the Hessian of self-attention, which might be useful for the community for the theoretical understanding of Transformers.
- The presentation is good. I especially appreciate that the authors write different symbols in different colors.

**Weaknesses:**

- Except the expression of the Hessian, I don't see any deep and detailed analysis in this paper. For example, the authors claim that understanding the structure of Hessian can help understand the optimization of Transformers, such as why Transformers have to be trained by Adam(W). However, I don't see any detailed discussion on this point in the paper. I would like to see a deeper discussion showing that how the stucture of Hessian derived in this paper connects to real Transformer behaviours.
- This whole analysis is based on a single-layer self-attention. it is unclear how this analysis (or the conclusions drawn from this one-layer model) can possibly extend to deeper models.

**Questions:**

See Weaknesses.

---

> ### Author Response · Authors · 2024-11-25
> **Response to Reviewer xwuX**
>
> We thank the reviewer for their feedback. We are glad they think that it might be useful to the community and that they appreciate the presentation.
>
> > Except the expression of the Hessian, I don't see any deep and detailed analysis in this paper.
>
> We respectfully disagree with the reviewer. One main contribution of our work is to interpret the components of our derived Hessian:
> * We isolate the self-attention moment matrices in the raw Hessian expressions (Section 3.2).
> * We analyze how different components scale with data and experimentally confirm that our predictions hold for the Transformer block without layer norm, not only the self-attention layer (Section 3.1).
> * We analyze how Transformer design decisions, like the use of softmax or parametrizing the attention matrix with two weight matrices, influence the Hessian (Section 4).
> * We also highlight the differences between the Transformer and MLP Hessian (Section 4.1).
>
> > [...] the authors claim that understanding the structure of Hessian can help understand the optimization of Transformers, such as why Transformers have to be trained by Adam(W). However, I don't see any detailed discussion on this point in the paper. I would like to see a deeper discussion showing that how the stucture of Hessian derived in this paper connects to real Transformer behaviours.
>
> The goal of the statement recalled by the reviewer is merely to motivate why we think that studying the exact structure of the Hessian brings value to the research community.
>
> For example, [1] provides evidence that Transformers train better with Adam because the spectra of their diagonal blocks are diverse. We now precisely know the algebraic structure of the blocks corresponding to matrices of a self-attention layer. Moreover, by connecting our results with the ones of Singh et al. (2021) we further characterize the structural differences between self-attention and the MLP blocks of the Transformer Hessian. This could be used to better characterize the block spectra and hopefully pinpoint the exact design components of Transformers that make optimization harder.
>
> In this paper, we claim that we identified sources of block heterogeneity in the self-attention Hessian, like the use of softmax and strong data dependence. While we believe this heterogeneity might transfer to the spectra, we did not directly study the Hessian block spectra. Hence, we cannot state a causal link between the sources we identified and the need for Adam, but this is definitely worth future investigation.
>
> > This whole analysis is based on a single-layer self-attention. it is unclear how this analysis (or the conclusions drawn from this one-layer model) can possibly extend to deeper models.
>
> Please see the general response on the further discussion of the multi-layer case.
>
> References:
>
> [1] Why Transformers Need Adam: A Hessian Perspective

---

> ### Author Response · Authors · 2024-12-01
>
> With only one day remaining in the discussion phase, we would like to kindly follow up to see if the reviewer has any further comments or feedback on our reply.
>
> We would also like to emphasize that the analyses presented in our work are non-trivial and require highly involved derivations. These efforts uncover intriguing aspects of the Transformer loss Hassian, like its block heterogeneity, heavy data dependence, dependence on the attention moment matrices, or the role of the softmax in the Hessian properties. With Transformers being so prevalent in today's landscape of machine learning, our Hessian calculations will be of fundamental utility for future theoretical and practical work.

---

### Official Review · Reviewer_jto3 · 2024-11-04

**Soundness:** 3
**Presentation:** 2
**Contribution:** 4
**Rating:** 8
**Confidence:** 4

**Summary:**

The paper compares the self-attention Hessian to classical networks such as CNN to better understand the unique optimization landscape  of self-attention based transformer architectures. The paper provides a understanding self-attention from hessian perspective, which is an interesting line to understand the inner workings of transformers. The empirical experiments on digit addition task validates the theoretical observations by considering CE loss.

**Strengths:**

1. The paper makes an attempt to understand self-attention based models using hessian analysis. This allows authors to compare transformers with architectures such as CNN.

2. The empirical evidence on digit addition task framed as next token prediction task validates the theoretical observations.

**Weaknesses:**

1.  The paper is not well written and is difficult to follow.

2. Authors should clearly state how their observations leads to better understanding of self-attention. It will also be beneficial for the readers if author mentions the consequences of their observations, such does it lead to better interpretability, or sparse attention or stable training.

3. In section 4.2 author discuss alternative to standard query-key parameterization and discusses change in loss landscape when single matrix W_{QK} is used instead of W_{Q}W_{K}^{\top}. Authors should discuss it briefly about how this change effects the overall performance in transformers, does it even make any difference in terms of overall performance for specific task or does it have any effect on interpretability of self-attention.

**Questions:**

Please answer the questions mentioned in previous section.

---

> ### Author Response · Authors · 2024-11-25
> **Response to Reviewer jto3**
>
> We thank the reviewer for their feedback. We address the specific parts of the review below.
>
> > The paper is not well written and is difficult to follow.
>
> We made the following changes to make the paper easier to follow:
> * For every figure presenting experimental results, we added (in bold) the main message that should be taken from the figure (see Figures 1,3,4 and the new figures in the appendix).
> * We added more emphasis on how our theoretical and empirical results relate to each other, especially the experiment behind Figure 4 in Section 4.1.
> * We changed the formatting and description of Figure 3 to make the interpretation of the results clearer.
> * We now highlight a summary of Section 3 in a box, making it easier for the reader to quickly locate the key insights.
>
> If there are any other things we could implement to improve the paper's clarity, please let us know, and we will be happy to incorporate them.
>
> > Authors should clearly state how their observations leads to better understanding of self-attention.
>
> We theoretically characterize fundamental differences in the Hessian of self-attention compared to other traditional architectures, specifically the strongly different role of query/key and value weights which is unique to this architecture. The Hessian is fundamental for understanding the behaviour of various phenomena (which we motivate in the introduction). Therefore, highlighting its characteristics is a meaningful step towards a deeper understanding of Transformers.
>
> > It will also be beneficial for the readers if author mentions the consequences of their observations, such does it lead to better interpretability, or sparse attention or stable training.
>
> * **Better interpretability and sparse attention.**  Sparse attention has an interesting influence on the Hessian. Consider the extreme case where some tokens attend to only a single token. The second and third central attention moments equal zero for such one-hot attention vectors. Therefore, the contribution of the query-key part of the Hessian diminishes because the query and key blocks are determined by these moments. This is discussed in the last paragraph of Section 3.2.
>
>     In the revision, we added a note that it leads to better interpretability.
>
> * **Stable training.** The Hessian’s heavy dependence on $\mathbf{X}$ can lead to unstable training, which highlights the importance of layer norm. We also believe that the Hessian’s heterogeneity we discover translates to its spectrum. Heterogenous block spectra have been linked with better performance of Adam vs SGD for Transformers [1].
>
> > In section 4.2 author discuss alternative to standard query-key parameterization [...] Authors should discuss it briefly about how this change effects the overall performance in transformers, does it even make any difference in terms of overall performance for specific task or does it have any effect on interpretability of self-attention.
>
> We run experiments on the Nanodo https://github.com/google-deepmind/nanodo language modelling setup, for 3 models having (non-embedding) parameters of about 2M, 10M, and 42M. Each of them were trained with AdamW on a subset of the C4 dataset [2] and the baselines evaluation loss with **classical attention** for these models were:
>
> (all numbers, for both attention types, are averaged over 3 seeds)
>
> * 2M: 3.91
> * 10M: 3.57
> * 42M: 3.22
>
> **Single matrix attention.** In the case of 2M, this gave 3.85, which in terms of loss is a significant improvement.
>
> For larger models such as 10M, this attention was on par and resulted in an evaluation loss of 3.59, while for 42M,  this was significantly worse resulting in an evaluation loss of 3.39.
>
> However, just as classical attention has $1/\sqrt{d_K}$ scaling inside softmax, we also experimented with scaling this single matrix attention by $1/\sqrt{d_{model}}$. Note, the previous results have no scale factor.
>
> With this scaling for the single matrix attention, the 42M evaluation loss improved to 3.15, outperforming the classical attention.
>
> **Conclusion.**
> Thus, we observe that the single matrix attention can perform, at least, just as good in the experimentation we could do in the discussion phase. While single matrix attention seems to also be slightly better, it must be noted that single matrix attention can also result in more parameters, since each head has now $d_{model}^2$ parameters instead of $2 * d_{model} * d_K$ as for classic attention. This might be the underlying factor for slight improvement, but these results suggest that it could be a promising direction for future exploration.
>
> In the revision, we commented on the effect of the choice of parametrization on the interpretability of self-attention.
>
> References:
>
> [1] Why Transformers Need Adam: A Hessian Perspective
>
> [2] Documenting Large Webtext Corpora: A Case Study on the Colossal Clean Crawled Corpus

---

> ### Comment · Reviewer_jto3 · 2024-11-26
> **Response to Author Comments**
>
> Thanks for the response. Please see the following points in regard to author's rebuttal
>
> 1.  The changes made to the paper do make the results easier to understand.
>
> 2. In section 3.2 L338-L347 authors states that "If the attention scores are highly dispersed across different tokens, the query-key outer product Hessian will dominate. Conversely, if the attention scores were data-independent,2 which happens almost surely at initialization for large $d_K$ sequences with more similar words will result in a lower contribution from the query-key block".
>
> Is the above observation for a particular sequence with multiple similar words or across the sequences with similar words. This seems counterintuitive as we apply self-attention, so that similar words have higher attention weights, and thus have similar contextual representations. Please answer this.
>
> 3. The performance improvement results in case of single matrix attention are interesting and valuable.
>
> Author's have answered all my queries, and I think the paper will be good addition to the conference. Thus i have increased my rating for the paper.

---

> > ### Author Response · Authors · 2024-12-01
> >
> > We thank the reviewer for reading our rebuttal, revising the rating, and further feedback. Please, see our answer to your question below.
> >
> > > [...] Is the [...] observation for a particular sequence with multiple similar words or across the sequences with similar words. This seems counterintuitive as we apply self-attention, so that similar words have higher attention weights, and thus have similar contextual representations.
> >
> > * The observation concerns a single sequence. Having a single sequence and its attention scores, we can define the attention moment matrices as in Definition 3.1. In the case of the query and key Hessian blocks, we are specifically interested in the second and third central moments of attention $M_2, M_3$.
> > * Let us clarify the setting for these two observations:
> >     * “If the attention scores are highly dispersed across different tokens, the query-key outer product Hessian will dominate.”
> >
> >          Here we answer a question: Given a fixed input sequence, how do changes in attention scores (possibly caused by varying values in $W_K$ and $W_Q$) influence the second and the third central moment matrices $M_2, M_3$? These matrices directly influence the query and key Hessian blocks, by being part of $Z_1$ and $Z_2$ (see Theorems 3.1 and 3.2).
> >     * “... if the attention scores were data-independent, [...] more similar words will result in a lower contribution from the query-key block”
> >
> >         Here we assume that we are given some data-independent attention scores (like under the uniform attention assumption, which happens almost surely *at initialization* for large enough $d_K$ [3]). Under this assumption, we comment on what happens with the attention higher central moment matrices for similar $X_{i,:}$. For example, if elements of $X$ are similar, we would expect the entries of their second central moment matrix (think variance) to be small.
> >
> >     In the last revision, we updated these two sentences to clarify the setting.
> >
> > * We agree with the reviewer that similar words will likely have higher attention weights (in a *trained* model) and similar contextual representations. However, please note that similar contextual representations should mostly influence the first attention moment matrix $M_1 = AX$, which explicitly encapsulates them. $M_1$ does not directly influence the query-key Hessian block (it only enters as part of the higher central moments), but the value block (see Theorems 3.1 and 3.2).
> >
> > [3] Signal Propagation in Transformers: Theoretical Perspectives and the Role of Rank Collapse.

---

### Official Review · Reviewer_ikrU · 2024-11-05

**Soundness:** 4
**Presentation:** 4
**Contribution:** 2
**Rating:** 6
**Confidence:** 3

**Summary:**

This paper derives and analyzes the Hessian matrix of a single Transformer self-attention layer. It examines how the Hessian depends on the data, the weights, and the attention mechanism's internal moments. The Hessian is found to be highly non-linear and varies significantly across different parts of the self-attention layer. This variation is caused by the way data enters the attention mechanism as keys, queries, and values. It is also due to the softmax function and how the attention mechanism's query and key components are parameterized. These factors create complex relationships between the data, weights, and the Hessian. The authors believe this analysis helps explain why Transformers have a unique optimization landscape. They also suggest it explains why certain architectural choices, such as using adaptive optimizers and layer normalization, are beneficial for training Transformers.

**Strengths:**

- This paper tackles an important theoretical question regarding the dynamics of Transformers by directly analyzing the Hessian.
- A thorough theoretical derivation and analysis like this is novel and provides a valuable new perspective.
- The categorization of Hessian dependencies offers a structured framework for understanding the complex interactions within the architecture.
- The derivations appear sound and are presented with sufficient detail.
- The exploration of how different Transformer components impact the Hessian adds depth and rigor to the study.
- The paper is well written and is generally a pleasure to read. The authors incorporate the existing literature nicely. While the Hessian structure is inherently complex, the authors have made a good effort to explain the key takeaways in an accessible way.

**Weaknesses:**

- The paper analyses only single layer, without saying much about multi-layer.
- A lot of important aspects are not addressed, e.g. multi-layer, role of residual connection in the Hessian, multi-head attention. Additionally, can you comment on the implications of (W_V) often being a low-rank matrix with rank (d_k)?
- The paper doesn't have a solid narrative and rather presents a reader with a bag of tricks. See some of the examples in the Question section below. It also makes claims that are not justified, e.g. that it can help  explaining the performance gap between Adam and SGD in lines 516-519.
- To strengthen the paper's narrative, the author should have started with the analysis of the gradient before delving into the Hessian, since it is much simpler. Comparing and contrasting the properties of the gradient and Hessian could provide a more comprehensive understanding.

**Questions:**

- In Figure 3, several plots show a mismatch between the predicted and observed Hessian scaling. The top right plot in Figure 3a doesn't display a prediction at all. Could the authors elaborate on these discrepancies?
- Some analysis is presented more like a log book without explaining why is it important. For example, what are the key takeaways from Figure 4? More broadly, could the authors clarify the overarching message and how the different analyses contribute to it?
- The paper claims to provide a Hessian-based perspective on the performance gap between Adam and SGD, referencing Ahn et al. (2023). However, this explanation isn't explicitly provided in the paper. Could the authors elaborate on this point and clarify how their analysis explains this performance gap?

---

> ### Author Response · Authors · 2024-11-25
> **Response (1/3) to Reviewer ikrU**
>
> Thanks for your feedback. We are glad you found the questions we study in this paper important, our theoretical derivations thorough, and our analysis novel. We address the point on the multi-layer setup in our global response.
>
> > A lot of important aspects are not addressed, e.g.[...], role of residual connection in the Hessian, multi-head attention.
>
> * We understand your point. The Transformer architecture is complex, which complicates its theoretical analysis. So we had to start our analysis from the basic building block, which, in our opinion, is a single self-attention layer. We tried to cover additional design components through our experiments too, to probe whether our analysis extends to fully-fledged Transformers. E.g., we used a single GPT-2 Transformer block (with skip connections), not just a single self-attention layer. Still, our theoretical predictions were well-aligned with the experimental data (see Figure 3).
>
> * For a single self-attention layer, the residual connection does not introduce new dependencies in the Hessian, as it skips the block where the parameters are used.
>
> * **Multi-head self-attention.** We agree that multi-head self-attention is relevant to the paper. We added a discussion of the multi-head case in Appendix E in the revision. Our Theorems 3.1 and 3.2 directly apply to the weights of an individual head, and the main takeaway is that with multiple heads the functional Hessian w.r.t. combined heads becomes block-sparse, because heads process data independently. Please, let us know in case you have any further questions.
>
> > Additionally, can you comment on the implications of (W_V) often being a low-rank matrix with rank (d_k)?
>
> * We assume that the reviewer means a further analysis of the case when a low-rank structure on $W_V$ is imposed through a parametrization with two matrices $W_V = W_OW_U$, where $W_O \in \mathbb{R}^{d_V, d_K}$ and $W_U \in \mathbb{R}^{d_K,d_V}$, which is frequently seen in the multi-head attention setting ($W_U$ and $W_O$ would correspond to $W^V$ and $W^O$ in [4]).
> * In this case, the Hessian diagonal block corresponding to these matrices exhibits the structure from [5], because this parameterization is basically a two-layer linear MLP discussed in [5],  i.e. the functional Hessian is a block-hollow matrix.
> * We also added the discussion of the Hessian in this parametrization and the emerging scaling laws to the manuscript in Appendix E (see the last paragraph).  The main conclusions are:
>     * compared to the parametrization with a single matrix, the functional Hessian is not zero (but instead block-hollow) and brings a dependence on $M_1$,
>     * the outer product Hessian now additionally depends on the value weight matrices $W_O$ and $W_U$.
>
> > To strengthen the paper's narrative, the author should have started with the analysis of the gradient before delving into the Hessian, since it is much simpler.
>
> You are right that the attention layer’s gradient is also interesting to study. This has been done both theoretically and empirically in other papers [1, 2, 3]. [1] provides the starting point for our work as mentioned in Lemma B.1. Our goal is to go beyond the gradient and study the Hessian, as it is a fundamental object related to optimization, generalization (think sharpness), and provides the rate at which the gradient changes.

---

> ### Author Response · Authors · 2024-11-25
> **Response (2/3) to Reviewer ikrU**
>
> > [The paper] also makes claims that are not justified, e.g. that it can help explain the performance gap between Adam and SGD in lines 516-519. [...] The paper claims to provide a Hessian-based perspective on the performance gap between Adam and SGD, referencing Ahn et al. (2023). However, this explanation isn't explicitly provided in the paper. Could the authors elaborate on this point and clarify how their analysis explains this performance gap?
>
> * Ahn et al. (2023) show that one can recreate many Transformer-specific phenomena with just two layers of linear self-attention. We mention the superior performance of Adam over SGD as one of the phenomena reported in Ahn et al. (2023).
> * Our work is complementary to theirs - we theoretically show how simplifications like removing softmax or using a single matrix to parametrize $A$ influence the Hessian - an object frequently studied to understand optimization and loss landscape, hence providing valuable additional context.
>
> * For example:
>     * They show that the gap in loss between adaptive methods and SGD becomes more and more pronounced with the increased network depth.
>     * We discuss that the dependence of the Hessian of linear self-attention network on $\mathbf{X}$ scales super-exponentially with depth (end of Section 4.1).
>     * This suggests a possible link between the heavy data dependence and the superiority of Adam. Of course, this requires further investigation.
>
> * By describing fundamental differences in the Hessian of Transformers and MLPs, we generate new hypotheses as to why Adam may be more suitable than SGD to train Transformers because the Hessian *is* fundamental for optimization.
>
> * We modified the lines you mentioned to Our work provides a Hessian-based perspective on this model, providing
> new hypotheses which components may drive Transformer optimization challenges.”.  We hope this clarifies how our work contributes to understanding Transformer optimization.
>
> > In Figure 3, several plots show a mismatch between the predicted and observed Hessian scaling. The top right plot in Figure 3a doesn't display a prediction at all. Could the authors elaborate on these discrepancies?
>
> * We updated Figure 3, by extending the range of $\sigma$ as per request of reviewer gfTd. The only plots that should follow the predicted trend are the ones in Figure 3a, and please note that they do, especially for more practical values of $\sigma < 1$:
>     * the value outer product Hessian depends on $\mathbf{X}^2$ and its Frobenius norm scales quadratically with $\sigma$,
>     * the query outer product and functional Hessian depend on $\mathbf{X}^6$ and $\mathbf{X}^5$ respectively and their Frobenius norms scale accordingly with $\sigma$.
>
> * Figure 3b shows trends for a more practical version of the Transformer that does not satisfy all our assumptions. As noted in the figure description, the trends displayed in Figure 3b are not theoretical but estimated from the measured Frobenius norm values $\bar{f}(\sigma)$, by fitting linear regression to points $(\log(\sigma), \log(\bar{f}(\sigma)))$ (for details, see Appendix F). We updated the Figure 3 description and changed the line style of the trends to make that clearer.
>
> * The top right plot in Figure 3a doesn’t display a prediction, because the value functional Hessian equals zero (which is technically an $\mathcal{O}(1)$ dependence on $\mathbf{X}$, which we denote in Equation 5). Note that this is exactly what our Theorem 3.2 predicts for this Hessian block. In the revision we removed the top right panel from Figure 3a, because we agree that it might be confusing for the reader not to see a prediction. Instead, we added a sentence about it to the figure description and to the discussion of Figure 3a.
>
>
> > what are the key takeaways from Figure 4?
>
> The key takeaway from Figure 4 is that softmax causes the heterogeneity between the entries of the value and query block Hessians (purple and green histogram respectively). Softmax self-attention (low-saturation histograms) has query and value Hessian blocks whose entries differ by orders of magnitude. Removing softmax makes them more similar (linear self-attention, high-saturation histograms).
>
> All figure captions now state the plot’s takeaway in bold. We hope that this makes them clearer.

---

> ### Author Response · Authors · 2024-11-25
> **Response (3/3) to Reviewer ikrU**
>
> > could the authors clarify the overarching message and how the different analyses contribute to it?
>
> There are two main but connected messages from this paper:
>
> 1. The self-attention Hessian has a heterogenous algebraic structure which translates to varied scaling laws across Hessian blocks.
> 2. The self-attention Hessian is vastly different than the Hessian of other well-studied architectures, like MLPs and CNNs.
>
> To support (1) we:
> * derive the precise expressions for self-attention loss Hessian (Theorems 3.1 and 3.2) and then
> * interpret the expressions by:
>     * analyzing their scaling laws w.r.t. the scale of the input (Section 3.1, Figure 3a),
>     * noticing their varied dependence on the attention moment matrices (Section 3.2) and weight matrices (Section 3.3).
> * identify the culprit behind these scaling laws as the softmax activation (Section 4.1).
>
> To support (2) we relate our results to those for MLPs and CNNs throughout the paper. For example:
> * When introducing Theorems 3.1 and 3.2 we comment on the similarity between MLP Hessian and the value Hessian block.
> * We discuss how components not present in MLPs and CNNs impact the Hessian structure; specifically, how much more data-dependent the Transformer Hessian is (Section 4.1).
>
> References:
>
> [1] Signal Propagation in Transformers: Theoretical Perspectives and the Role of Rank Collapse
>
> [2] Understanding the Difficulty of Training Transformers
>
> [3] On Layer Normalization in the Transformer Architecture
>
> [4] Attention Is All You Need
>
> [5] Analytic Insights into Structure and Rank of Neural Network Hessian Maps

---

> > ### Comment · Reviewer_ikrU · 2024-11-26
> >
> > Thank you for the responses. I'll adjust my score.
> >
> > To enhance readability, I strongly suggest adding one or two paragraphs adding the formulae for the gradient and its relationship to the Hessian findings presented in the paper. Connecting these terms explicitly would significantly improve understanding for a broader audience. While this addition wouldn't necessarily increase the novelty of the work, it would make the paper more readable.
> >
> > Please replace the term "scaling laws" with "asymptotic growth rate" throughout the paper. "Scaling laws" carries a very specific meaning within the research community, which doesn't seem to align with the authors' intended usage.
> >
> > Why not lower score: The authors have diligently addressed many of the questions and concerns raised by myself and other reviewers. The paper presents valuable information and will likely be of interest to researchers exploring the application of the Hessian in analyzing attention optimization.
> >
> > Why not higher score: While the paper contains numerous interesting observations, it currently lacks a strong central conclusion or actionable insights that could directly guide further analysis of transformers.

---

> > > ### Author Response · Authors · 2024-12-01
> > >
> > > We thank the reviewer for reading our response, revising the rating, and further suggestions.
> > > * In the last revision, we replaced the term “scaling laws” with ”asymptotic growth rate”/“growth rate”.
> > > * In the final version of the manuscript, we will include a discussion of the gradient and its relation to the Hessian in the appendix and reference it in the main text. We cannot add the formulae and the additional discussion to the main part of the paper because it would require removing some other content.

---

### Author Response · Authors · 2024-11-25
**General Response to All Reviewers**

We thank all reviewers for their thoughtful feedback and constructive comments. We are pleased that the reviewers find that our paper “tackles an important theoretical question” [ikrU] and that our discoveries are “useful for the community” [xwuX], “fundamental for future works” [TFvz] and “serv[ing] as foundation for better understanding the role of the self attention” [gfTd].

We reply to each of the reviewers’ questions and concerns individually below. To track the significant changes to the manuscript, we colored any additions to the PDF in blue if they go beyond minor edits.

In our global response, we address the multi-layer scenario that was requested by some of the reviewers.

**Multi-layer setting:**
* Our goal was to derive and interpret in detail the closed-form expressions for the Hessian. Following other works on a single layer or a very shallow Transformer (for example [1, 2, 3]), we believe a single self-attention layer is an interesting object worth studying. Our work serves as an important building block for extending the theory to multiple layers. You have a point that we currently can only address certain scenarios with depth (see below). We highlighted this limitation of our theory in the revision.

* To facilitate the discussion of the multi-layer Transformer Hessian, we empirically check what scaling laws can be observed for its blocks:
    * New in Appendix G: Empirically, we find out that for a multi-layer GPT-2 Transformer and practically relevant ranges of $\sigma$ we can apply our block scaling laws **to every single layer in isolation**. We added the figures and the discussion of this experiment to Appendix G (Figure 6).

* Our theoretical claims extend to some extent into the deep case. We made the following additions:
    * New in Appendix G: We empirically verified the discussion of depth in a Transformer with linear attention from L445-451 and also compared it with a deep linear MLP. As predicted by our theory, we observe the super-exponential (with depth) data dependency for the Transformer (Figure 7), in contrast to the constant (with depth) dependency from the MLP (Figure 8).
    * Theorems 3.1 and 3.2 directly apply to the last attention layer in a self-attention net, by replacing $\mathbf{X}$ with the second to last layer’s output. Similarly, parts of our analysis carry over to any self-attention layer in a deep Transformer by applying the chain rule between the layer and the remainder of the network. We have added these comments as a footnote in the main text.

References:

[1] Understanding Addition in Transformers

[2] Are Transformers with One Layer Self-Attention Using Low-Rank Weight Matrices Universal Approximators?

[3] Linear attention is (maybe) all you need (to understand Transformer optimization)

---

### Meta-Review · Area_Chair_Nf8o · 2024-12-19

**Metareview:**

The paper provides a theoretical analysis of the Hessian matrix of a (single) attention layer in transformer architectures. This analysis is highly relevant, as it opens up characterization of the relationship between data, weight and attention. In turn, new insights are obtained on why transformers may be particularly amenable to e.g. adaptive optimizers or layer normalization, but in more general why they have a unique optimization landscape different from other deep networks. Reviewers all agreed that this is a fundamental and important contribution to a field where transformers are prevalent in usage. The contributions of the paper thus lay out theoretical building blocks to further understanding and open up the way for more theory to be developed. Apart from some concerns on clarity and writing, that were mostly resolved throughout the discussion phase, a common reviewer concern was the focus of the analysis on a single layer. The authors have conducted tangible revision steps to ensure that their proposed growth rates are nevertheless valuable for the analysis of multiple layers and that they may empirically hold up. Overall, reviewers are happy to accept the paper to the conference. As a consequence of the significance of the contribution, the reviewers’ positive feedback, and the constructive revisions, the AC proposed to accept the paper. Because the paper provides a well-written analysis on a highly popular topic, the AC further suggests the paper for an oral presentation, as it is likely that a large part of the community will benefit from the obtained insights.

**Additional Comments On Reviewer Discussion:**

Reviewers acknowledged the paper’s contribution from the start and unanimously agreed on the value of the contribution. However, there were various initial concerns on the readability, clarity, and practical take-aways of the paper. Many of these concerns have been addressed in the discussion phase and have been updated in the paper. The reviewers who engaged in discussion acknowledged that this improved the paper. Reviewer xwuX, with the overall lowest yet borderline score, did not provide a very detailed initial review and did not engage with the responses/updates. One of the main concerns seems to have been that the reviewer thought the analysis to be limited due to the focus on a single layer. This concern was in parts shared by other reviewers, but the AC agrees with other reviewers that tangible additions have been made to the paper to clarify this concern and open up an extended analysis. The AC thus believes that the reviewer’s concern has been addressed in practice. The other three reviewers all agree to accept the paper.

---

### Decision · Program_Chairs · 2025-01-22

Accept (Spotlight)